# Evolutionary histories of breast cancer and related clones

Tomomi Nishimura[1,2,3,26], Nobuyuki Kakiuchi[1,4,5,26], Kenichi Yoshida[1,6], Takaki Sakurai[7,8], Tatsuki R. Kataoka[7,9], Eiji Kondoh[10,11], Yoshitsugu Chigusa[10], Masahiko Kawai[12], Morio Sawada[13], Takuya Inoue[13], Yasuhide Takeuchi[1,7], Hirona Maeda[1,7], Satoko Baba[14,15,16], Yusuke Shiozawa[1], Ryunosuke Saiki[1], Masahiro M. Nakagawa[1,2], Yasuhito Nannya[1,17], Yotaro Ochi[1], Tomonori Hirano[1,5,18], Tomoe Nakagawa[1,18], Yukiko Inagaki-Kawata[1,3], Kosuke Aoki[1], Masahiro Hirata[7], Kosaku Nanki[19], Mami Matano[19], Megumu Saito[19,20], Eiji Suzuki[3,21], Masahiro Takada[3], Masahiro Kawashima[3], Kosuke Kawaguchi[3], Kenichi Chiba[22], Yuichi Shiraishi[22], Junko Takita[12], Satoru Miyano[23,24], Masaki Mandai[10], Toshiro Sato[19], Kengo Takeuchi[14,15,16], Hironori Haga[7], Masakazu Toi[3,27] & Seishi Ogawa[1,18,25,27 ✉]

Recent studies have documented frequent evolution of clones carrying common cancer mutations in apparently normal tissues, which are implicated in cancer development[1–3]. However, our knowledge is still missing with regard to what additional driver events take place in what order, before one or more of these clones in normal tissues ultimately evolve to cancer. Here, using phylogenetic analyses of multiple microdissected samples from both cancer and non-cancer lesions, we show unique evolutionary histories of breast cancers harbouring der(1;16), a common driver alteration found in roughly 20% of breast cancers. The approximate timing of early evolutionary events was estimated from the mutation rate measured in normal epithelial cells. In der(1;16)(+) cancers, the derivative chromosome was acquired from early puberty to late adolescence, followed by the emergence of a common ancestor by the patient's early 30s, from which both cancer and non-cancer clones evolved. Replacing the pre-existing mammary epithelium in the following years, these clones occupied a large area within the premenopausal breast tissues by the time of cancer diagnosis. Evolution of multiple independent cancer founders from the non-cancer ancestors was common, contributing to intratumour heterogeneity. The number of driver events did not correlate with histology, suggesting the role of local microenvironments and/or epigenetic driver events. A similar evolutionary pattern was also observed in another case evolving from an *AKT1*-mutated founder. Taken together, our findings provide new insight into how breast cancer evolves.

Revolutionized sequencing technologies have enabled a series of recent studies on clones evolving in apparently normal tissues, which are documented by sensitively detecting clone-defining somatic mutations[1–7]. In view of cancer development, a key observation through these studies is that clonal outgrowth in normal or non-cancer tissues is common, often pervasive and frequently driven by common cancer mutations[1]. This immediately points to an important implication to the early history of cancer that one or more of those positively selected non-cancer clones should be destined for subsequent cancer development[1,8]. Here, among key questions that studies on normal tissues cannot answer are when cancer arises from these non-cancer clones by acquiring what additional mutations, while other clones partially sharing common mutations are still normal or precancer, and what the difference in mutation profile is between cancer clones and those non-cancer relatives. Phylogenetic analyses using multisampling of cancer specimens have been used to infer the life history of cancer in terms of driver events. However, analyses of cancer tissue alone frequently obscure the order of early driver events that are often assigned together to a long major trunk in the phylogenetic tree[9–11]. Moreover, they do not help map the timing at which cancer clones emerged or track the fate of other related non-cancer clones. To answer these issues, analyses of both cancer and non-cancer lesions are absolutely needed, although these are frequently hampered by the fact that at the time of cancer diagnosis or surgery, genetically related non-cancer clones are probably swept out by rapidly expanded cancer clones[8,12,13].

Breast cancer is one of the most prevalent cancers among women. Annually, 2.3 million women are diagnosed with breast cancer worldwide, with 685,000 deaths reported in 2020, and the incidence is still increasing in many countries[14,15]. Whereas so far there have been no large studies of clonal expansion in normal breast tissues, several reports have shown that, in some patients, cancer is accompanied by satellite benign breast lesions (BBLs), such as proliferative lesions with and without atypia, wherein both lesions share common genetic alterations and hence a common ancestor[16–20]. The phylogenetic analysis of

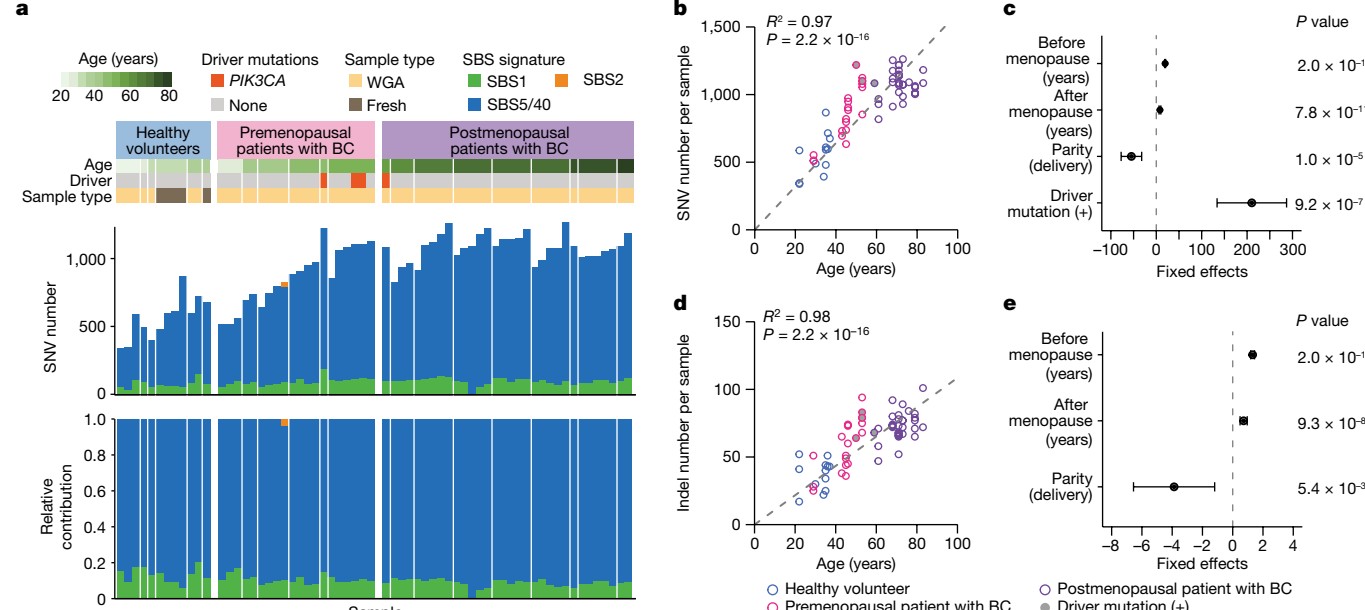

**Fig. 1 | Mutations in normal mammary epithelium. a**, Summary of SNVs found in 64 single-cell-derived organoids established from six healthy premenopausal breastfeeding women (healthy volunteers) and six premenopausal and nine postmenopausal patients with breast cancer (BC). Information about participant's age, driver mutations and the presence or absence of WGA, are shown in the top panel. The stacked bar plots in the middle and bottom panels show the number of mutations and the proportion of indicated mutational signatures, respectively. **b,d**, The number of SNVs (**b**) and indels (**d**) in organoids (n = 64) are plotted for participant's age. Regression lines assuming a zero intercept are applied to mean number of mutations for each participant (n = 21) and age, with $R^2$ and P values from the two-sided F-test (grey dashed lines). **c,e**, Linear regression models were applied to 61 organoids with information on age at menopause and parity. Estimates of coefficients that significantly affect the number of mutations in the linear regression model are shown for SNVs (**c**) and indels (**e**), with 95% CI and P values from the two-sided t-test.

such cases containing both cancer and associated BBLs could provide a unique opportunity to decipher the entire life history and evolutionary dynamics of breast cancer, which are instrumental in understanding when and where the breast cancer ancestor is born and evolves to acquire the cancer phenotype, and also to establish new strategies to predict or even prevent breast cancer development[21,22].

In the current study, we performed whole-genome sequencing (WGS) of multiple microscale samples obtained by laser-capture microdissection (LCM) from both cancer and clonally related BBLs, together with apparently normal lobules. Then, on the basis of the rate of mutation accumulation estimated from WGS of single-cell-derived organoids established from mammary epithelia, we reconstructed the phylogenetic trees including both cancer and non-cancer clones to infer the entire history of breast cancer.

## Mutations in mammary epithelium

To estimate the rate of mutation accumulation in normal mammary epithelial cells with ageing, we established 71 single-cell-derived organoids from epithelial cell adhesion molecule (EpCAM)-positive cells isolated from histologically normal mammary tissues in patients with breast cancer or from breast milk provided by healthy volunteers who were breastfeeding (Methods, Extended Data Fig. 1a, Supplementary Note 1 and Supplementary Fig. 1). Excluding six organoids for which polyclonal origins were suspected and another with a germline *CDH1* mutation, we finally evaluated somatic mutations for 64 organoids established from six pre- (n = 20) and nine postmenopausal (n = 32) patients with breast cancer and 12 from six healthy women (Fig. 1a and Extended Data Figs. 1 and 2). The mutation distribution in 15 organoids showed a minor subpeak at a variant allele frequency (VAF) less than 0.25 corresponding to subclonal mutations acquired during cell culture, which were eliminated assuming a Gaussian mixture model (Methods). We identified a total of 58,385 single nucleotide variants

(SNVs) and 3,955 small insertions and deletions (indels) as clonal somatic variants, including four *PIK3CA* mutations (Fig. 1a). When fit to the known Catalogue Of Somatic Mutations In Cancer (COSMIC) single base substitution (SBS) signatures, most SNVs were assigned to three clock-like signatures[23], SBS1 (9.9%), SBS5 (80.7%) and SBS40 (9.4%). SBS1 is characterized by the prominence of C>T transitions at CpG dinucleotides resulting from the spontaneous deamination of 5-methyl-cytosine[24], whereas SBS5 and SBS40 are 'flat' signatures of unknown aetiology[24,25], which are difficult to separate from each other and, hence, designated collectively as 'SBS5/40' in the subsequent analyses. According to the linear regression model, the number of SNVs significantly depended on age at sample collection, years after menopause, parity and the presence of a driver mutation. SNVs were accumulated at 19.5 mutations per genome per year before menopause, which was reduced to 8.1 mutations per genome per year after menopause, while the mutation number was reduced by 54.8 per delivery (Fig. 1b,c). The mutation rate was also affected by *PIK3CA* mutations, which increased the number of SNVs by 210.4, although this needs to be validated using additional *PIK3CA*-mutant clones, because the number of driver-mutated samples was still small (n = 4). The accumulation rate of indels was also reduced by 45%, from 1.3 mutations per genome per year before to 0.72 mutations per genome per year after menopause, while the mutation number was spared by 3.9 per delivery (Fig. 1d,e).

## History of breast cancer evolution

The evolutionary history of breast cancer was then investigated using phylogenetic analysis using those surgical specimens that contained both pathologically confirmed cancer and multifocal (three or more) non-cancer proliferative lesions (greater than or equal to 3 mm in diameter). After reviewing pathology reports of 156 patients with breast cancer, we found five patients for whom formalin-fixed paraffin-embedded (FFPE) specimens fulfilling the above-mentioned criteria had been

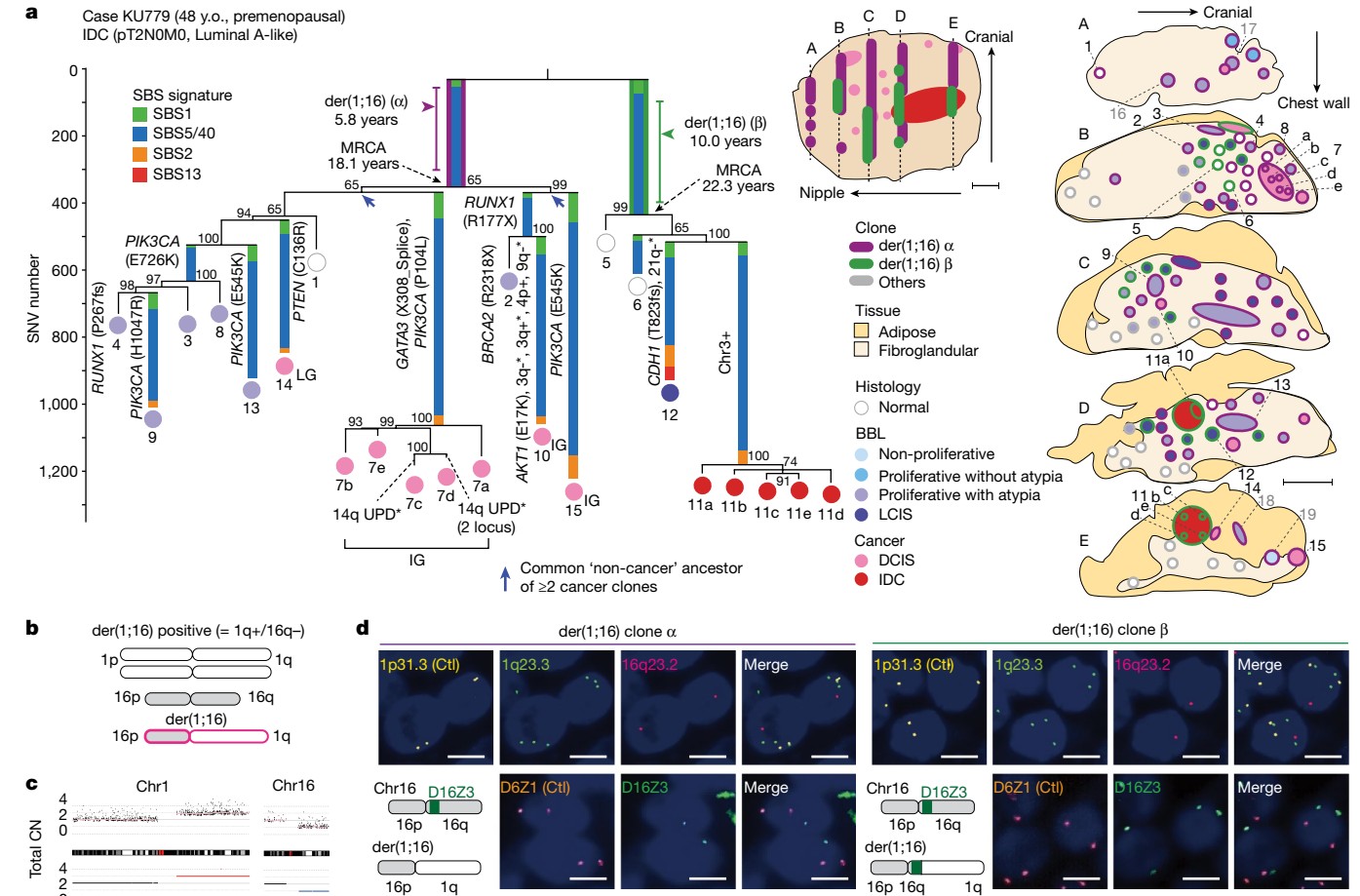

**Fig. 2 | Clonal evolution of breast cancers. a**, Phylogenetic tree (left) and corresponding geographical maps of clones detected in the surgical specimens of a patient with breast cancer who underwent lumpectomy (KU779) (middle, an overview of the surgical specimen; right, split faces of the sliced specimens indicated by dotted lines in the overview image). Signature extraction was performed on branches with more than 100 SNVs, and each branch is coloured according to the proportion of SBS signatures. Bootstrap values (%) are shown in grey. Driver mutations and CNAs (*focal changes) are shown on each branch. Estimated timing of der(1;16) acquisition and MRCA emergence are shown, with 95% CI for der(1;16) acquisition. Colours inside the circles indicate histological results. Numbers indicate samples in which each clone representing the tip of tree was identified. Grade of DCIS is shown in the tree (LG, low grade; IG, intermediate grade). Blue arrows indicate common 'non-cancer' ancestors of

two or more cancer clones. Colours around the circles in split faces depict clones to which lesions belong. Each circle was analysed by WGS (black numbers) or targeted sequencing (grey numbers), and/or FISH (unnumbered). UPD, uniparental disomy. Scale bars, 10 mm; y.o., years old. **b,c**, Schema of der(1;16) (q10;p10), which leads to concurrent whole-arm 1q gain and 16q loss (**b**); copy-number plots on chromosomes 1 and 16 (no. 7d in KU779) are shown in **c**. **d**, Representative FISH images of der(1;16) clones α (out of 62 lesions) and β (out of 22 lesions) in KU779 (nos. 13 and 12, respectively). The probe set of 1p.31.3, 1q23.3 and 16q23.2 (top row) was designed to detect all types of der(1;16)(+) clone, wherein 1p.31.3 signals were used as controls (Ctl). The probe set of D6Z1 and D16Z3 (bottom row) was designed to distinguish between clones α and β in KU779, wherein D6Z1 signals were used as Ctl (schemas are shown at the bottom left). Scale bar, 5 µm.

preserved. We obtained multiple microscale samples (2.9 mm (0.5–10) in diameter on average) from both cancer and non-cancer lesions (13.8 per patient) using LCM (Extended Data Fig. 3). We analysed a total of 69 LCM samples for somatic mutations and copy-number alterations (CNAs) using WGS, based on which we reconstructed phylogenetic trees (Methods, Supplementary Notes 1 and 2 and Supplementary Figs. 2 and 4). These samples comprised histologically normal lobules ($n = 6$), non-proliferative ($n = 1$) and proliferative lesions ($n = 33$), classic-type lobular carcinoma in situ (LCIS) ($n = 1$), ductal carcinoma in situ (DCIS) ($n = 20$) and invasive ductal carcinoma (IDC) ($n = 8$). All patients were in their 40s, premenopause and carried Luminal A-like IDC ($n = 3$) (KU539, KU779 and KU957) or oestrogen receptor (ER) (+) human epidermal growth factor receptor 2 (HER2) (−) DCIS ($n = 2$) (KU582 and KU873) (Supplementary Table 1). As was the case with normal organoids, the mutations in non-cancer lesions were dominated by SBS1 and SBS5/40 mutations (Fig. 2a and Extended Data Figs. 4 and 5). A minority of lesions had varying contributions from APOBEC-related signatures (that is, SBS2 and SBS13, ref. 23) in peripheral branches, which supports the

role of APOBEC-induced mutagenesis during relatively late phases of carcinogenesis (Extended Data Fig. 4c).

In all five cases, the phylogenetic trees comprised just one or two large clades, in which frequently acquiring unique driver alterations of their own, multiple progenies derived from a single common ancestor evolved to give rise to both cancer and BBLs. These progenies expanded over years and by the time of cancer diagnosis, occupied a large area in the affected breast, replenishing both cancer and non-cancer tissues as well as normal ones (Fig. 2a and Extended Data Fig. 5). Conspicuously, in four of the five cases, the most recent common ancestors (MRCAs) in each clade harboured der(1;16) (Fig. 2 and Extended Data Fig. 5a–c). Moreover, in KU779 and KU539, there were two independent der(1;16)(+) clones having distinct breakpoints, suggesting a strong selective advantage conferred by der(1;16) (Fig. 2a,d and Extended Data Fig. 5a). der(1;16) is a recurrent abnormal chromosome highly characteristic of Luminal A breast cancer, particularly those with invasive lobular histology (ILC)[26] and generated by an unbalanced translocation that fuses chromosome 1q and 16p arms near centromere sequences, that is, der(1;16)(q10;p10),

leading to 1q gain and 16q loss in common (Fig. 2b,c). The widespread distribution of der(1;16) clones was further investigated using fluorescence in situ hybridization (FISH) by detecting unbalanced chromosomal copy numbers of 1q and 16q arms. Showing varying histologies, including non-cancer proliferative lesions, LCIS, DCIS and IDC, the der(1;16)(+) clones were widely distributed within the mammary gland, spanning 62 mm (range 35–90 mm) regions on average (Fig. 2a and Extended Data Figs. 3c and 5a–c). der(1;16) signals were also detected in histologically normal lobules (Fig. 2a and Extended Data Figs. 3c and 5b).

In line with previous reports[9,10], all cancer clones found in the large DCIS and IDC lesions in KU779 (nos. 7a–e and 11a–e) were branched off from a long common trunk with short branches, which otherwise obscured the past history of these cancer clones (Fig. 2a). However, many normal and BBL samples collected at the same time enabled the analysis of the early history of these cancer clones, although the very recent history was still unclear. In addition, we were able to estimate the approximate timing of early branch points in the tree using the mutation rate measured for normal mammary epithelial cells. In particular, the timing of the acquisition of der(1;16) was more accurately pinpointed than that of other driver events, by maximizing the posterior probability of the observed numbers of duplicated and unduplicated mutations on 1q arm in der(1;16)(+) MRCA (Extended Data Fig. 6a–d and Methods). On an average, der(1;16) in six clones was estimated to be acquired at 10.6 (range 5.8–16.9) years of age (Fig. 2a and Extended Data Fig. 5a–c). We also estimated the average timing at which MRCA emerged as 26.5 (range 18.1–34.4) years of age, assuming a constant mutation rate until the emergence of the MRCA. For example, two distinct der(1;16) detected in a 48-year-old woman (KU779) were estimated to occur in two mammary cells at the ages of 5.8 and 10.0 years, respectively (Fig. 2a). These ancestor cells then gave rise to the MRCAs at the ages of 18.1 and 22.3 years, respectively, from which a number of non-cancer progenies evolved, followed by the appearance of cancer founders at least 10 years after the initial acquisition of der(1;16). In the remaining case (KU582) lacking der(1;16), the MRCA carrying an AKT1 mutation emerged by 4.4 years of age. It was 9.0 years later when the most recent common non-cancer ancestor of cancer clones was confirmed (Extended Data Fig. 5d). Targeted-capture sequencing of additional LCM samples for mutations in the main trunk confirmed that AKT1-mutated lobules derived from the MRCA widely expanded over the 65 mm area, showing a variety of histologies, including proliferative and non-proliferative as well as DCIS lesions (Extended Data Figs. 3c and, 5d). Of particular interest is the observation that multiple cancer founders (nos. 7a–e, 10, 14 and 15 in Fig. 2a, for example) independently evolved from non-cancer ancestors (blue arrows in Fig. 2a and Extended Data Fig. 5a,c,d) (KU779, KU539, KU957 and KU582). This suggests that a cancer population can be initiated at different time points by multiple independent cancer founders originating from common 'non-cancer' ancestors (Extended Data Fig. 6e).

Because these cancer and non-cancer lesions shared many driver alterations, in addition to der(1;16) and an AKT1 mutation, and had an identical germline background, WGS of these lesions provides a unique opportunity to investigate the critical genetic events that discriminate among IDC, DCIS, non-cancer proliferative lesions and histologically normal lobules. In some cases, additional driver alterations were observed only in high grade lesions but not in non-cancer clones in the most recent clade, whereas the opposite was true of other lobules (Fig. 2a and Extended Data Fig. 5). In KU873, for example, a DCIS clone (in no. 3) had two driver mutations in GATA3 and CBFB in addition to der(1;16) and a PTEN mutation, whereas no additional driver mutations were found in the remaining seven non-cancer lesions except for no. 8 carrying a GATA3 mutation (Extended Data Fig. 5b). Conversely, in KU779, a DCIS lobule (no. 14) had no known driver alterations other than der(1;16), whereas lobules in the same clade (nos. 3, 4, 8, 9 and 13) were still proliferative lesions without progression to DCIS or IDC, although they acquired an additional PIK3CA (E726K) with or without a RUNX1 (P267fs) and another PIK3CA (H1047R) mutation, or a PIK3CA

(E545K) mutation (Fig. 2a). In KU539, the DCIS clone in no. 5 had two driver mutations in PIK3CA and CBFB in addition to a GATA3 mutation and der(1;16), although the invasive cancer clone (in no. 4) in the same clade had no additional driver alterations (Extended Data Fig. 5a). Similarly, no additional driver alterations were found in DCIS clones (in nos. 1 and 2) in KU957, whereas proliferative but non-cancer lesions (nos. 4 and 7) carried an additional copy of chromosomes 10 and 15q, respectively, with no. 4 also carrying an IDH1 mutation (Extended Data Fig. 5c). Overall, the most common additional mutations acquired during evolution were those affecting PIK3CA and GATA3 (Extended Data Fig. 5e), which showed no clear correlations with cancer clones, although they are among the most frequent mutations in breast cancer[27,28]. Thus, no consistent patterns of additional driver mutations were observed between cancer and non-cancer clones.

## Characterization of der(1;16)(+) cancers

The unexpected enrichment of der(1;16) in the five index cases suggested that the widespread expansion of satellite lesions of varying histology was a common feature of der(1;16)(+) breast cancer. To confirm this, we screened another set of 33 specimens of Luminal A-like invasive cancer (n = 28) or its putative precursor lesion (ER(+)HER2(−) DCIS) (n = 5) for der(1;16) using FISH and identified an additional eight der(1;16)(+) specimens, two from premenopausal and six from postmenopausal patients (Fig. 3a and Extended Data Figs. 7 and 8). As was the case with der(1;16)(+) clones in the index specimens (Fig. 2a and Extended Data Fig. 5a–c), which were all from premenopausal patients, the two der(1;16)(+) clones in premenopausal patients showed a macroscopic expansion over an area greater than 20 mm in diameter (Fig. 3b and Extended Data Fig. 7), supporting the above-mentioned hypothesis. By contrast, most of the remaining der(1;16)(+) clones from six postmenopausal patients were found in cancer lesions, rarely involving non-cancer lesions and, if ever, the surrounding der(1;16)(+) non-cancer lesions were confined within small lobules less than 10 mm in diameter (Fig. 3a,b and Extended Data Fig. 8a–e). To exclude the possibility that this was due to the late acquisition of der(1;16), we estimated the timing of der(1;16) acquisition in five of the six postmenopausal patients on the basis of phylogenetic analysis. Of interest, the mean age of the acquisition of der(1;16) in the five postmenopausal patients was estimated as 11.7 years (0–18.7), which is comparable to the 10.6 years (5.8–16.9) (P = 0.54) in premenopausal patients (Fig. 3c,d). Thus, we speculate that there should have been a larger expansion of der(1;16)(+) clones, including non-cancer lesions, before menopause, which, however, regressed after menopause in the face of reduced oestrogen levels.

At the onset of puberty, the mammary glands start to proliferate rapidly and undergo a remarkable expansion until the end of adolescence[29,30]. Thus, to see whether the large expansion of der(1;16)(+) clones could be explained only by the physiological enlargement of the growing mammary glands in this period, we evaluated the extent to which der(1;16)(−) non-cancer clones can expand after puberty. For this purpose, we microdissected multiple non-cancer lobules of the surgically resected and fresh-frozen specimens from three newly recruited premenopausal patients with breast cancer (Supplementary Table 1 and Methods). On average, 2.1 (1–6) LCM samples were collected from each of the $10 \times 10$ mm² areas aligned over 40–50 mm lengths from three consecutive tissue slices separated from adjacent slices by 10 mm (Fig. 4a and Extended Data Fig. 9). In total, we collected 77 LCM samples, comprising 66 histologically normal lobules, eight proliferative and three classic-type LCIS lesions, which were subjected to WGS (Methods, Supplementary Note 1 and Supplementary Fig. 3). Six of the eight proliferative lobules harboured one or more breast cancer driver mutations (Fig. 4b,c). Driver mutations were also found in histologically normal lobules, although less common (12 of 66, P = 2.0 × 10⁻³), in which PIK3CA and PIK3R1 were significantly mutated (dN/dS > 1.0) (Fig. 4c and Supplementary Table 10). The presence of driver mutations was

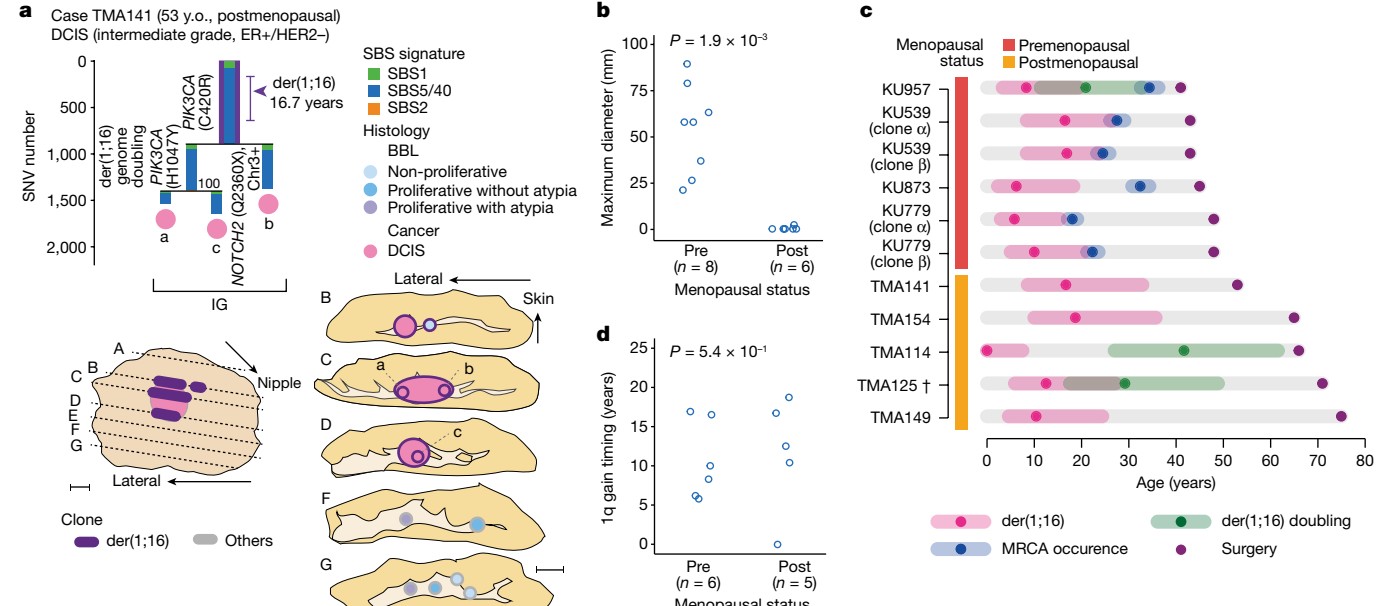

**Fig. 3 | Clonal expansion in patients with der(1;16)(+) breast cancer.**
**a**, Phylogenetic tree (top) and corresponding geographical map (bottom) of clones detected in the surgical specimen of a postmenopausal patient with der(1;16)(+) breast cancer who underwent lumpectomy (TMA141). All the representations follow those in Fig. 2a. Scale bars, 10 mm. **b**, Maximum diameter of the area where der(1;16)(+) non-cancer clones were observed in premenopausal (eight clones in six cases) and postmenopausal (six clones in six cases) patients with der(1;16)(+) breast cancer; *P* values were calculated using the two-sided Mann–Whitney *U*-test. **c**, Estimated timing of der(1;16) acquisition and MRCA emergence (only for premenopausal cases) is shown with the 95% CI. †, the first and second 1q gain timing were estimated because the order of der(1;16) and another 1q gain could not be determined in TMA125. **d**, Estimates of the first 1q gain timing of der(1;16) clones were compared between premenopausal and postmenopausal patients. *P* values from the two-sided Mann–Whitney *U*-test.

associated with a higher clonality as suggested by a significantly larger median VAF of mutations in driver-mutated versus unmutated samples (0.33 versus 0.25, $P = 1.8 \times 10^{-3}$), supporting the role of driver mutations in positive selection (Fig. 4d). One patient had a known pathogenic germline *BRCA2* variant that, however, did not seem to influence the clonality in normal lobules (Extended Data Fig. 9a,d). Among 48 clones that were estimated to have emerged by 1 year of age, 13 (27.1%) involved two or more lobules, of which three clones affected the lobules more than 10 mm apart (Fig. 4e,f). By contrast, only five (9.3%) of 54 clones that emerged after 13 years affected two or more lobules, all of which were still confined within an area less than or equal to 10 mm in diameter. These observations indicate that most of the der(1;16)(−) clones that emerge after puberty stay within a single lobule or, if not, are confined to adjacent lobules and rarely expand to a larger area as observed for those carrying der(1;16). Thus, the large expansion of all der(1;16)(+) clones is not explained only by the physiological development of the breasts during puberty but suggests the driver role of der(1;16). The role of der(1;16) in clonal expansion is further highlighted by the unexpected detection of der(1;16) in three LCIS lesions in KU1215, which was shared with an ILC lesion located in another quadrant and shown to have expanded over the region spanning more than 70 mm in diameter in the subsequent FISH analysis (Extended Data Fig. 9b,c).

## Role of der(1;16) in clonal expansion
Finally, we characterized clinical and pathological features of der(1;16)(+) breast cancers and also investigated its role in breast cancer pathogenesis, using published data of 610 breast cancer cases from The Cancer Genome Atlas (TCGA) (Extended Data Fig. 10). According to the copy-number measurement based on exome sequencing[2], der(1;16) was detected in 119 (19.5%) cases as a concomitant whole-arm 1q gain and 16q loss (Extended Data Fig. 10b). In accordance with previous reports[26,31], most of the der(1;16)(+) cases (86.6%) were classified as Luminal A-type cancers (Extended Data Fig. 10c), in which der(1;16) was

more enriched in the ILC tumours compared to that in the IDC tumours (49.6 versus 12.7%, $P = 3.2 \times 10^{-16}$) (Extended Data Fig. 10e). Whereas Luminal A tumours generally have a favourable prognosis, der(1;16)(+) tumours were associated with a significantly longer overall survival than der(1;16)(−) Luminal A tumours (a median of 33.1 months, compared with 28.3 months, $P = 1.0 \times 10^{-3}$) (Extended Data Fig. 10h). In multivariable analysis, der(1;16) remained to be a significant predictor of prolonged overall survival, even after adjustment for age and stage ($P = 2.4 \times 10^{-3}$, Supplementary Table 16). These pathological subtypes and mutation profiles were largely recapitulated in the der(1;16)(+) tumours that were analysed (Extended Data Fig. 8f). After excluding *PIK3CA*, which was frequently mutated in both der(1;16)(+) and der(1;16)(−) cases, the most frequent mutational targets in der(1;16)(+) Luminal A tumours included *CDH1*, and *GATA3* and *CBFB* in ILC and IDC tumours, respectively. *MAP2K4* and *TP53* mutations were less common compared to those in der(1;16)(−) tumours (Extended Data Fig. 10i,j and Supplementary Table 17). *CDH1* is a putative target of 16q deletion[26,32] and was mutated in 86 (14.1%) of the 610 TCGA cancer cases. Among these, most cases (94%) had biallelic alteration, in which most were associated with der(1;16) (*n* = 46), followed by other 16q loss (*n* = 21) and 16q UPD (*n* = 14). Heterozygous mutation was found in only five cases. Except for one frameshift change, all were variants of unknown significance that were rarely registered in the COSMIC database. These findings indicate that *CDH1* is a bona fide recessive tumour suppressor gene, and haploid loss of *CDH1* alone may not be sufficient for positive selection or clonal expansion during breast cancer development. Regarding this, we analysed the effect of the allelic imbalance caused by der(1;16) on gene expression in 323 Luminal A cancer cases, including 103 der(1;16)(+) cases from the TCGA. The mean expression levels of 1q and 16q genes were substantially increased or decreased in der(1;16)(+) cases compared to those in cases without 1q gain or 16q loss, respectively (Extended Data Fig. 10k). In particular, the mean expression levels of *CDH1* in der(1;16)(+) cases without *CDH1* mutations were almost halved compared with those in cases without 16q loss. However, the effect of haploid gain and loss was not confined

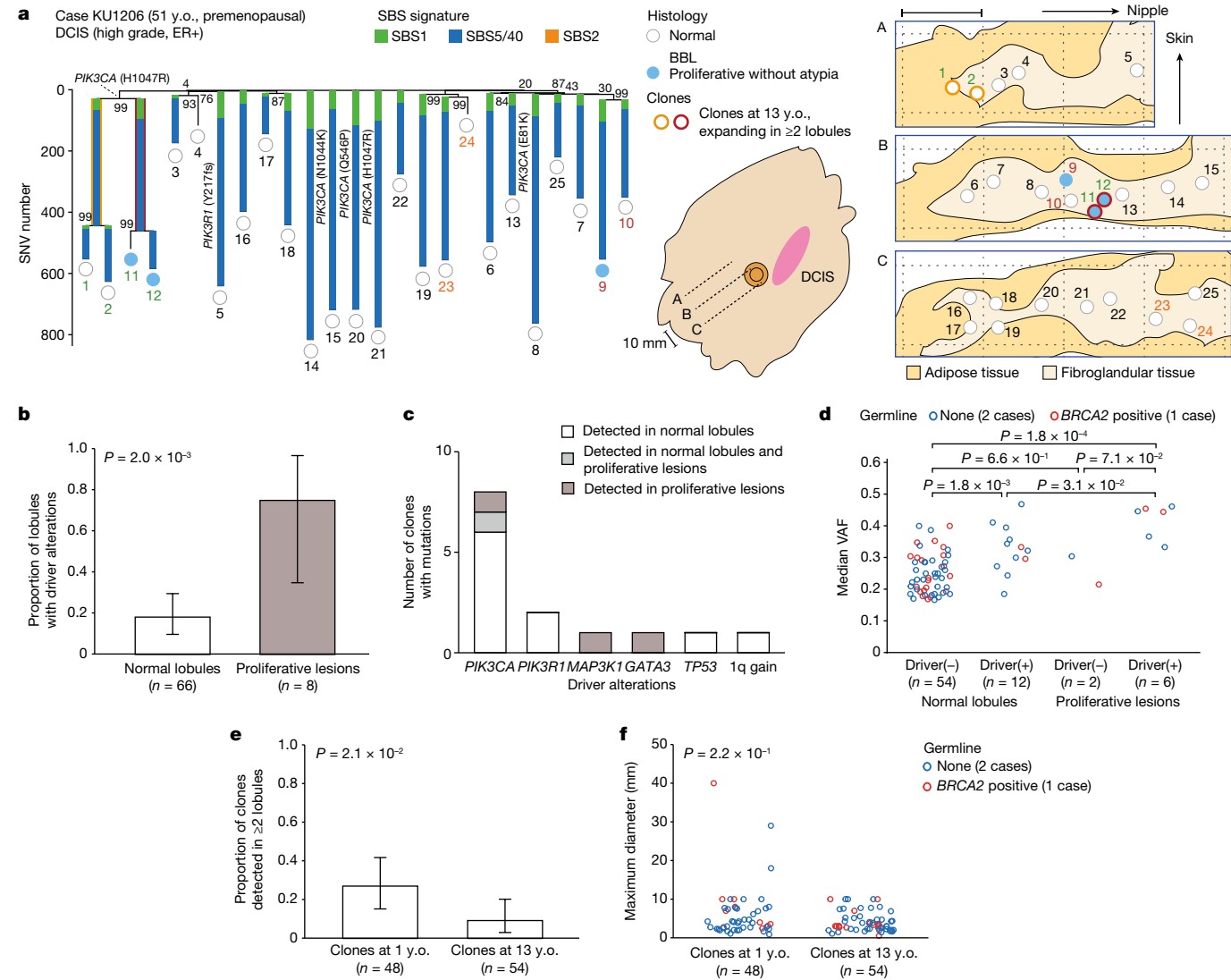

**Fig. 4 | Clonal expansion of non-cancer clones without der(1;16).**
**a**, Phylogenetic tree (left) and corresponding geographical maps of clones detected in non-cancer lobules multi-sampled from the contralateral quadrant of the cancer-containing quadrant in a premenopausal patients with breast cancer without pathogenic germline variants (KU1206) (middle shows an overview of the surgical specimen, right shows split faces of the sliced specimens indicated by black dotted lines in the overview image). SBS signatures, bootstrap values, driver mutations, histological results and numbers are shown as in Fig. 2a. Numbers with the same colour depict samples belonging to the same clones that were present at the age of 1 year. As for clones that were present at the age of 13 years and detected in two or more lobules, corresponding shared branches in the trees and samples in the split faces are highlighted by colours and depicted with colours around circles, respectively. Scale bar, 10 mm.

**b**, Proportion of lobules with driver alterations in histologically normal lobules ($n = 66$) and proliferative lesions ($n = 8$), with $P$ values calculated using two-sided Fisher's exact test. **c**, Number of clones carrying each driver alteration detected in the normal lobules and/or proliferative lesions. **d**, Median VAF in histologically normal lobules and proliferative lesions with and without driver alterations, with $P$ values from the two-sided Mann–Whitney $U$-test. **e,f**, Proportion of der(1;16)(−) non-cancer clones detected in two or more lobules (**e**), and the maximum diameter of the area where each clone was observed (**f**); the clones present at the age of 1 year and 13 years, with $P$ values from two-sided Fisher's exact test (**e**) and the two-sided Mann–Whitney $U$-test (**f**), respectively. Whiskers in **b** and **e** indicate the 95% CI from the binomial distribution. The colours of plots in **d** and **f** depict the status of the germline variants.

to *CDH1* but also seen in other known or putative oncogenes and tumour suppressor genes[27,28], although the effect was highly variable across genes. Thus, the exact molecular targets of der(1;16) are still elusive.

## Discussion

Through phylogenetic analyses, we successfully traced the evolution of breast cancer and precursor lesions, from the acquisition of initial driver alterations to the development of clinically diagnosed disease. The absolute timing and the order of early driver events were more accurately estimated than in previous studies[9–11,33,34] by analysing both cancer and non-cancer lesions and by using the rate of mutation accumulation measured for normal mammary epithelium. As demonstrated in a recent study on myeloproliferative neoplasms[35], the first driver events occurred long before the cancer diagnosis, around puberty or late adolescence or, in one case, as early as in early infancy. However, unlike the case with the myeloproliferative neoplasms study, discrimination between cancer and non-cancer clones along the evolutionary tree was enabled to some time point after the acquisition of initial driver events. In most cases, the appearance of the MRCA of cancer and non-cancer clones was no earlier than the patient's 20s to early 30s. Thus, it seems to still take more than 10 years from the acquisition of the initial driver alterations (at 6–17 years of age) before the initial cancer founders appear. Expanding along the mammary ducts, these clones ultimately

occupied an unexpectedly large area in the breast by the time of cancer diagnosis. Of note, the cancer clones often evolved multifocally from clonally related but still 'non-cancer' ancestors. Being distinct from the classical linear model for the evolution of a single cancer founder, such a branching pattern of evolution of multiple cancer founders from within a non-cancer population might be more common than expected during cancer development (Extended Data Fig. 6e). Another finding of interest is the lack of consistent correlations between histologies and the number and/or type of driver events. Although we cannot exclude the possibility of the presence of undetected driver mutations and structural variations, this may suggest the role of epigenetic changes[36] and/or locally defined microenvironments in cancer development.

It should be noted that such a unique pattern of cancer evolution could be biased by the selection of specimens harbouring multiple satellite BBLs for LCM, which was highly enriched for der(1;16). The analysis of additional cases with der(1;16) confirmed that the presence of persistent non-cancer clones in a large area is an intrinsic feature of der(1;16)(+) breast cancer at least in premenopausal cases. The parallel evolution of multiple independent der(1;16) clones in two cases supports the strong driver role of der(1;16) in puberty or late adolescence. Accounting for 20% of all breast cancers and one- and two-thirds of Luminal A and invasive lobular breast cancers, respectively, der(1;16) defines a major subtype of breast cancers. However, it is still open to question whether or not this pattern of cancer evolution is also common in other breast cancer subtypes. It was observed at least in an *AKT1*-mutated case (KU582). Mutations affecting *PIK3CA* and *PIK3R1* are among the most frequent targets of somatic mutations in breast cancer[27,28] and also common in apparently normal mammary lobules (10 out of 66 lobules) (Fig. 4a and Extended Data Fig. 9a,b). However, none of the clones carrying these mutations showed a widespread expansion. Further investigations are needed to clarify this.

Another finding of interest is the mutational profile of mammary epithelium, which is distinct from that in other tissues[1,37,38]. Throughout a woman's lifespan, the mammary epithelium undergoes dynamic changes that are synchronized with menstrual cycles as well as pregnancy, delivery and breastfeeding thereafter[29,30,39–42], which is also reflected by its unique mutation profile. A significantly reduced mutation rate after menopause might be explained by the reduced cell turnover associated with the cessation of menstrual cycles and/or reduced oestrogen levels. In agreement with this is the negative effect on accumulation of mutations imposed by pregnancy or delivery, during which menstrual cycles are spared. The findings on mutation profile are also in agreement with the epidemiological findings that late menopause and low parity correlate with an elevated risk of breast cancer development[43–45]. However, the reduction of 50 SNVs per parity seems substantially larger than expected from the typical period (1.1–1.5 years) when menstrual cycles are interrupted by pregnancy and breastfeeding (roughly 30 SNVs)[46]. This raises the possibility that after the effacement of markedly proliferated mammary glands after delivery or breastfeeding, the mammary epithelium might be reconstructed by newly recruited, 'dormant' stem cells, in which the SNV burden had been spared. Such dormant stem cells have been proposed to explain the disappearance of clones carrying tobacco signatures in bronchial epithelium after cessation of tobacco smoking[47].

In summary, we revealed mutational processes in the mammary epithelium and the entire life history of breast cancer, highlighting a unique role of der(1;16) in the major subset of Luminal A breast cancer. Our findings may contribute to the understanding of breast carcinogenesis and the development of new strategies for prediction, early diagnosis and even prevention of breast cancer.

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

[1]Department of Pathology and Tumour Biology, Graduate School of Medicine, Kyoto University, Kyoto, Japan. [2]Department of Next-generation Clinical Genomic Medicine, Graduate School of Medicine, Kyoto University, Kyoto, Japan. [3]Department of Breast Surgery, Graduate School of Medicine, Kyoto University, Kyoto, Japan. [4]The Hakubi Center for Advanced Research, Kyoto University, Kyoto, Japan. [5]Department of Gastroenterology and Hepatology, Graduate School of Medicine, Kyoto University, Kyoto, Japan. [6]Division of Cancer Evolution, National Cancer Center Research Institute, Tokyo, Japan. [7]Department of Diagnostic Pathology, Kyoto University Hospital, Kyoto, Japan. [8]Department of Diagnostic Pathology, Osaka Red Cross Hospital, Osaka, Japan. [9]Department of Pathology, Iwate Medical University, Iwate, Japan. [10]Department of Gynecology and Obstetrics, Graduate School of Medicine, Kyoto University, Kyoto, Japan. [11]Department of Obstetrics and Gynecology Faculty of Life Sciences, Kumamoto University, Kumamoto, Japan. [12]Department of Pediatrics, Graduate School of Medicine, Kyoto University, Kyoto, Japan. [13]Adachi Hospital, Kyoto, Japan. [14]Pathology Project for Molecular Targets, Cancer Institute, Japanese Foundation for Cancer Research, Tokyo, Japan. [15]Division of Pathology, Cancer Institute, Japanese Foundation for Cancer Research, Tokyo, Japan. [16]Department of Pathology, Cancer Institute Hospital, Japanese Foundation for Cancer Research, Tokyo, Japan. [17]Division of Hematopoietic Disease Control, The Institute of Medical Science, The University of Tokyo, Tokyo, Japan. [18]Institute for the Advanced Study of Human Biology (WPI-ASHBi), Kyoto University, Kyoto, Japan. [19]Department of Organoid Medicine, Sakaguchi Laboratory, Keio University School of Medicine, Tokyo, Japan. [20]Osaka Research Center for Drug Discovery, Otsuka Pharmaceutical Company, Limited, Osaka, Japan. [21]Breast Surgery Department, Kobe City Medical Center General Hospital, Hyogo, Japan. [22]Division of Genome Analysis Platform Development, National Cancer Center Research Institute, Tokyo, Japan. [23]Department of Integrated Analytics, M&D Data Science Center, Tokyo Medical and Dental University, Tokyo, Japan. [24]Human Genome Center, The Institute of Medical Science, The University of Tokyo, Tokyo, Japan. [25]Department of Medicine, Centre for Haematology and Regenerative Medicine, Karolinska Institute, Stockholm, Sweden. [26]These authors contributed equally: Tomomi Nishimura, Nobuyuki Kakiuchi. [27]These authors jointly supervised this work: Masakazu Toi, Seishi Ogawa. ✉e-mail: sogawa-tky@umin.ac.jp

# Methods

## Data reporting

No statistical methods were used to determine the sample size. The experiments were not randomized. Pathologists were blinded to the genetic alterations in each sample during histopathological evaluation.

## Participants and materials

We enroled 207 female patients with breast cancer who underwent surgery at the Kyoto University Hospital and eight healthy breastfeeding women who delivered at the Kyoto University Hospital or Adachi Hospital. Written informed consent was obtained from all the participants. The study was reviewed and approved by the ethics committees of the Kyoto University and Adachi Hospital, and all the participants consented to publication of data. The characteristics of the participants are summarized in Supplementary Table 1. Invasive cancer lesions in FFPE surgical specimens were immunostained for ER (no dilution), PR (no dilution), HER2 (no dilution) and Ki-67 (1:100 dilution), for which histological grade was also evaluated according to the modified Scarff–Bloom–Richardson grading system[48], to surrogate cancer subtypes. The cut-off for ER and PR-positivity was set at greater than or equal to 1%. For HER2 status, immunohistochemistry scores of 0 and 1+ were considered negative, whereas 3+ was considered positive. Tumours with scores of 2+ were further evaluated by means of dual colour in situ hybridization, wherein an HER2/CEP17 ratio greater than or equal to 2.2 was considered positive. The Ki-67 labelling index was determined in the hotspots. Surrogate subtype classification was defined as follows: Luminal A-like, ER(+) and HER2(−) and histological grade 1 or 2 and Ki-67 less than or equal to 15%; Luminal B-like, ER(+) and HER2(−) and histological grade 3 or Ki-67 greater than 15%; Luminal HER2, ER(+) and HER2(+); HER2-enriched, ER(−) and HER2(+); triple-negative breast cancer (TNBC), ER(−) and HER2(−). Only the nuclear grade was evaluated for DCIS classification according to the *WHO Classification of Tumours of the Breast*[49]; ER, PR, HER2 and Ki-67 were not routinely evaluated. BBLs were also evaluated according to the World Health Organization (WHO) classification and classified as follows: non-proliferative lesions (fibroadenoma without atypia, columnar cell change and apocrine metaplasia), proliferative lesions without atypia (usual ductal hyperplasia, columnar cell hyperplasia, sclerosing adenosis, radial scar and papilloma) and proliferative lesions with atypia (flat epithelial atypia, atypical ductal hyperplasia and atypical lobular hyperplasia). Classic-type LCIS was also classified as BBL per the recent clinical practice wherein LCIS and atypical lobular hyperplasia were grouped as 'lobular neoplasia', which is considered to be a risk factor for cancer development but is not an obligate precursor of invasive cancer[22,50]. When a lesion consisted two or more differentially classified epithelia, the most severe diagnostic category was assigned. Breast cancers were classified by referring to the medical records, whereas BBLs and cancer lesions subjected to genetic analysis were independently reviewed by three experienced pathologists blinded to the genetic alterations. In case a unanimous agreement was not reached for the lesions, the most experienced pathologist reviewed them again and determined the consensus diagnosis.

To establish normal epithelial cell-derived organoids, fresh normal breast tissue and matched blood samples were obtained from 15 patients with breast cancer who underwent total mastectomy, and excess breast milk and oral mucosa were obtained from eight healthy breastfeeding women. For the analysis of cancer-related clonal evolution, 156 patients with cancer who underwent surgery without any preoperative treatment were screened by searching terms related to proliferative lesions in the pathological reports, based on which five archival FFPE surgical specimens and matched blood DNA or FFPE normal lymph nodes were provided by the Kyoto Breast Cancer Research Network (KBCRN) Breast Oncology Research Network (BORN)-BioBank. We also obtained an additional 33 FFPE surgical specimens and five matched FFPE normal lymph node or skin samples from the BORN-Biobank, for which tumour cores in the tissue microarray had already been evaluated for surrogate subtype classification. To investigate the structures of der(1;16)(−) non-cancer clones, fresh-frozen tissues and matched blood samples were obtained from three breast cancer patients who had undergone total mastectomy; FFPE surgical specimens were also used for one of three patients. The sample information is summarized in Supplementary Table 2.

## External datasets

Whole-exome sequencing (WES) data (.bam files) of paired tumour and germline control samples from female patients with invasive breast cancer ($n$ = 661) were downloaded from TCGA data portal (https://portal.gdc.cancer.gov/). The .bam files were converted to the fastq format using biobambam[51] (v.0.0.191) and processed using the Genomon2 pipeline (v.2.6) for mutation calling (below) and our in-house pipeline 'CNACS' for copy-number analysis as described by Yokoyama et al.[2]. RNA-sequencing (RNA-seq) data in the transcripts per million (TPM) format were also downloaded from the TCGA data portal. Clinicopathological information of these samples was also downloaded from TCGA data portal, whereas the information about PAM50 messenger RNA subtypes was extracted from the study by Ciriello et al.[32]; if data were lacking, information was extracted from TCGA Network[27]. Samples included in this study are summarized in Supplementary Table 2.

## Organoid culture

Single normal epithelial cell-derived organoids were established according to the protocols described by Lim et al.[52], Wong et al.[53] and Dekkers et al.[54], with some modifications. First, single-cell suspensions were obtained from normal breast tissues of patients with breast cancer or the breast milk of healthy breastfeeding women. Fresh mammary tissue from the contralateral quadrant of the cancer-containing one was obtained from surgical specimens, which were confirmed to be pathologically normal by three pathologists reviewing the haematoxylin and eosin (HE)-stained fresh-frozen and FFPE sections. The tissue was minced manually and digested for 8–10 h at 37 °C with 150 U ml$^{-1}$ collagenase I (Thermo Fisher Scientific (Thermo)), 50 U ml$^{-1}$ hyaluronidase (Merck) and 100 U ml$^{-1}$ DNase I (Roche) in Dulbecco's modified Eagle's medium (DMEM)/F-12 supplemented with 5% fetal bovine serum (FBS), 0.5 mM glutamine (Thermo), 5 µg ml$^{-1}$ insulin (Merck), 10 ng ml$^{-1}$ epidermal growth factor (EGF) (PeproTech) and 500 ng ml$^{-1}$ hydrocortisone (Merck)[52]. The resulting cell suspension was sequentially digested with 0.25% trypsin and 1 mM EDTA (1 min, 37 °C), and 5 mg ml$^{-1}$ dispase (Thermo; 1 min, 37 °C). A single-cell suspension was obtained by means of filtration through a 40 µm cell strainer (Corning) after removing red blood cells using RBC Lysis Solution (QIAGEN). Excess breast milk was stored at 4 °C and transported to the laboratory within 24 h. Milk was diluted 1:1 with PBS and centrifuged at 1,000g for 10 min (ref. 53). The supernatant, including milk fat, was discarded and the cell pellet was washed 3–5 times with PBS; a single-cell suspension was then obtained by means of filtration through a 40 µm cell strainer.

Next, mammary epithelial cells were isolated from the single-cell suspension using CD326 EpCAM MicroBeads (Miltenyi Biotec (Miltenyi)) (1:5 dilution) and the MACS Cell Separation System (Miltenyi) according to the manufacturer's instructions. For the cell suspension obtained from the breast milk of healthy participants without available paired oral mucosa, the negative selection was performed using CD45 MicroBeads (Miltenyi) (1:5 dilution) before EpCAM-positive epithelial cell isolation to use CD45-positive leukocytes as paired control samples for WGS.

The isolated epithelial cells were cultured using two different methods. In culture method 1, the cells were resuspended in Matrigel (Corning) at a density of 5,000–20,000 cells per ml for tissue-derived cells and 20,000–80,000 cells per ml for milk-derived cells; 100 µl suspension was seeded into each well of six-well plates, which was filled

with 2 ml DMEM/F-12 supplemented with 1× B27 supplement (Thermo), 0.5 mM glutamine, 5 µg ml⁻¹ insulin, 10 ng ml⁻¹ EGF, 500 ng ml⁻¹ hydrocortisone, 20 ng ml⁻¹ cholera toxin (Merck), 100 U ml⁻¹ penicillin and 100 µg ml⁻¹ streptomycin[52]. The cells were cultured for 13–37 days (median 17 days) until the organoids grew to equal to or more than 80 µm in diameter and were collected. In culture method 2, the epithelial cells that had been stocked frozen in CELLBANKER 1 (Takara Bio (Takara)) were suspended in 25 µl Matrigel and plated on 48-well plates, wherein each well was filled with the culture medium based on the previous protocols[54] with modifications: 250 µl DMEM/F-12 supplemented with 10 mM HEPES, 1× B27 supplement, 2 mM glutamine, 10 nM Gastrin I (Merck), 1 mM N-acetylcysteine (FUJIFILM Wako Pure Chemical (FUJIFILM)), 1 µg ml⁻¹ R-Spondin 1 (R&D Systems), 5 nM Neuregulin 1 (PeproTech), 20 ng ml⁻¹ FGF-10 (PeproTech), 5 ng ml⁻¹ EGF, 100 ng ml⁻¹ Noggin (PeproTech), 500 nM A83-01 (Tocris Bioscience), 5 µM Y-27632 (FUJIFILM), 0.5 µg ml⁻¹ hydrocortisone (Selleck Chemicals), 100 nM β-Oestradiol (Cayman Chemical Company), 10 µM Forskolin (Merck), 100 U ml⁻¹ penicillin, 100 µg ml⁻¹ streptomycin and 10% Afamin-Wnt-3A serum-free conditioned medium[55]. After 15–82 days (median 20 days) of primary culture, single-cell-derived organoids were established by seeding dissociated cells at a limiting dilution, treated with TrypLE Express (Thermo) and then cultured for 13–88 days (median 15 days). In total, we established 71 single-cell-derived organoids from 15 patients with breast cancer and eight healthy participants.

## LCM

FFPE surgical specimens from patients with breast cancer were prepared for LCM using the protocol described by Uehiro et al.[56]. Individual 10 µm-thick FFPE specimens were placed on PEN membrane (4 µm)-coated glass slides (Leica Microsystems (Leica)) and immunohistochemically stained with pan-cytokeratin antibody cocktails (AE1/AE3) no. 412811 (NICHIREI) (no dilution). HistoZyme (Diagnostic BioSystems) was used for antigen retrieval, and the VECTOR Red Alkaline Phosphatase Substrate Kit (VECTOR Laboratories) was used for visualization. For the analysis of der(1;16)(−) non-cancer lobules, surgical specimens of premenopausal patients with breast cancer who underwent total mastectomy were sliced into 10-mm-wide slices and 10 × 10 mm² tissues were consecutively obtained from the slices of the contralateral quadrant of the cancer-containing one and immediately frozen. Frozen tissues were sectioned using a cryostat CM1950 (Leica). Ten micrometre-thick sections were placed on PEN membrane-coated slides and stained with Mayer's Hematoxylin solution (FUJIFILM) and Eosin Y (FUJIFILM). LCM of the stained FFPE and frozen tissue slides was performed using the LMD7000 or LMD7 system (Leica). The pathologists reviewed the HE-stained slides, and CK5- and E-cadherin-stained slides (1:100 and 1:50 dilutions, respectively) if needed, for each of the 10–15 sequentially sectioned 10-µm-thick LCM slides and diagnosed each dissected lesion.

## WGS

The DNA extracted from each single organoid established in culture method 1 was divided into two aliquots, each of which was independently subjected to whole-genome amplification (WGA) with the REPLI-g Single Cell Kit (QIAGEN) and analysed by means of WGS and subsequent validation sequencing[3]. DNA from the fresh organoids successfully expanded in culture method 2, peripheral blood, oral mucosa and leukocytes derived from breast milk was extracted using the QuickGene DNA whole blood kit (Kurabo Industries), Gentra Puregene Kit (QIAGEN), QIAamp DNA Blood Mini Kit (QIAGEN) or QIAamp DNA Micro Kit (QIAGEN). DNA from FFPE and fresh-frozen LCM samples was extracted using the GeneRead DNA FFPE Kit (QIAGEN) and Maxwell 16 Cell LEV DNA Purification Kit (Promega), respectively.

WGS libraries were prepared as follows: 100 ng of the WGA DNA extracted from single organoids in culture method 1 was used to prepare a library using the TruSeq Nano DNA Library Prep Kit (Illumina)

or Lotus DNA Library Prep Kit (Integrated DNA Technologies (IDT)); 5–50 ng of DNA extracted from fresh organoids in culture method 2 was used to prepare a library using the xGen Prism DNA Library Prep Kit (IDT); 10–200 ng of DNA extracted from FFPE LCM samples was used to prepare a library using the SMARTer ThruPLEX DNA-seq Kit (Takara) or xGen Prism DNA Library Prep Kit; 2.5–30 ng of DNA extracted from fresh-frozen LCM samples was used to prepare a library using the Lotus DNA Library Prep Kit. These libraries were sequenced on a NovaSeq 6000 system (Illumina) or DNBSEQ-G400RS (MGI Tech) in 100–150-basepair (bp) paired-end mode, according to the manufacturer's instructions. In total, paired WGA samples from 65 organoids, six fresh organoid samples, 84 FFPE LCM samples, 79 fresh-frozen LCM samples and 36 matched germline controls were used. The target coverage was 35× for FFPE samples and 30× for other types of sample, and the actual average coverage was as follows: 35× (19–59×) in WGA organoid samples, 45× (40–61×) in fresh organoid samples, 46× (24–100×) in FFPE LCM samples, 34× (18–63×) in frozen LCM samples and 41× (28–73×) in germline samples.

Raw sequence data were processed into .bam files using Genomon2, as previously described[2]. In brief, sequencing reads were aligned to the human reference genome (GRCh37) using the Burrows–Wheeler Aligner[57] (v.0.7.8) with default parameter settings. The PCR duplicates were eliminated using biobambam. For organoid samples, mouse-derived sequencing reads resulting from Matrigel contamination were removed using Xenome[58] (v.1.0.0), and only the reads classified as 'human-mapped' were processed into .bam files. Mutation calling for WGA organoid samples was performed after merging each pair of two .bam files of WGA samples derived from a single organoid using Samtools[59] (v.1.10); each mutation call was then reviewed back to the original two .bam files using GenomonMutationFilter[2] (v.0.2.1); mutations detected in both the WGA samples with two or more variant reads each were considered true somatic mutations, whereas mutations detected in only one sample were eliminated as WGA-related errors. The mutations listed in our in-house mouse-derived variant list were eliminated as artefacts due to Matrigel contamination.

In the entire WGS analysis, mutation calling was performed by means of paired analysis using a 'three-caller combination' to improve the sensitivity and true positive rate, wherein mutations were called by three different callers (Genomon2, Mutect2 (ref. 60) (GATK4, ref. 61 (v.4.1.2)), and Strelka2 (ref. 62) (v.2.9.3)) independently; the mutations detected by two or three callers were considered 'high-confidence' mutations. As described previously[2], Genomon2 first discards the low-quality, unreliable reads and variants with mapping quality of less than 20 and/or base call quality of 15 or lower. Next, the variants that did not meet the following criteria were further excluded as sequencing errors: (1) a sufficient depth (six or more) in both samples and the matched controls; (2) VAFs greater than 0.1, 0.15 and 0.12 in WGA and/or fresh organoid, FFPE LCM and fresh-frozen LCM samples, respectively; (3) variant reads three, five and four or more in WGA and/or fresh organoid, FFPE LCM and fresh-frozen LCM samples, respectively; (4) VAFs less than 0.07 and variant reads one or less in matched controls; (5) a strand ratio not equal to 0 or 1; (6) Fisher's $P < 10^{-1.5}$ for WGA and/or fresh organoid samples and $<10^{-1.3}$ for FFPE and fresh-frozen LCM samples; (7) EBCall[63] $P < 10^{-5}$, $<10^{-4}$, $<10^{-5}$ and $<10^{-3}$ for WGA organoid, fresh organoid, FFPE LCM and fresh-frozen LCM samples, respectively, which were evaluated with a 'control panel' consisting of WGS data of 39, 20, 42 and 14 blood or normal tissue samples of unrelated participants prepared using the corresponding library preparation kits; for the analysis of WGA organoid and FFPE LCM samples, which were expected to contain a lot of sample type-specific artefacts, a 'control panel' consisting of an increasing number of normal samples of corresponding sample type was used to eliminate artefacts. The .bam files were further edited by means of Samtools to remove sequencing reads with mapping quality below 20 and duplicated reads and then analysed using Mutect2 and Strelka2. Mutation calling using Mutect2 was performed as follows:

initially, variants were called by Mutect2 using panel of normals made from 'control panel' data, to filter out sequencing noise; the raw output of Mutect2 was subsequently processed by means of FilterMutectCalls and FilterByOrientationBias in the default settings to filter out the remaining sequencing errors. We excluded the variants that were not supported by (1) a sufficient read depth (eight or more) in both samples and the matched controls; (2) VAFs greater than 0.1, 0.15 and 0.12 in WGA and/or fresh organoid, FFPE LCM, and fresh-frozen LCM samples, respectively; (3) allelic depths for the alternative alleles four or more in WGA and/or fresh organoid and fresh-frozen LCM samples and five or more in FFPE LCM samples; (4) VAFs less than 0.07 and allelic depths for the alternative alleles two or less in matched controls; (5) Phred-scaled quality for the possibility of sequencing errors (SEQQ) greater than 40 (for SNVs); (6) Phred-scaled quality of strand bias artefact (STRANDQ) greater than 70 for WGA and/or fresh organoid samples and greater than 50 for FFPE and fresh-frozen LCM samples (for SNVs). Mutation calling using Strelka2 was performed in the default settings, and the variants that were not supported by (1) a sufficient read depth (ten or more) in both samples and the matched controls; (2) VAFs greater than 0.1, 0.15 and 0.12 in WGA and/or fresh organoid, FFPE LCM and fresh-frozen LCM samples, respectively; (3) the alternative alleles five or more in WGA and/or fresh organoid and fresh-frozen LCM samples and six or more in FFPE LCM samples; (4) VAFs less than 0.07 and the alternative alleles two or less in matched controls; (5) a somatic Empirical Variant Score (SomaticEVS) greater than 17 for SNVs, greater than 16 for indels in WGA organoid samples and greater than 6 for indels in fresh organoid, FFPE LCM and fresh-frozen LCM samples, were excluded. The variants identified by each caller were annotated using ANNOVAR[64]; the variants that were listed in the 1000 Genomes Project dataset or gnomAD database with a minor allele frequency of more than or equal to 0.001 and variants within segmental duplications reported in the GenomicSuperDups database or repetitive sequences reported in the University of California, Santa Cruz (UCSC) Genome Browser[65] were further excluded, except for the driver mutations defined below, to achieve a high true positive rate. After that, each variant was reviewed in .bam files of each sample and the matched control as well as 'control panel' samples using GenomonMutationFilter: variants that were not supported by (1) VAFs greater than 0.1, 0.15 and 0.12 in WGA and/or fresh organoid, FFPE LCM and fresh-frozen LCM samples, respectively; (2) the variant reads of three, four and five or more in fresh organoid samples, WGA organoid and fresh-frozen LCM samples and FFPE LCM samples, respectively; (3) variant read one or less in matched controls; (4) average VAFs less than or equal to 0.05 and total variant read of ten or less in 'control panel' samples; (5) VAFs in samples 15 times or more higher than average VAFs in 'control panel' samples, were further excluded as sequencing artefacts. Variants clustered within a short length (150 bp) were subjected to visual inspection using the Integrative Genomics Viewer[66] to further eliminate sequencing errors. Somatic driver mutations were defined as loss-of-function mutations in tumour suppressor genes or mutations reported in the COSMIC database with ten or more mutated tumours in all types of cancer or five or more mutated tumours in breast cancers.

For the evaluation of germline variants, germline samples were analysed using Genomon2 in the unpaired mode. The variants in breast cancer susceptibility genes that were registered as 'pathogenic' or 'likely-pathogenic' in the ClinVar database were considered pathogenic variants.

## Copy-number analysis
CNAs were analysed using Control-FREEC[67] (v.11.0) with the contaminationAdjustment option, except for the WGA organoid samples in which detection of small copy-number changes was difficult due to WGA-related artefacts. The copy-number gains and losses, and the uniparental disomies were visually confirmed using normalized copy numbers and beta allele frequency plots. CNA results are summarized in Supplementary Table 3.

## Estimation of mutation rate in normal epithelial cells
The mutation accumulation rate was estimated using clonal mutations detected in each single-cell-derived organoid, which was thought to have been inherited from the original single cell. We excluded six organoids in which polyclonal origins were suspected (median VAF less than 0.4); furthermore, we excluded an organoid from a participant carrying a *CDH1* germline pathogenic variant from the analysis. Clonal mutations were defined by the following method: (1) the Gaussian mixture model was adapted to VAF distribution of SNVs in each organoid ($n$ = 64) using the R package mclust[68] (v.5.4.7) to determine the optimal number of mixture components, one or two, by which the organoids consisting of two components with VAF peaks greater than or equal to 0.4 and less than 0.25 were classified as 'bimodal', and those consisting of one or two components with VAF peaks greater than or equal to 0.4 were classified as 'unimodal' and (2) all the mutations in 'unimodal' organoids ($n$ = 49) were considered clonal mutations; by contrast, mutations with VAF values lower than the intersection point of two components in 'bimodal' organoids ($n$ = 15) were eliminated as subclonal mutations acquired during cell culture.

A linear regression model without intercept was fitted to estimate the effects of age and known breast cancer risk factors on the number of clonal SNVs and indels in 61 organoids, wherein the reciprocal of the number of analysed organoids per case was weighted for each organoid. Initially, we tested the effects of years before menopause (age at sample collection in premenopausal women and age at menopause in postmenopausal women), years after menopause (zero in premenopausal women and the difference between age at sample collection and age at menopause in postmenopausal women), the presence of driver mutation, presence of breast cancer, known breast cancer risk factors (body mass index, parity, alcohol consumption per day, smoking pack-years and number of first- and second-degree relatives with breast cancers) and the factors that might affect the sensitivity of mutation call (sample type (WGA or fresh organoids) and the fraction of genomic region with more than or equal to 15× coverage) on the number of mutations simultaneously. Then, we removed the least significant variable one by one by comparing models with and without each variable until the removal of any variable led to a significant difference ($P$ < 0.05) by analysis of variance. Finally, we defined the final model with the least significant coefficients. Years before and after menopause were both extracted as significant coefficients on mutation number in the final models for SNVs and indels, which showed significant improvement compared to the models that did not incorporate the age at menopause (by analysis of variance). All statistical analyses are summarized in Supplementary Table 4.

## Phylogenetic analysis
In the multi-sampling analysis, phylogenetic trees were reconstructed using somatic mutation data, copy-number information and mutant cell fraction (MCF) across all LCM samples in each subject. First, the Gaussian mixture model was adapted to VAF distribution of mutations within diploid regions to decompose major and minor clones. Then, MCF was calculated by doubling the mean value of the largest Gaussian distribution.

Before reconstructing a phylogenetic tree using MEGA[69] (v.11.0.11), we made an input matrix comprising all mutations detected across all samples, combined with their mutation status for all samples, which was determined according to the depth, number of supportive reads and copy-number status, as follows: (1) for samples with two or more supportive reads, the mutation status was assigned as 'mutation_present'; (2) for samples with only one supportive read, the mutation status was assigned as 'mutation_unknown'; (3) for samples with no supportive reads, (a) when chromosomal loss or other loss of heterozygosity was present at the mutation locus, the mutation status was assigned as 'mutation_unknown'; (b) when no supportive read for the mutation

was well expected ($P > 0.05$) according to the binomial distribution determined by sequencing depth, total copy number (TCN) and MCF, the mutation status was also assigned as 'mutation_unknown' and (c) otherwise, the mutation status was assigned as 'mutation_absent'. On the basis of these criteria, we confirmed high accuracy for mutation status assignment as 'mutation_present' or 'mutation_absent' by validation sequencing (below) for 780 randomly selected mutations in two cases, wherein mutation status in two and nine samples, respectively, was evaluated for each mutation (accuracy 99.4%, 3,055 out of 3,072 mutation statuses); the results of validation sequencing are summarized in Supplementary Table 5. Then, a maximum parsimony tree was established using MEGA with 1,000 bootstrap replicates.

Subsequently, we used the R package treemut[35] (v.1.1) to assign each mutation to a branch using the expectation maximization method based on the number of supportive reads and the sequencing depth for all potential mutations for all samples, as well as the tree information from MEGA (branching pattern and branch length). However, because treemut was originally developed for the analysis of monoclonal diploid samples and assumes that VAF = 0.5 for mutated loci and VAF = 0 for wild-type loci, we corrected variant read counts on the basis of adjusted VAF (aVAF), MCF, TCN and minor copy number (MCN) for bulk LCM samples, as if they consisted of a clonal population derived from a single cell. At first, mutant allele number (MAN) in a single mutant cell was calculated assuming an MCF of 100%, using the following formula:

$$MAN = VAF \times (MCF \times TCN + (1 - MCF) \times 2)/MCF.$$

Next, adjusted TCN (aTCN) was determined as follows:
if $MAN \leq 0.5$, aTCN = 2,
if $MAN > (TCN - MCN)$, aTCN = $(TCN - MCN) \times 2$,
otherwise, aTCN = (the nearest whole number from MAN) × 2.
Then, adjusted VAF (aVAF) was calculated by dividing MAN by aTCN.
Finally, corrected variant read counts were calculated by multiplying depth by aVAF.

The resulting phylogenetic trees were drawn with the SNV number to infer the time course of clonal evolution by adapting the SNV accumulation rate estimated in single-cell-derived organoids. Cancer driver mutations and CNAs were annotated manually in the trees as follows: in the cases in which more than one sample in a clade carried the same hotspot mutations with VAFs greater than 0.15 and variant reads of five or more for FFPE LCM samples, or VAFs greater than 0.12 and variant reads of four or more for fresh-frozen LCM samples, the mutation was assigned to the shared branch if it was accompanied by other shared mutations in that clade or assigned to the private branches if it was the only mutation shared by the samples in that clade; in the cases in which more than one sample in a clade shared CNAs, losses or gains of the same paternal or maternal alleles as determined by single-nucleotide polymorphism analysis were considered the same events and assigned to the shared branch, whereas those of the different paternal or maternal alleles were considered different events and assigned to the private branches. Information about whether each CNA was assigned to a shared or private branch is summarized in Supplementary Table 6.

The method of phylogenetic analysis using MEGA and treemut was validated by reconstructing the representative two trees (KU539 and KU779) using PyClone-VI (ref. 70) (v.0.1.0), wherein no major inconsistency was observed between the two methods in terms of the topology and assignment of driver mutations to corresponding branches (Supplementary Note 2).

## Analysis of mutational signatures

Mutational signatures were extracted using the R package MutationalPatterns[71] (v.3.4.0) in strict signature refitting mode. In the multi-sampling analysis, signature extraction was performed using SNVs on branches with 100 or more SNVs in the phylogenetic trees, wherein each branch was treated as an individual sample. Initially,

SNVs were allocated to all known COSMIC SBS signatures (v.3.1). The overall contribution of each COSMIC SBS signature was calculated in organoid, FFPE LCM and fresh-frozen LCM samples. Then, SBS signatures with a 5% or more contribution in one or more sample type were selected to prevent overfitting. Finally, SNVs were re-allocated to five SBS signatures (SBS1, SBS2, SBS5, SBS13 and SBS40). Because two clock-like 'flat' signatures, SBS5 and SBS40, are difficult to separate[24,25], they were collectively designated as 'SBS5/40' in the subsequent analyses. To validate the signatures extracted using MutationalPatterns, we also analysed SBS signatures using two more algorithms, the SigProfiler Bioinformatic Tools MatrixGenerator[72] (v.1.1.27) and Extractor[73] (v.1.1.1), and the R package HDP[74] (v.0.1.5). In the SigProfiler analysis, de novo signatures were extracted using 1,000 nonnegative matrix factorization iterations, followed by further decomposition to COSMIC SBS signatures (SBS1, SBS2, SBS5, SBS7a, SBS7b, SBS13 and SBS18). In the HDP analysis, SBS signatures were analysed using COSMIC SBS signatures as priors, wherein six prior SBS signatures (SBS1, SBS2, SBS13, SBS16, SBS18 and SBS45) and two new signatures were extracted. The new signatures were further deconvoluted to COSMIC SBS signatures using the R package deconstructSigs[75] (v.1.8.0), resulting in SBS1, SBS5 and SBS7a. Because SBS7a, SBS7b, SBS16, SBS18 and SBS45 were not extracted by MutationalPatterns and were inconsistent between Sig-Profiler and HDP in many parts, we considered these signatures with less confidence (Extended Data Fig. 4a). SBS1, SBS2, SBS5/40 and SBS13 were extracted by means of all three extraction algorithms, and the contributions in each sample showed a similar pattern (Extended Data Fig. 4b).

## Estimation of the timing of MRCA emergence and der(1;16) acquisition

The MRCA in the clonally related samples was identified on the basis of the phylogenetic tree. For common MRCAs of cancer and non-cancer clones (Fig. 2 and Extended Data Fig. 5), we assumed that the mutation accumulation rate was constant until the timing of emergence of the MRCA ($T_{MRCA}$) and equal to that of normal cells. A point estimate of $T_{MRCA}$ was calculated by dividing the number of SNVs in MRCA ($N_{MRCA}$) by the SNV accumulation rate before menopause in normal cells ($R_0$), and the 95% CI (confidence interval) was calculated using the $R_0$ values randomly sampled 1,000 times on the basis of a normal distribution estimated from the linear regression model in the single-cell-derived organoids.

For the timing of der(1;16) acquisition, that is, the timing of 1q gain ($T_{1q\_gain}$), we first obtained the number of duplicated mutations in MRCA using the MAN for each SNV on 1q, which was calculated as described above; if the average number of MAN of the related samples was 1.5 or less, such a mutation was considered 'unduplicated' or otherwise considered 'duplicated'; then, the number of 'unduplicated' and 'duplicated' SNVs in MRCA was counted as N1 and N2, respectively. $T_{1q\_gain}$ was estimated by maximizing the posterior probability of the observed number of N2 based on the simulation method wherein the value of $T_{1q\_gain}$ was increased by 0.1 years from 0 to $T_{MRCA}$ or age at sampling and N2 was simulated 1,000,000 times using the $R_0$ value that was randomly sampled 1,000 times as in $T_{MRCA}$ estimation, assuming that mutations occurred on each 1q allele at a constant rate ($R_{0\_1q}$ = $R_0$ × (the size of 1q gained)/(the size of the genome) × 0.5 (for haploid)) according to the Poisson distribution. For the clones with two 1q gains (KU957, TMA114 and TMA125 (Extended Data Figs. 5c and 8a,b)), the timing of der(1;16) genome doubling ($T_{gd}$) was also simulated, wherein $T_{1q\_gain}$ and $T_{gd}$ were increased by 0.1 years from 0 to $T_{gd}$, and from $T_{1q\_gain}$ to $T_{MRCA}$ or age at sampling, respectively. The SNVs with the average MAN of more than 2.5 were considered 'triplicated' (N3). The values of $T_{1q\_gain}$ and $T_{gd}$ with the maximum probability for the observed N2 and N3 were considered to be point estimates of $T_{1q\_gain}$ and $T_{gd}$, respectively. The details of the simulation are shown in Extended Data Fig. 6a–d, and the results are summarized in Supplementary Table 7.

## FISH analysis

Unstained, 4 µm-thick FFPE sections were subjected to hybridization with bacterial artificial chromosome clone-derived probes for 1p.31.3, 1q23.3 and 16q23.2 to detect all types of der(1;16)(+) clone with concurrent whole-arm 1q gain and 16q loss, wherein the 1p.31.3 signals were used as controls. For KU779, in which two independent der(1;16)(+) MRCAs were identified with WGS, FFPE sections were also subjected to hybridization with Vysis DNA probes for CEP6 (D6Z1) and CEP16 (D16Z3) (Abbott Laboratories) to distinguish the two clones by D16Z3 signal counts, wherein the D6Z1 signals were used as controls (Fig. 2a,d). The probes used are listed in Supplementary Table 8. The hybridized slides were stained with 4,6-diamidino-2-phenylindole and examined under a BZ-X800 fluorescence microscope (KEYENCE). The FISH-probed signals were counted in 100 or more nuclei or in three or more microscopic fields per lesion.

## Measurement of the clonal expansion areas

In the der(1;16)(+) breast cancer cases, lesions originating from der(1;16)(+) MRCAs were identified by performing FISH on FFPE surgical specimens. For five of 141 der(1;16)(+) lesions identified by means of FISH, we confirmed that they actually carried both der(1;16) and more than 90% of shared mutations in the MRCAs by means of targeted-capture sequencing using Next-Generation Sequencing (NGS) Discovery Pools (IDT) and the xGen CNV Backbone Hyb Panel (IDT) (below) (Supplementary Table 9). In the case KU539 with two independent der(1;16)(+) MRCAs (Extended Data Fig. 5a), two clones could not be distinguished using FISH, unlike that in the case KU779 (Fig. 2a,d), but could only be distinguished by evaluating the shared mutations in the MRCAs by means of sequencing, wherein the der(1;16)(+) lesions evaluated by means of FISH were considered as carrying 'undetermined' type of der(1;16) and were eliminated from the target lesions for measurement of the clonal expansion area. In the *AKT1*-mutated breast cancer case (Extended Data Fig. 5d), lesions originating from an *AKT1*-mutation(+) MRCA were identified by means of targeted-capture sequencing (below), wherein lesions carrying both *AKT1* mutation and more than 50% of shared mutations in the MRCAs were defined as originating from the MRCA (Supplementary Table 9). In these cases, the distance between the most distant non-cancer lesions originating from the same MRCA was measured as the maximum diameter of the area over which the non-cancer clones had expanded. In der(1;16)(+) cancer cases that were not accompanied by the formation of any der(1;16)(+) non-cancer lesions, the diameter was considered to be zero. For the fresh-frozen LCM samples, we first reconstructed phylogenetic trees as described above. Next, the cell fraction of mutations allocated to private branches was estimated using PyClone-VI. The mutations in shared branches and those in private branches whose cellular prevalence was high enough to determine clonal structure on the basis of the Pigeonhole principle, were used to detect clones that existed at the age of 1 or 13 years: only when the sum of their cellular prevalence exceeded 1.0, the two mutations were thought to be in the same structures. These clones unrelated to cancer were identified only through WGS, which might lead to lower sensitivity in the detection of clonally related lesions because of the limited number of observations compared to those obtained through FISH analysis. We measured the distance between clonally related lesions and the nearest clonally unrelated lesions in these cases as the maximum diameter of the area of clonal expansion to prevent underestimation.

## Analysis of significantly mutated genes

Significantly mutated genes in the pathologically normal lobules, non-cancer proliferative lesions and cancer lesions in multi-sampled cases were analysed based on d$N$/d$S$ using the R package dndscv[76] (v.0.0.1.0) to investigate the difference in mutational processes during the evolution of cancer. Mutations detected in the shared branches were counted once in each group of lesions. Genes with d$N$/d$S$ > 1 and $q$ < 0.1 were considered to be significantly mutated (Supplementary Table 10).

## Targeted-capture sequencing of breast cancer-associated genes

For the ten breast cancer cases in which tumour cores in the tissue microarray were defined as der(1;16)(+) by FISH analysis, tumour lesions were macro-dissected from FFPE surgical specimens and analysed by means of targeted-capture sequencing using the xGen Predesigned Gene Capture Pools (IDT) designed for 189 genes associated with breast cancer (Supplementary Table 11) and the xGen CNV Backbone Hyb Panel. Driver mutations were called using Genomon2 as described above: the variants that were not supported by (1) a sufficient depth (eight or more); (2) VAFs greater than 0.02 and variant reads of five or more and (3) EBCall $P < 10^{-4}$ were excluded as sequencing artefacts and the variants that met the criteria of driver mutations used in WGS analysis were identified. In these cases, no pathogenic variants of breast cancer susceptibility genes were identified. Copy-number changes were evaluated on the basis of the sequencing data using CNACS as described above. Eight out of nine tumours successfully evaluated for copy-number changes were confirmed to carry der(1;16), detected as concurrent whole-arm 1q gain and 16q loss. The results are summarized in Supplementary Table 12.

## WES and transcriptomic analysis of TCGA cohort

Germline variants and somatic mutations in TCGA breast cancer WES cohort were evaluated using Genomon2, as described for the WGS samples. Somatic mutations were evaluated for 610 sporadic cases as follows: the variants that were not supported by (1) a sufficient depth (eight or more) in both samples and matched controls; (2) VAFs greater than 0.05 and variant reads of four or more in samples; (3) VAFs below 0.05 and variant reads of two or less in matched controls; (4) a strand ratio not equal to 0 or 1; (5) Fisher's $P < 10^{-1.3}$ and (6) EBCall $P < 10^{-4}$ were excluded as sequencing artefacts; the variants within repetitive sequences were further excluded; the variants clustered within a short length (150 bp) were subjected to visual inspection on the Integrative Genomics Viewer to further eliminate sequencing errors. Copy-number changes were evaluated using CNACS, as described above. Tumours with concurrent flat whole-arm 1q gain and 16q loss were considered der(1;16) positive (Extended Data Fig. 10b). If there were any apparent breakpoints in the middle of the 1q or 16q arm, the tumours were considered der(1;16) negative.

To investigate the effect of allelic imbalance of der(1;16), that is, trisomy 1q and monosomy 16q, on gene expression, TPM values of genes on 1q or 16q in der(1;16) positive tumours were compared with those in tumours without +1q or −16q, respectively. Tumours with mutations in *CDH1*, *CBFB* and *CTCF* were excluded from the analysis of genes on 16q.

## Estimation of sensitivity and evaluation for true positive rate in WGS mutation calling

The sensitivity of WGS mutation calling was estimated by calculating the fraction of unique heterozygous germline polymorphisms in a sample detected by means of paired analysis using another participant's germline sample to mimic a matched control (Supplementary Note 1). The estimated sensitivity for each sample type is summarized in Supplementary Table 13.

The true positive rate of WGS mutation calling was evaluated using targeted-capture sequencing for randomly selected mutations in WGA organoid and fresh-frozen LCM samples, and randomly selected private mutations and all of the shared mutations assigned to the MRCAs in five cases for FFPE LCM samples. Libraries were prepared using the xGen Prism DNA Library Prep Kit for WGA organoid and FFPE LCM samples; extra WGS libraries were reused for fresh-frozen LCM samples because of the small amount of DNA extracted, followed by target capture using NGS Discovery Pools or xGen Custom Hyb Panel-Accel (IDT), and then sequenced on a DNBSEQ-G400RS. The sequence data were processed

into .bam files through Genomon2, as described above, and each mutation was reviewed using GenomonMutationFilter, wherein mutations with sequencing depths of 100 or more in both test and germline control samples were evaluated. A mutation was considered to be validated when (1) the VAF in the test sample was five or more times higher than that in the corresponding germline control sample; (2) the VAFs in the test sample greater than or equal to 0.05 and (3) for WGA organoid samples only, (1) and (2) was achieved in both of the two paired WGA samples. The true positive rates were 96.6% for organoid WGA samples (134 out of 139 SNVs and 9 out of 9 indels), 97.3% for private mutations in FFPE LCM samples (209 out of 215 SNVs and 7 out of 7 indels) and 99.7% for fresh-frozen LCM samples (308 out of 309 SNVs and 23 out of 23 indels). As for the validation of the MRCA mutations in FFPE LCM samples, we successfully evaluated 10,190 of 10,629 mutations (95.9%); the true positive rate was 99.3% (9,657 out of 9,728 mutations) for SNVs and 98.7% (456 out of 462 mutations) for indels. The results of validation sequencing and true positive rate are summarized in Supplementary Tables 14 and 15, respectively.

## Statistical analysis

Statistical analyses were performed using the R software (v.3.6.3). All *P* values were calculated using a two-sided analysis. Fisher's exact test or Mann–Whitney *U*-test was used for group comparisons. Survival analysis for TCGA dataset was performed using the R package survival[77] (v.3.2.11), wherein the overall survival time was determined from the date of diagnosis of breast cancer to the time of last follow-up or death, and survival curves were estimated using the Kaplan–Meier product-limit method. The differences in overall survival between the patients with der(1;16)-positive and der(1;16)-negative Luminal A breast cancer were tested for statistical significance using the log-rank test. We also performed multivariate analysis using a Cox proportional hazards regression model to evaluate the effect of der(1;16) on survival when adjusted for age and stage. The results of survival analysis are summarized in Supplementary Table 16. Multiple testing was corrected based on the Benjamini–Hochberg method to compare the frequency of driver mutations in TCGA Luminal A cancer cases with and without der(1;16) (Supplementary Table 17).

## Reporting summary

Further information on research design is available in the Nature Portfolio Reporting Summary linked to this article.

## Data availability

All WGS data have been deposited in the European Genome-phenome Archive under accession number EGAS00001006282. Data for estimation of mutation rate, phylogenetic analysis and timing estimation are available at https://doi.org/10.5281/zenodo.8015913 (ref. 78). TCGA datasets including WES .bam files, RNA-seq data in the TPM format and clinicopathological information were downloaded from TCGA data portal (https://portal.gdc.cancer.gov/), whereas the information about PAM50 mRNA subtypes was extracted from the study by Ciriello et al.[32]; if data were lacking, information was extracted from TCGA Network[27]. The publicly available GRCh37 (hg19, https://www.ncbi.nlm.nih.gov/assembly/GCF_000001405.13/) was used as a human reference genome in this study. We referred to the 1000 Genomes Project dataset (1000g2015aug, downloaded through ANNOVAR[64]), the gnomAD database (gnomad_genome, downloaded through ANNOVAR[64]), the GenomicSuperDups database (downloaded through ANNOVAR[64]), repetitive sequences reported in the UCSC Genome Browser[65], the COSMIC database (https://cancer.sanger.ac.uk/cosmic) and ClinVar database (https://www.ncbi.nlm.nih.gov/clinvar/), for variant annotation. COSMIC SBS signatures (v.3.1) were also obtained from COSMIC (https://cancer.sanger.ac.uk/signatures/downloads/). Source data are provided with this paper.

## Code availability

The R codes for estimation of mutation rate, phylogenetic analysis, and estimation of the timing of the MRCA emergence and der(1;16) acquisition are available at https://doi.org/10.5281/zenodo.8015913 (ref. 78).

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

**Acknowledgements** This work was supported by the Japan Agency for Medical Research and Development (AMED): The Core Research for Evolutional Science and Technology (CREST) (grant no. JP22gm1110011 to S.O.) and the Moonshot Research and Development Program (grant no. JP22zf0127009 to S.O.); the Japan Society for the Promotion of Science (JSPS): Scientific Research on Innovative Areas (grant no. JP15H05909 to S.O.) and KAKENHI (grant nos. JP19H05656 to S.O., JP20H0351 to N.K. and JP21K08664 to M. Kawashima); Ministry of Education, Culture, Sports, Science and Technology of Japan: The High Performance

Computing Infrastructure System Research Project (grant nos. hp160219, hp170227, hp180198 and hp190158 to S.O. and S.M.) and the Program for Promoting Research on the Supercomputer Fugaku (grant nos. hp200138, hp210167 and hp220163 to S.O. and S.M.) (this research used computational resources of the supercomputer K and Fugaku provided by the RIKEN Center for Computational Science); the Japan Science and Technology Agency (JST): Fusion Oriented REsearch for disruptive Science and Technology (FOREST) Program (grant nos. JPMJFR215V to N.K. and JPMJFR200K to M. Kawashima) and the Moonshot Research and Development Program (grant no. JPMJMS2022-25 to N.K.); Takeda Science Foundation (to S.O. and N.K.); Princess Takamatsu Cancer Research Fund (to N.K.); The Naito Foundation (to N.K.); Inoue Foundation for Science (to N.K.); AstraZeneca Externally Sponsored Research (to M. Toi) and Nanpuh Hospital (to S.O.). S.O. is a recipient of the JSPS Core-to-Core Program A: Advanced Research Networks. T. Nishimura and M.M.N. have been employed as joint research chairs with Nanpuh Hospital. We thank A. Ryu, A. Takatsu, T. Shirahari, F. Pu and K. Koishihara for technical assistance; K. Muta, Y. Yamaguchi and all the staff in the outpatient units and wards at Kyoto University Hospital and Adachi Hospital for their support in sample collection; the Kyoto Breast Cancer Research Network (KBCRN) Breast Oncology Research Network (BORN)-BioBank for providing samples; the Centre for Anatomical, Pathological and Forensic Medical Research at Kyoto University Graduate School of Medicine for preparing slides for microscopy and the TCGA Consortium and all its members for making publicly available their invaluable data.

**Author contributions** T. Nishimura, N.K., K.Y., T. Sakurai, T.R.K., M. Toi and S.O. designed the study. E.K., Y.C., M. Kawai, M. Sawada, T.I., E.S., M. Takada, M. Kawashima, K.K., J.T., M. Mandai and M. Toi provided specimens. T. Sakurai, T.R.K., Y.T., H.M. and H.H. performed histological analysis. T. Nishimura, N.K., Y.T., S.B., T. Nakagawa and M.H. performed sample preparation. T. Nishimura, N.K., K.Y., Y.T., Y. Shiozawa, R.S., Y.O., T.H., Y.I.-K. and K.A. performed mutation calling, validated the results and analysed CNAs, mutational signature, and clonal dynamics. T. Nishimura, N.K., M.M.N., Y.N., K.C., Y. Shiraishi and S.M. performed bioinformatic analysis. S.B. and K.T. performed FISH analysis. T. Nishimura, K.N., M. Matano, M. Saito and T. Sato established organoids. T. Nishimura, N.K., M. Toi and S.O. prepared the manuscript.

**Competing interests** The authors declare no competing interests.

**Additional information**
**Correspondence and requests for materials** should be addressed to Seishi Ogawa.

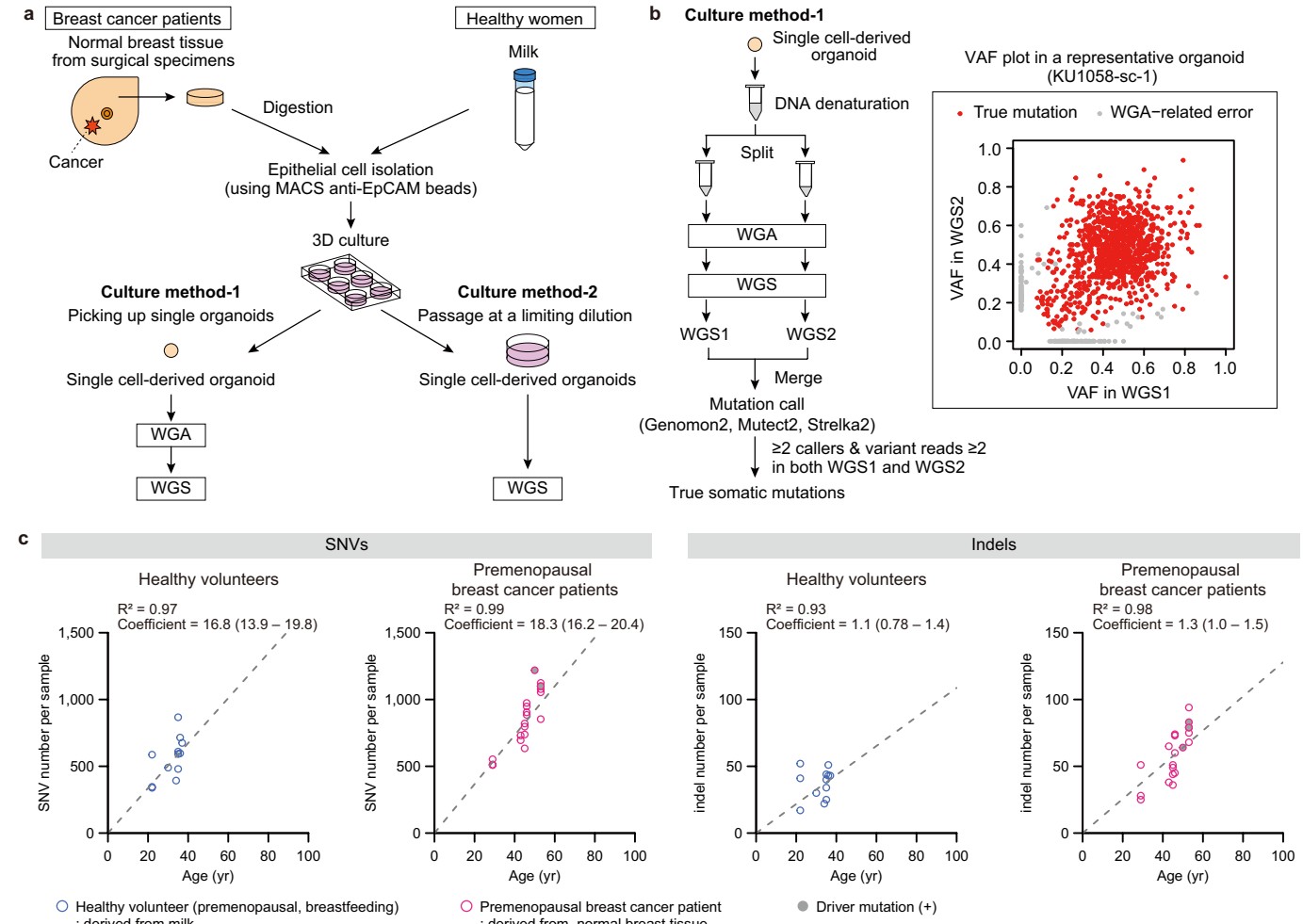

**Extended Data Fig. 1 | Methods for WGS of single cell-derived organoids.**
**a**, Schema of single normal epithelial cell-derived organoid establishment.
WGA, whole-genome amplification. **b**, Methods for mutation calling of WGA
samples established via culture method-1 shown in **a**. **c**, The number of SNVs
(left) and indels (right) in single organoids plotted against the age of
participants. Twelve organoids derived from milk of healthy breastfeeding
women (healthy volunteers, $n = 6$) and 20 derived from the normal breast tissue
of premenopausal breast cancer patients ($n = 6$) are shown. Linear regression
models assuming a zero intercept were applied to mean number of mutations
per participant and age, which are shown in grey dashed lines with $R^2$ and
coefficient values, wherein three organoids carrying driver mutations were
excluded from the analysis.

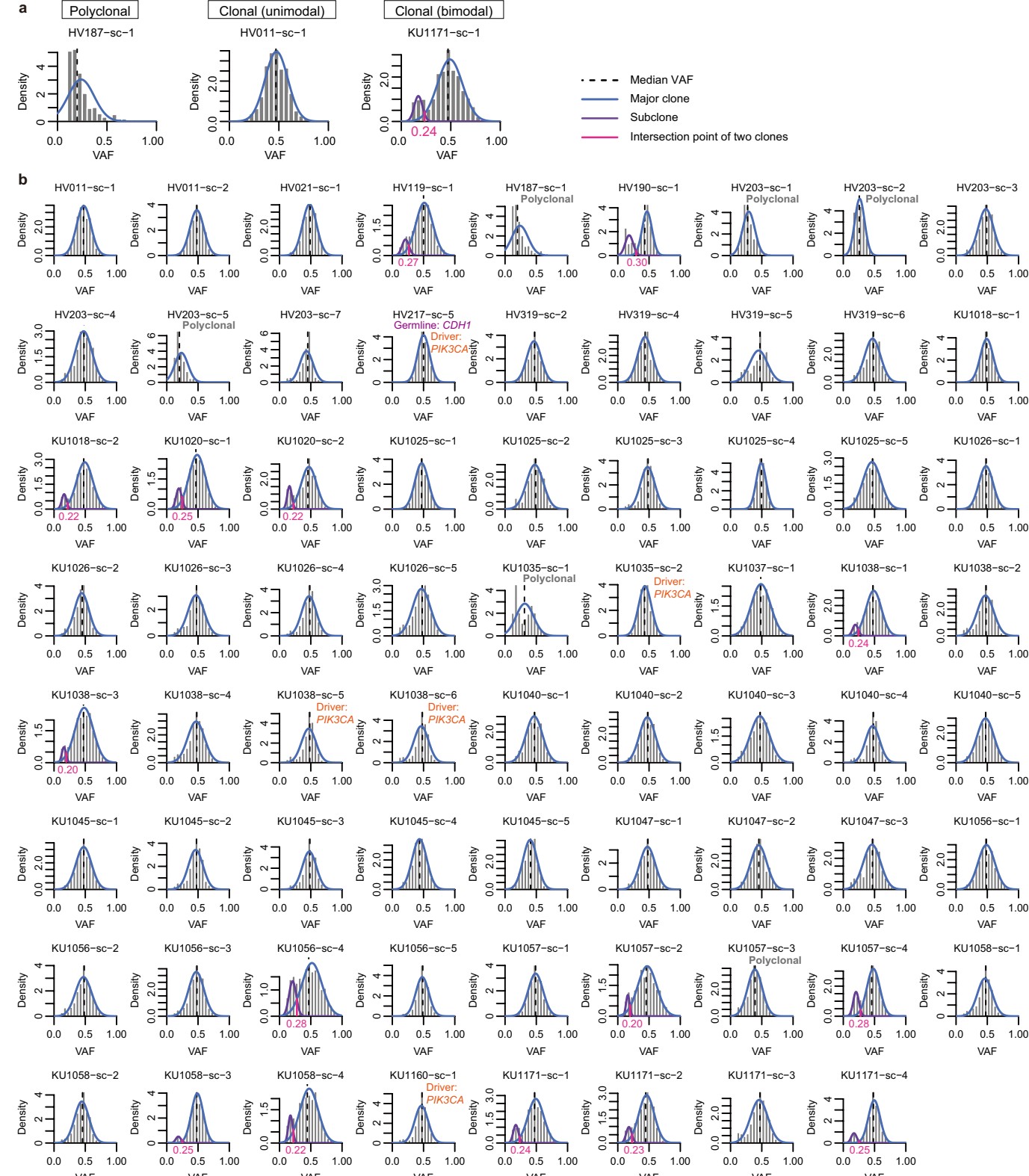

**Extended Data Fig. 2 | VAF distribution of SNVs in single cell-derived organoids. a**, VAF densities of SNVs in representative organoids. Organoids with median VAF values ≥0.4 were defined as clonal, wherein the existence of subclones and VAF density of each clone were evaluated using Gaussian mixture models. In bimodal organoids with subclones, mutations with VAF values lower than the intersection point of the two clones were eliminated as subclonal mutations. **b**, VAF densities of SNVs identified in 71 single cell-derived organoids are depicted. The germline variants and somatic driver genes mutated in each sample are indicated.

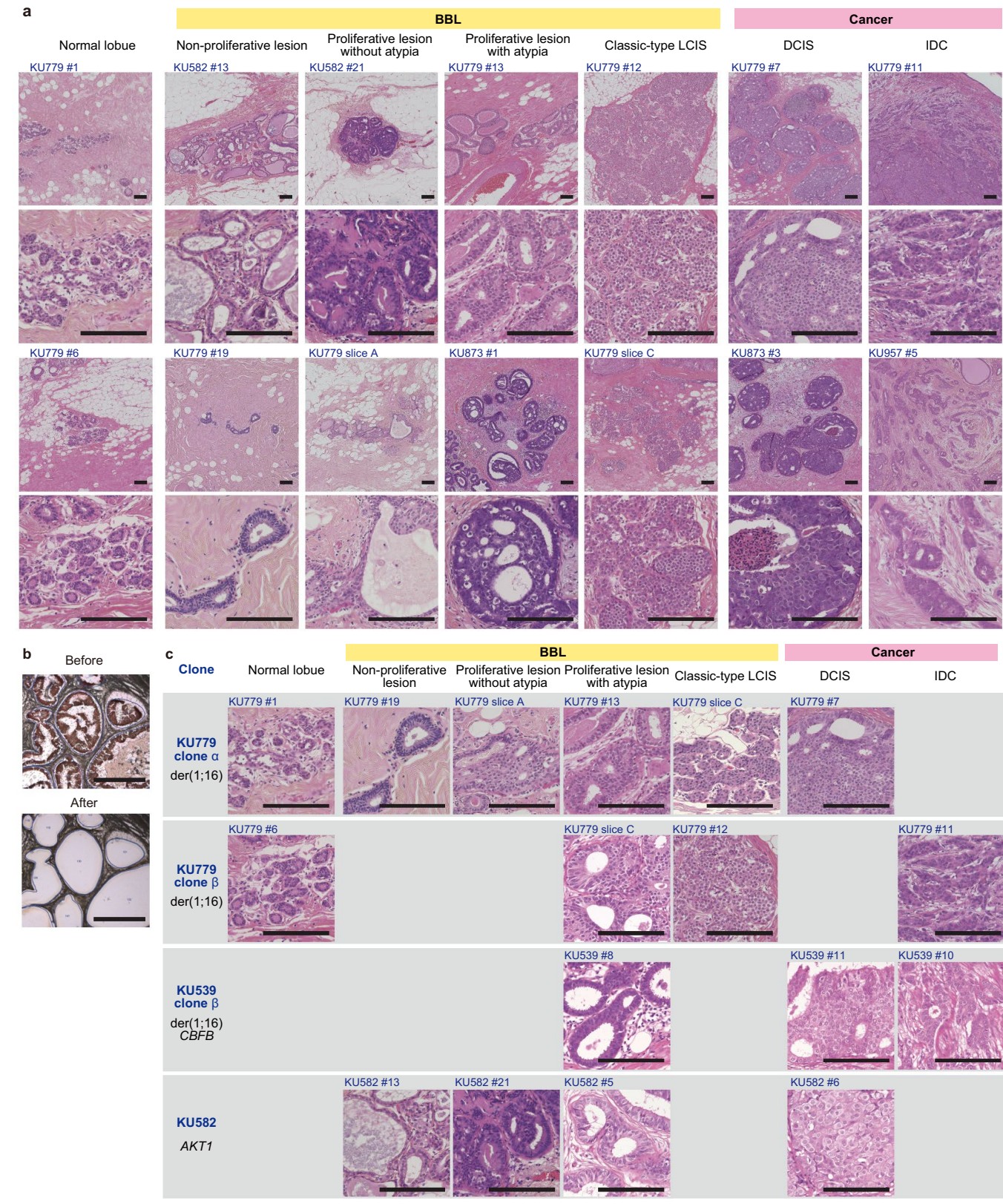

**Extended Data Fig. 3 | Multi-sampling via LCM to investigate clonal evolution of breast cancers. a**, Representative haematoxylin and eosin staining of normal lobules, benign breast lesions (BBL), and cancer lesions (top row), and the corresponding images at a high power magnification (bottom row) (14 out of 337 lesions). **b**, Representative LCM images before and after the dissection (one out of 194 lesions). **c**, Representative sequence of lesions originating from the same clones (four out of seven clones). Founder driver alterations in each clone are shown in black. Scale bar = 150 μm.

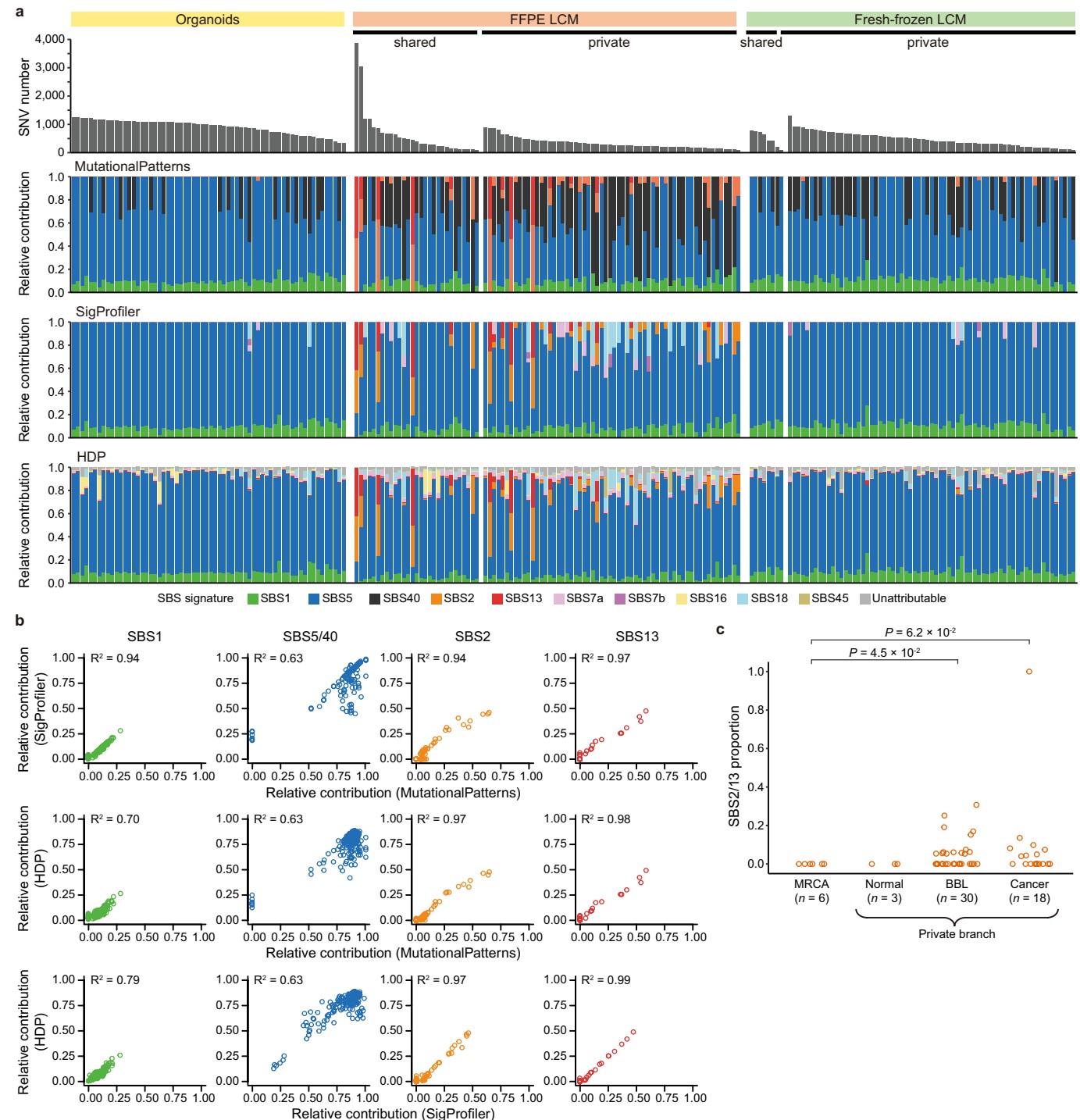

**Extended Data Fig. 4 | Analysis of mutational signatures. a**, Number of SNVs in each sample (top) and the relative contribution of SBS signatures extracted using MutationalPatterns (second panel), SigProfiler (third panel), and HDP (bottom), fitted to known COSMIC SBS signatures. Signature extraction was performed on branches with more than 100 SNVs in FFPE and fresh-frozen LCM samples. **b**, Scatter plots showing the relative contribution of mutations in each sample assigned to each signature by MutationalPatterns (x-axis) versus

SigProfiler (y-axis) (top), by MutationalPatterns (x-axis) versus HDP (y-axis) (middle), and by SigProfiler (x-axis) versus HDP (y-axis) (bottom), with the coefficient of determination ($R^2$) derived from Pearson's correlation. **c**, The proportion of APOBEC signatures (SBS2/SBS13) in the MRCA and peripheral private branches in normal, BBL, and cancer lesions in five FFPE multi-sampled premenopausal breast cancer patients (Supplementary Table 1), with $P$-values from the two-sided Mann–Whitney $U$ test.

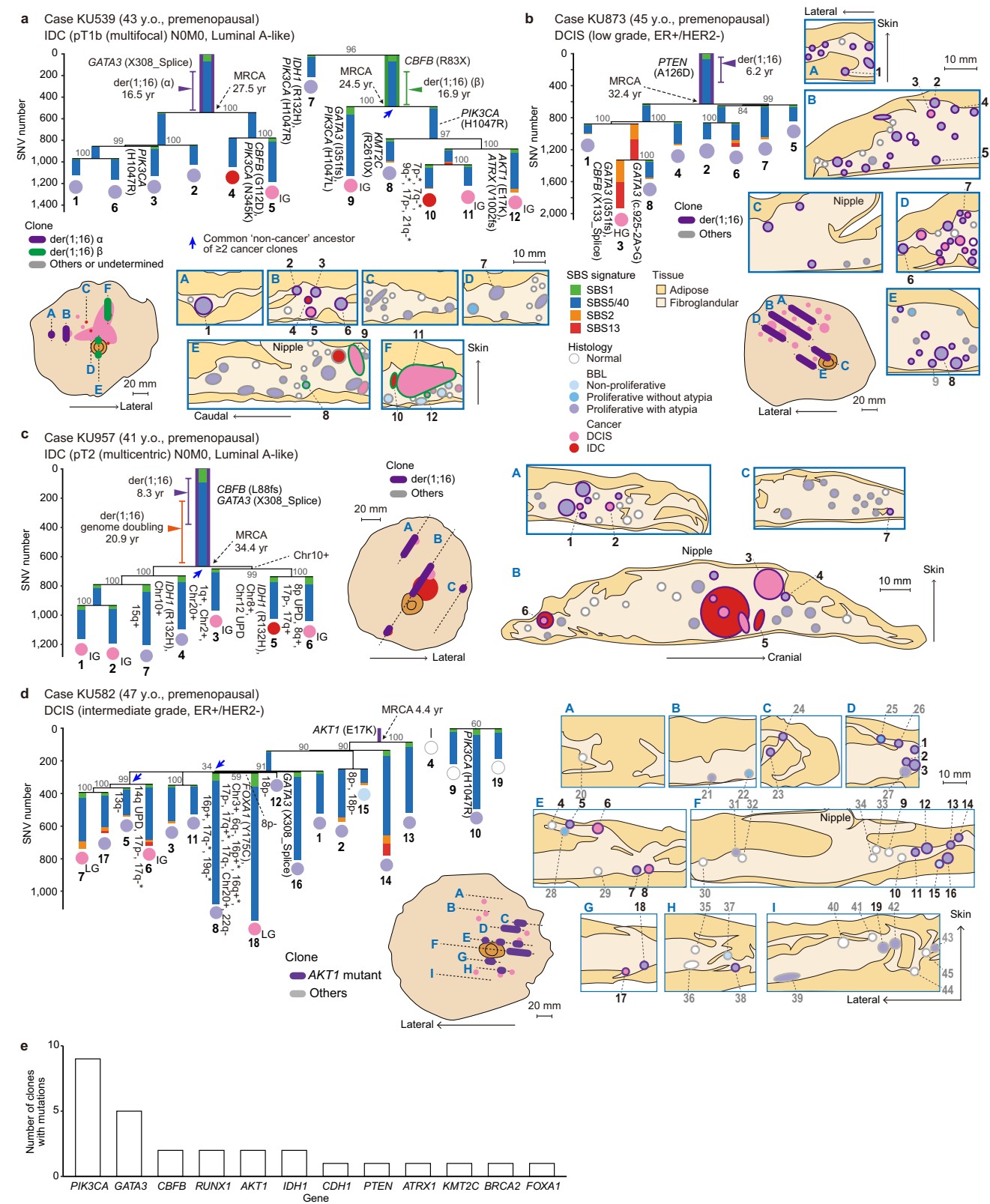

**Extended Data Fig. 5 | Life history of breast cancers in premenopausal patients. a–d**, Phylogenetic trees and the corresponding geographical maps of clones detected in cancer and non-cancer lesions in four premenopausal breast cancer patients who underwent total mastectomy (KU539 (**a**), KU873 (**b**),

KU957 (**c**), and KU582 (**d**)). All the representations follow those in Fig. 2a. LG, low grade DCIS; IG, intermediate grade DCIS; HG, high grade DCIS; UPD, uniparental disomy. **e**, Number of clones which had acquired each driver mutation in peripheral branches after MRCA emerged.

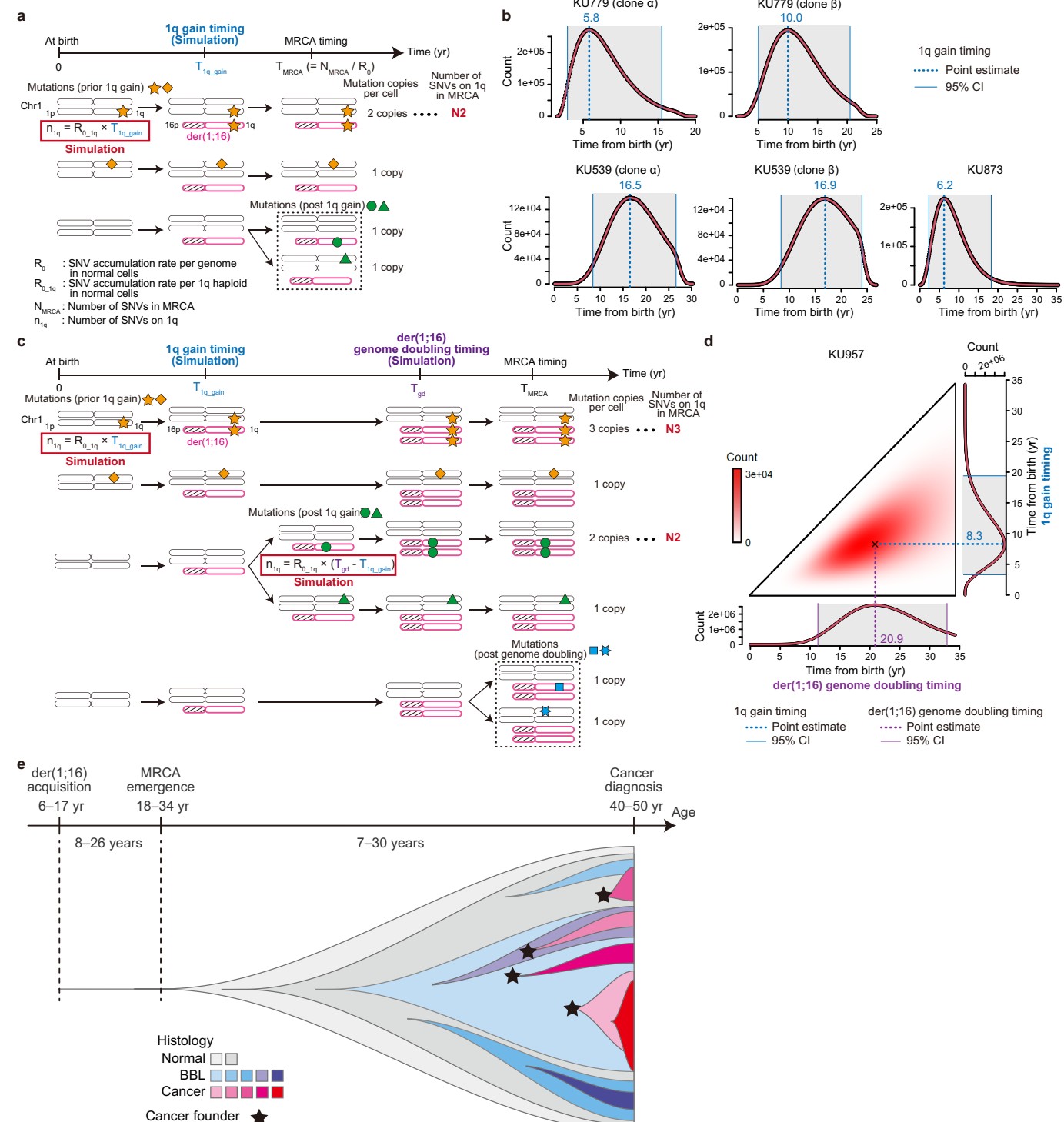

**Extended Data Fig. 6 | Estimation of the timing of der(1;16) acquisition.**
**a**–**d**, Schema of the simulation to estimate the timing of der(1;16) acquisition, that is, 1q gain timing ($T_{1q\_gain}$), in clones with one der(1;16) (**a**) and two der(1;16) derivatives (**c**), and the corresponding simulation results related to each case shown in Fig. 2a and Extended Data Fig. 5a–c (**b**,**d**) **(Methods)**. **e**, Schematic diagram of clonal evolution in premenopausal der(1;16)(+) breast cancer cases is shown with a time course. The colours of each clone depict the histology. The black-coloured stars indicate multiple cancer founder clones.

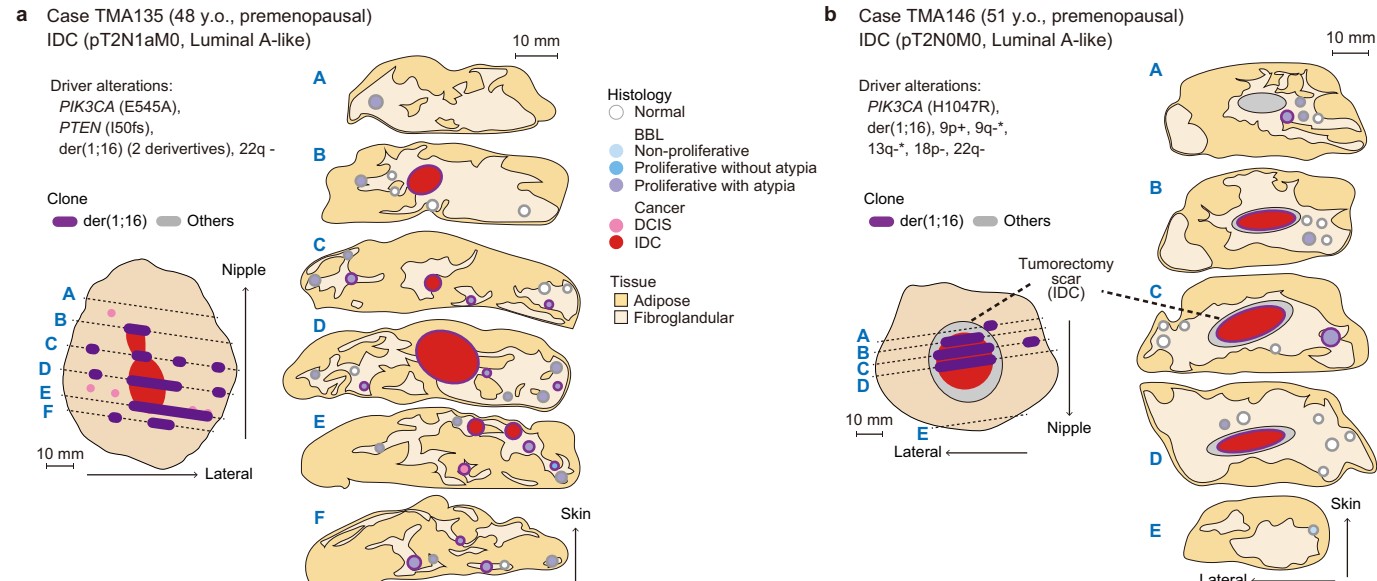

**a** Case TMA135 (48 y.o., premenopausal)
IDC (pT2N1aM0, Luminal A-like)

Driver alterations:
*PIK3CA* (E545A),
*PTEN* (I50fs),
der(1;16) (2 derivertives), 22q -

Clone
der(1;16)    Others

**Histology**
○ Normal
BBL
Non-proliferative
Proliferative without atypia
Proliferative with atypia
Cancer
DCIS
IDC

**Tissue**
Adipose
Fibroglandular

**b** Case TMA146 (51 y.o., premenopausal)
IDC (pT2N0M0, Luminal A-like)

Driver alterations:
*PIK3CA* (H1047R),
der(1;16), 9p+, 9q-*,
13q-*, 18p-, 22q-

Clone
der(1;16)    Others

Tumorectomy
scar
(IDC)

**Extended Data Fig. 7 | Expansion of der(1;16)(+) clones in premenopausal cases in another set of samples. a,b**, Geographical maps of clones detected via FISH in the surgical specimens of two premenopausal der(1;16)(+) breast cancer patients who underwent lumpectomy (left: overview of the surgical specimens, right: split faces of the sliced specimens indicated by dotted lines in the overview images). Driver mutations and copy number alterations (*, focal changes) detected in cancer lesions via targeted capture sequencing are shown. Histological results are depicted by the colours inside the circles. Coloured bars in the overview images and colours around the circles in the split faces show the areas wherein der(1;16)(+) clones were detected using FISH.

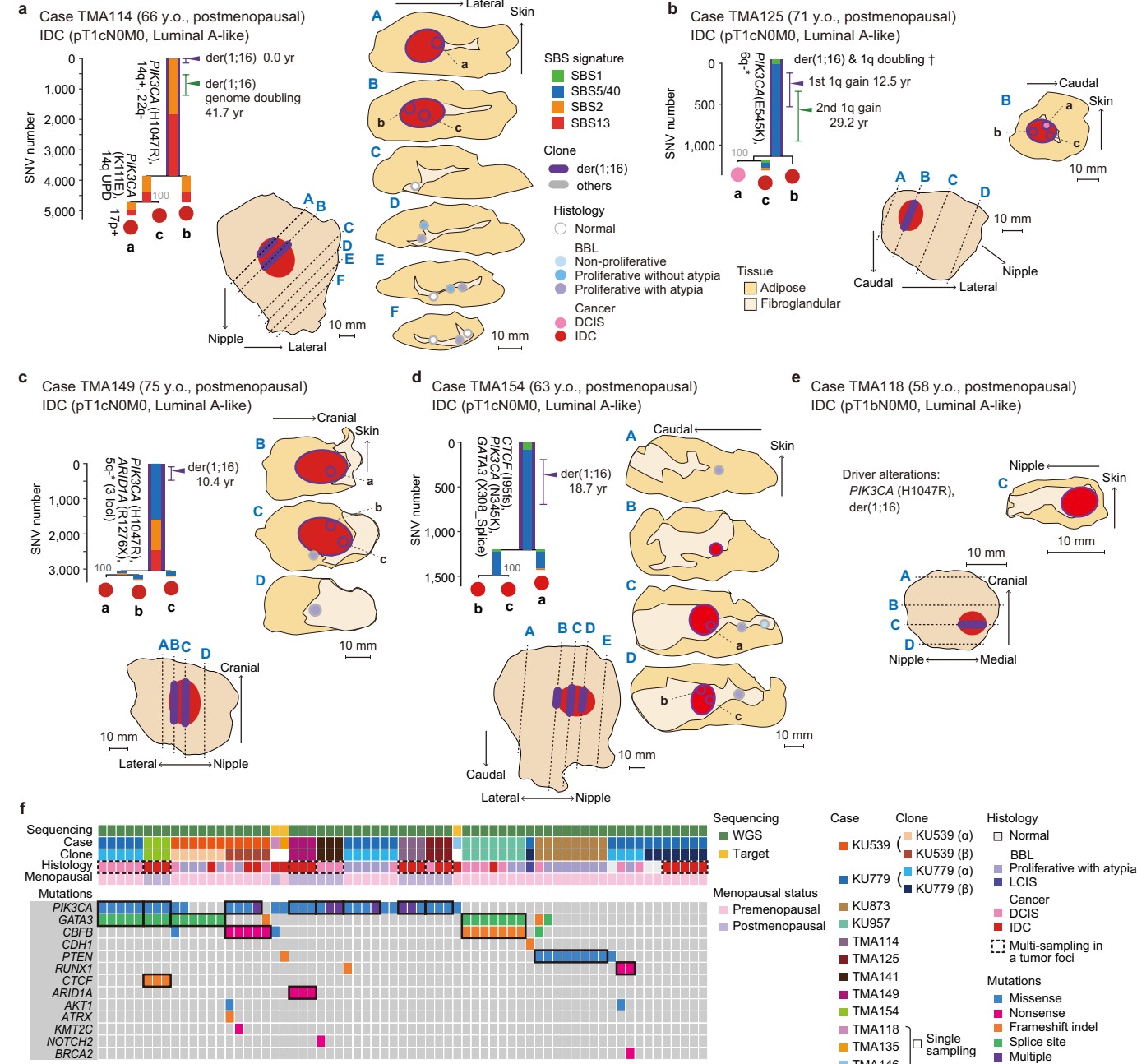

**Extended Data Fig. 8 | Expansion of der(1;16)(+) clones in postmenopausal cases in another set of samples. a–e**, Phylogenetic trees (**a–d**) and corresponding geographical maps of clones (**a–e**) detected in the surgical specimens of five postmenopausal der(1;16)(+) breast cancer patients who underwent lumpectomy (middle: overview of the surgical specimens, right: split faces of the sliced specimens indicated by dotted lines in the overview images). All the representations follow those in Fig. 2a. †, cancer lesions in TMA125 (**b**) carried one der(1;16) derivative and another 1q gain, and the

order of these alterations could not be determined; thus, timing of the first and second 1q gain was estimated; UPD, uniparental disomy. **f**, Landscape of driver mutations in der(1;16)(+) non-cancer and cancer lesions, including 64 WGS samples multi-sampled from nine cancer cases and three targeted sequencing samples from three cancer cases. Multiple samples obtained from a single tumour focus are enclosed in black dotted squares. Mutations enclosed in black bold squares represent shared mutations.

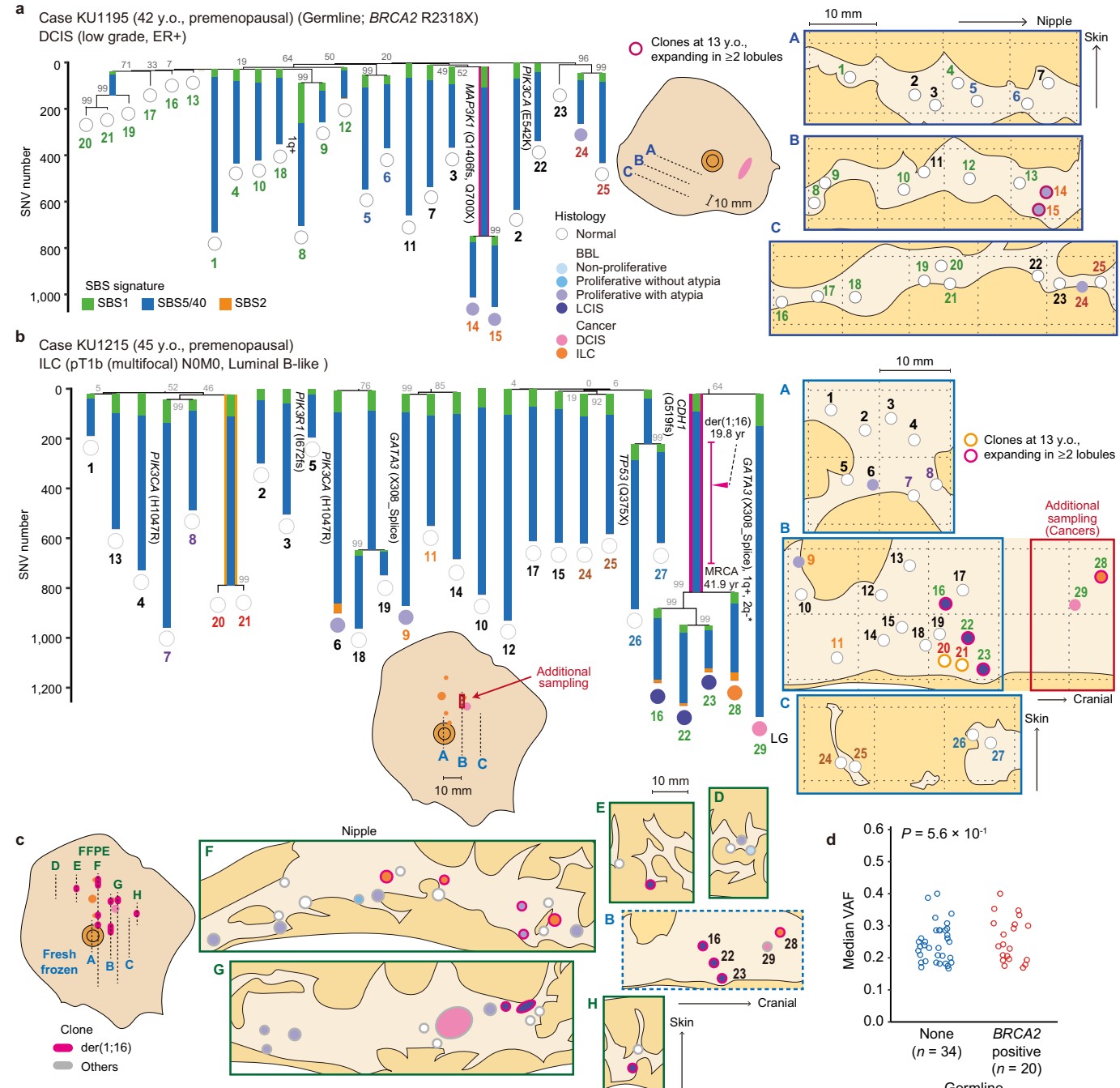

**Extended Data Fig. 9 | Clonal expansion in lobules without der(1;16).**
**a,b**, Phylogenetic trees and the corresponding geographical maps of clones
detected in multi-sampled lobules of two premenopausal breast cancer patients
with a pathogenic *BRCA2* variant (KU1195 (**a**)) and without pathogenic germline
variants (KU1215 (**b**)). All the representations follow those in Fig. 4a. In the case
of KU1215 (**b**), three der(1;16)(+) LCIS lesions (#16, #22, and #23) were detected
unexpectedly; thus, additional sampling was performed in red-coloured
tissue to investigate the correlation between der(1;16)(+) LCIS and cancers.

LG, low grade DCIS. **c**, Geographical maps of clones detected with FISH using
FFPE specimens in the case shown in **b** (KU1215). Coloured bars in the overview
image and colours around the circles in split faces depict the clones to which
samples belong. Circles numbered with black characters are samples analysed
via WGS, whereas unnumbered circles show the lesions analysed only via FISH.
**d**, Median VAF in histologically normal lobules carrying no somatic driver
alterations in breast cancer patients with and without pathogenic germline
variants, with *P*-values from the two-sided Mann–Whitney *U* test.

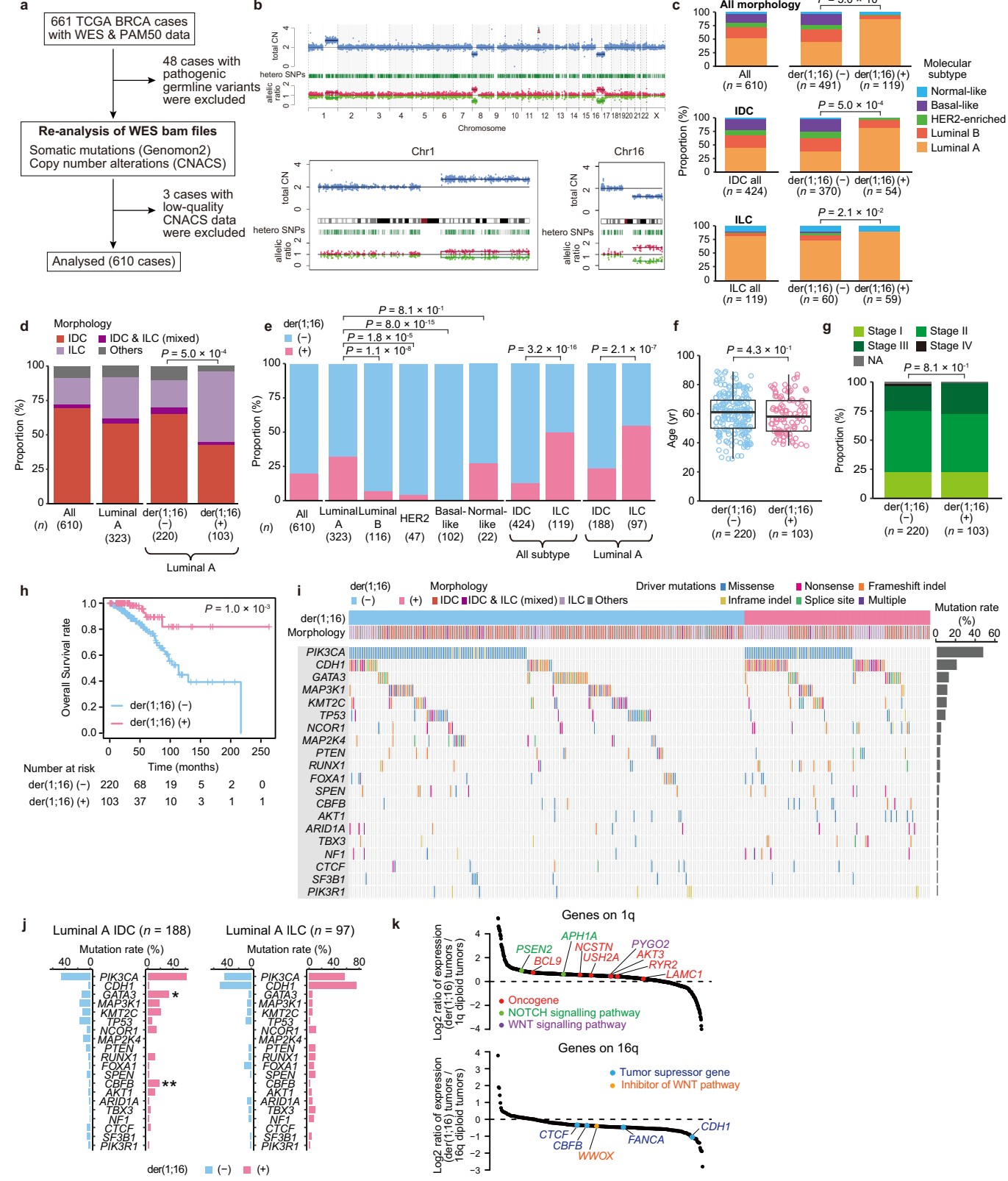

**Extended Data Fig. 10 |** See next page for caption.

**Extended Data Fig. 10 | Features of der(1;16)(+) breast cancer in TCGA cohort.**
**a**, Schema of re-analysis of TCGA breast cancer (BRCA) cohort (**Methods**).
**b**, Representative copy number plots in der(1;16)(+) cancers analysed using
CNACS. **c**, Distribution of PAM50 molecular subtypes in all cancers ($n = 610$),
IDC ($n = 424$), and ILC ($n = 119$), with and without der(1;16), with $P$-values from
two-sided Fisher's exact test. **d**, Distribution of morphology, which was
compared between der(1;16)(−) and der(1;16)(+) Luminal A cancers using
two-sided Fisher's exact test. **e**, Proportion of der(1;16)(+) cancer in each PAM50
molecular subtype, and IDC and ILC in all subtypes and Luminal A, respectively,
with $P$-values from two-sided Fisher's exact test. **f,g**, Distribution of age (**f**) and
stage (**g**) in der(1;16)(−) and der(1;16)(+) Luminal A cancer cases, with $P$-values
from two-sided Mann–Whitney $U$ test and Fisher's exact test, respectively.
Box plots show the median, first and third quartiles, with whiskers that extend
to the furthest value within a 1.5× interquartile range. **h**, Kaplan-Meier survival
analysis of der(1;16)(−) and der(1;16)(+) Luminal A cancer cases with $P$-values
from two-sided log-rank test. **i**, Landscape of driver mutations in der(1;16)(−)
and der(1;16)(+) Luminal A cancers ($n = 220$ and 103, respectively). Frequency
of each mutation is indicated on the right-hand side. **j**, Frequency of driver
mutations among der(1;16)(−) and der(1;16)(+) cancers in Luminal A IDC ($n = 188$)
and Luminal A ILC ($n = 97$), respectively. **, q<0.05; *, q<0.1 (from two-sided
Fisher's exact test with Benjamini–Hochberg adjustment). **k**, Ratio of average
expression of genes on 1q in der(1;16)(+) Luminal A tumours ($n = 103$) to those in
1q-diploid Luminal A tumours ($n = 77$) was depicted on the top, and the ratio of
genes on 16q in der(1;16)(+) Luminal A tumours ($n = 51$) to those in 16q-diploid
Luminal A tumours ($n = 83$) was depicted on the bottom. Coloured dots indicate
known oncogenes, tumour suppressor genes, or other genes on the NOTCH
and WNT signalling pathways.

# Reporting Summary

## Statistics

For all statistical analyses, confirm that the following items are present in the figure legend, table legend, main text, or Methods section.

| n/a | Confirmed | |
|---|---|---|
| ☐ | ☒ | The exact sample size (*n*) for each experimental group/condition, given as a discrete number and unit of measurement |
| ☐ | ☒ | A statement on whether measurements were taken from distinct samples or whether the same sample was measured repeatedly |
| ☐ | ☒ | The statistical test(s) used AND whether they are one- or two-sided *Only common tests should be described solely by name; describe more complex techniques in the Methods section.* |
| ☐ | ☒ | A description of all covariates tested |
| ☐ | ☒ | A description of any assumptions or corrections, such as tests of normality and adjustment for multiple comparisons |
| ☐ | ☒ | A full description of the statistical parameters including central tendency (e.g. means) or other basic estimates (e.g. regression coefficient) AND variation (e.g. standard deviation) or associated estimates of uncertainty (e.g. confidence intervals) |
| ☐ | ☒ | For null hypothesis testing, the test statistic (e.g. *F*, *t*, *r*) with confidence intervals, effect sizes, degrees of freedom and *P* value noted *Give P values as exact values whenever suitable.* |
| ☒ | ☐ | For Bayesian analysis, information on the choice of priors and Markov chain Monte Carlo settings |
| ☒ | ☐ | For hierarchical and complex designs, identification of the appropriate level for tests and full reporting of outcomes |
| ☐ | ☒ | Estimates of effect sizes (e.g. Cohen's *d*, Pearson's *r*), indicating how they were calculated |

*Our web collection on statistics for biologists contains articles on many of the points above.*

## Software and code

Policy information about availability of computer code

| Data collection | No software was used for data collection. |
|---|---|
| Data analysis | Detailed descriptions of the software and analysis have been provided in Online Methods. Sequencing data was processed using Genomon2. External bam files were converted to fastq format using biobambam. Mutation calling was performed using Genomon2, Mutect2 (GATK4), and Strelka2. Copy number analysis using sequencing data was performed using Control-FREEC and in-house pipeline CNACS. Number of clonal mutations in single cell-derived organoids was estimated using mclust. Phylogenetic analysis was performed using MEGA, treemut, and PyClone-VI. Mutational signature was evaluated using MutationalPatterns, SigProfiler Bioinformatic Tools MatrixGenerator and Extractor, HDP, and deconstructSigs. Significantly mutated genes were identified using dndscv. Statistical analyses were performed using R (3.6.3). Survival analysis was performed using the R package survival. |

CNACS is deposited in GitHub (https://github.com/OgawaLabTumPath/CNACS).

List of programs and softwares:
Genomon2 pipeline: version 2.6.2 (https://genomon.readthedocs.io/ja/latest/)
- Burrows-Wheeler Aligner: version 0.7.8 (https://sourceforge.net/projects/bio-bwa/)
- biobambam: version 0.0.191 (https://www.sanger.ac.uk/science/tools/biobambam)
- GenomonMutationFilter: version 0.2.1 (https://github.com/Genomon-Project/GenomonMutationFilter)
Xenome: version 1.0.0 (https://github.com/data61/gossamer)
Samtools: version 1.10 (https://github.com/samtools/samtools)
GATK4: version 4.1.2 (https://github.com/broadinstitute/gatk/releases)
Strelka2: version 2.9.3 (https://github.com/Illumina/strelka)

ANNOVAR: 2020-06-07 (https://annovar.openbioinformatics.org/en/latest/)
Integrative Genomics Viewer (IGV): version 2.3.8 (http://software.broadinstitute.org/software/igv/)
Control-FREEC: version 11.0 (https://github.com/BoevaLab/FREEC/releases)
mclust: version 5.4.7 (https://cran.r-project.org/web/packages/mclust/index.html)
MEGA: version 11.0.11 (https://www.megasoftware.net/)
treemut: version 1.1 (https://github.com/NickWilliamsSanger/treemut)
PyClone-VI: version 0.1.0 (https://github.com/Roth-Lab/pyclone-vi)
MutationalPatterns: version 3.4.0 (https://bioconductor.org/packages/release/bioc/html/MutationalPatterns.html)
SigProfiler Bioinformatic Tools MatrixGenerator: version 1.1.27 (https://github.com/AlexandrovLab/SigProfilerMatrixGenerator)
SigProfiler Bioinformatic Tools Extractor: version 1.1.1 (https://github.com/AlexandrovLab/SigProfilerExtractor)
HDP: version 0.1.5. (https://github.com/nicolaroberts/hdp)
deconstructSigs: version 1.8.0 (https://github.com/raerose01/deconstructSigs)
dndscv: version 0.0.1.0 (https://github.com/im3sanger/dndscv)
R: version 3.6.3 (https://cran.r-project.org/)
survival: version 3.2.11 (https://cran.r-project.org/web/packages/survival/index.html)

The R codes for phylogenetic analysis and estimation of the timing of the MRCA emergence and der(1;16) acquisition are available in Supplementary Notes 3 and 4, respectively.
The R codes for estimation of mutation rate in normal cells are available at https://doi.org/10.5281/zenodo.8002434.

For manuscripts utilizing custom algorithms or software that are central to the research but not yet described in published literature, software must be made available to editors and reviewers. We strongly encourage code deposition in a community repository (e.g. GitHub). See the Nature Portfolio guidelines for submitting code & software for further information.

## Data

Policy information about availability of data

All manuscripts must include a data availability statement. This statement should provide the following information, where applicable:
- Accession codes, unique identifiers, or web links for publicly available datasets
- A description of any restrictions on data availability
- For clinical datasets or third party data, please ensure that the statement adheres to our policy

All WGS data have been deposited in the European Genome-phenome Archive (http://www.ebi.ac.uk/ega/) under accession number EGAS00001006282.
Data for estimation of mutation rate and phylogenetic analysis are available at https://doi.org/10.5281/zenodo.8002434.
Data for the Figures and Extended Data Figures are available as Source Data.

WES bam files, RNAseq data in the TPM format, and clinicopathological information of TCGA datasets were downloaded from TCGA data portal (https://portal.gdc.cancer.gov/), whereas the information about PAM50 mRNA subtypes was extracted from the study by Ciriello et al. (DOI: 10.1016/j.cell.2015.09.033); if data were lacking, information was extracted from TCGA Network (DOI: 10.1038/nature11412).
The publicly available GRCh37 (hg19, https://www.ncbi.nlm.nih.gov/assembly/GCF_000001405.13/) was used as human reference genome in this study.
We referred to the 1000 Genomes Project dataset (1000g2015aug, downloaded through ANNOVAR (DOI:10.1093/nar/gkq603)), the gnomAD database (gnomad_genome, downloaded through ANNOVAR), the GenomicSuperDups database (downloaded through ANNOVAR), repetitive sequences reported in the UCSC Genome Browser (DOI:10.1101/gr.229102), COSMIC (the Catalogue Of Somatic Mutations In Cancer) database (https://cancer.sanger.ac.uk/cosmic), and ClinVar database (https://www.ncbi.nlm.nih.gov/clinvar/), for variant annotation.
COSMIC SBS signatures (v3.1) were obtained from COSMIC (https://cancer.sanger.ac.uk/signatures/downloads/).

## Human research participants

Policy information about studies involving human research participants and Sex and Gender in Research.

| | |
|---|---|
| Reporting on sex and gender | We analysed specimens derived from female breast cancer patients and healthy women to investigate clonal evolution of normal mammary epithelial cells into breast cancer in women. |
| Population characteristics | We enrolled 207 female breast cancer patients who underwent surgery at the Kyoto University Hospital, aged 26 to 92, and eight healthy breastfeeding women who delivered at the Kyoto University Hospital or Adachi Hospital, aged 22 to 37. The characteristics of the participants are summarised in Supplementary Table 1. |
| Recruitment | To estimate the rate of mutation accumulation in normal mammary epithelial cells, we enrolled 15 sporadic female breast cancer patients who underwent total mastectomy at the Kyoto University Hospital, and eight healthy breastfeeding women with adequate breast milk supply who had delivered at the Kyoto University Hospital or Adachi Hospital. These participants were recruited at random. |
| | For the analysis of cancer-related clonal evolution, All 156 female breast cancer patients who underwent surgery without any preoperative treatment at the Kyoto University Hospital from 2015 to 2017 and agreed to offer surgical specimens were recruited. Next, they were screened based on the pathology reports to select the cases with multiple large non-cancerous proliferative lesions near cancers. In total, five sporadic cases with available archival FFPE surgical specimens were found, and all the cases were analysed via sequencing. The details of case selection are shown in Online methods. We dared to select breast cancers accompanied by multiple proliferative lesions to explore life history of breast cancer, which resulted in the enrichment of der(1;16)(+) cancers. |
| | To further evaluate the pathological feature of der(1;16)(+) breast cancers, we obtained 33 breast cancer tissue cores (28 Luminal A-like invasive cancers and 5 ER(+)HER2(-) non-invasive cancers) in the tissue microarray provided by the Kyoto |

Breast Cancer Research Network (KBCRN) BORN (Breast Oncology Research Network)-BioBank, which was established using surgical specimens of randomly recruited cases. Two premenopausal and six postmenopausal der(1;16)(+) cancer cases were found by FISH analysis, and they all were evaluated in this study. The details of case selection are also shown in Online Methods.

For the analysis of non-cancer clones unrelated to cancer, we enrolled 3 premenopausal breast cancer patients who underwent total mastectomy without any preoperative treatment at the Kyoto University Hospital, to obtain enough amount of normal epithelium before atrophy due to menopause. The patients were recruited at random.

Ethics oversight | This study was reviewed and approved by the ethics committees of the Kyoto University and Adachi Hospital.

Note that full information on the approval of the study protocol must also be provided in the manuscript.

# Field-specific reporting

Please select the one below that is the best fit for your research. If you are not sure, read the appropriate sections before making your selection.

☒ Life sciences ☐ Behavioural & social sciences ☐ Ecological, evolutionary & environmental sciences

For a reference copy of the document with all sections, see nature.com/documents/nr-reporting-summary-flat.pdf

# Life sciences study design

All studies must disclose on these points even when the disclosure is negative.

Sample size | No statistical methods were used to determine sample size since this is an exploratory study. We enrolled as many patients as possible who provided consent for our study during the enrollment period between 1/Jan/2015 and 1/Dec/2020.

Data exclusions | For the analysis of TCGA BRCA cohort, we excluded whole-genome amplified samples to avoid artefactual mutation callings. We also excluded samples with pathogenic germline variants and samples with low quality copy number data, to evaluate the association of specific copy number events, clinicopathological information, and somatic mutation profiles in the sporadic BRCA cohort.

Replication | We did not attempt replication in this study, except for WGS of WGA organoid samples, in which two independent experiments were performed to eliminate WGA-related sequencing errors. The details of sequencing methods for WGA samples are described in Online Methods.

As for sequencing experiments, we validated the results by confirmatory targeted capture sequencing for variants (n = 702) from 22 samples randomly selected from 126,653 variants from 228 samples. We also validated variants detected in the most common recent ancestors in FFPE multi-sampled cases as many as possible (n = 10,190 out of 10,629 variants) in 27 samples from five cases (out of 84 samples from 10 cases) to ensure the accuracy of phylogenetic tree reconstruction and the subsequent estimation of timing for initial events. The details of validation sequencing are described in Online methods and Supplementary Tables 14 and 15.

Randomization | Not applicable since this is a case-series study which was therefore not planned to detect any difference in effects between the cohorts with and without intervention.

Blinding | Pathologists were blinded to the genetic alterations in each sample during histopathological evaluation.

# Reporting for specific materials, systems and methods

We require information from authors about some types of materials, experimental systems and methods used in many studies. Here, indicate whether each material, system or method listed is relevant to your study. If you are not sure if a list item applies to your research, read the appropriate section before selecting a response.

## Materials & experimental systems

| n/a | Involved in the study |
|-----|----------------------|
| ☐ ☒ | Antibodies |
| ☒ ☐ | Eukaryotic cell lines |
| ☒ ☐ | Palaeontology and archaeology |
| ☒ ☐ | Animals and other organisms |
| ☒ ☐ | Clinical data |
| ☒ ☐ | Dual use research of concern |

## Methods

| n/a | Involved in the study |
|-----|----------------------|
| ☒ ☐ | ChIP-seq |
| ☒ ☐ | Flow cytometry |
| ☒ ☐ | MRI-based neuroimaging |

## Antibodies

Antibodies used | ER (supplier name, Roche Diagnostics; catalogue number, 790-4325; clone name, SP1)

| Antibodies used | PR (supplier name, Roche Diagnostics; catalogue number, 790-2223; clone name, 1E2)<br>HER2 (supplier name, Roche Diagnostics; catalogue number, 790-2991; clone name, 4B5)<br>Ki-67 (supplier name, Agilent Technologies; catalogue number, M7240; clone name, MIB-1)<br>CD326 (EpCAM) MicroBeads human (supplier name, Miltenyi Biotec; catalogue number, 130-061-101)<br>CD45 Microbeads human (supplier name, Miltenyi Biotec; catalogue number, 130-045-801)<br>pan-cytokeratin antibody cocktails (AE1/AE3) (supplier name, NICHIREI; catalogue number, 412811)<br>CK5 (supplier name, Leica Biosystems; catalogue number, CK5-L-CE-H; clone name, XM26)<br>E-cadherin (supplier name, Agilent Technologies; catalogue number, M3612; clone name, NCH-38) |
|---|---|
| Validation | ER (human; IHC; validation: Manufacturer - https://pim-eservices.roche.com/eLD/web/pi/en/documents/download/2b989f55-4533-ea11-fc90-005056a71a5d)<br>PR (human; IHC; validation: Manufacturer - https://pim-eservices.roche.com/eLD/web/pi/en/documents/download/76ea4fea-e112-ea11-fa90-005056a772fd)<br>HER2 (human; IHC; validation: Manufacturer - https://pim-eservices.roche.com/eLD/web/pi/en/documents/download/fc569ff5-2236-ea11-fc90-005056a71a5d)<br>Ki-67 (human; IHC; validation: Manufacturer - http://webzis.cytopathos.sk/Protilatky/store/MIB1.pdf)<br>CD326 (EpCAM) MicroBeads human (human; MicroBeads conjugated to monoclonal antibody: validation, Manufacturer - http://www.ulab360.com/files/prod/manuals/201603/28/596760001.pdf)<br>CD45 Microbeads human (human; MicroBeads conjugated to monoclonal antibody; validation: Manufacturer - https://www.miltenyibiotec.com/upload/assets/IM0001290.PDF)<br>pan-cytokeratin antibody cocktails (AE1/AE3) (human; IHC; validation: Tohyama R, Kayamori K, Sato K, et al. Establishment of a xenograft model to explore the mechanism of bone destruction by human oral cancers and its application to analysis of role of RANKL. J Oral Pathol Med. 2016; 45(5):356-64.)<br>CK5 (human; IHC; validation: Manufacturer - https://shop.leicabiosystems.com/ja-jp/actions/ViewProductAttachment-OpenFile?LocaleId=en_US&DirectoryPath=SDSs&FileName=ck5-l-ce.pdf&UnitName=LBS)<br>E-cadherin (human; IHC; validation: Manufacturer - http://webzis.cytopathos.sk/Protilatky/store/Cadherin%20E.pdf) |

