## [Peer Review File · Nature]

Manuscript Title: Evolutionary histories of breast cancer and related clones

Reviewer Comments & Author Rebuttals

Reviewer Reports on the Initial Version:

Referee expertise:

Referee #1: breast development

Referee #2: somatic evolution

Referee #3: cancer genomics

Referees' comments:

Referee #1 (Remarks to the Author):

The manuscript by Nishimura et al describes impressive LCM microdissection and whole genome sequence analysis of a range of lesions in a number of breast cancer patients (at least 16 patients in depth) to track mutations in overt lesions and adjacent normal tissue across cm's of breast tissue. They describe driver mutations in the earliest cancer ancestors and the acquisition of additional mutations in cancer foci. Notably, they estimate the evolution of these mutations across time in their cohort of breast cancer patients, concluding an unexpectedly early origin for mutant cancer ancestor clones. The study represents an enormous amount of work, however, further insights into the nature of the earliest driver mutations in the MRCAs are required. One of the major discoveries was a potential role for the der1;16 translocation in a subset of luminal A cancers - a key question is whether this reflects loss of Ecadherin/CDH1. Is there evidence that CDH1 loss is an initial driver of (subsequent) transformation of non-cancer ancestor clones or are there other mutated/deregulated genes such as transcription factor(s) at this locus that serve as the driver? Further analysis of this locus and whether there is haploinsufficiency of CDH1 is warranted as this represents one of the major conclusions of this study.

It should be clearly stated in the text how many patients per cohort were microdissected in detail. For example, five were analyzed from one cohort (line 142) but data for at least 7 women are shown in the main figures, plus other patients in the Extended figures. A Table summarizing the data from deep sequence analysis of multiple lesions from key patients would be helpful. Further, a Table documenting the types of atypia would be useful.

The histopathology of the specimens is paramount to the paper: were the proliferative lesions defined by mitotic figures or was KI67 staining also performed? How were normal lobules distinguished from proliferative lesions without atypia? There are at least 4 different types of TDLUs that vary between individuals and with differentiation and menopause status. An entire sequence of

lesions should be shown for 3 patients rather than showing a variety of sections from a range of patients in Ext data Fig 3.

Regarding the single cell organoid or 3D culture work: why do subclonal mutations arise given that these structures are not passaged- these are freshly plated cells that are cultured for a very short time and not passaged. In addition, please provide the breakdown of organoids derived from breast milk versus those from BC patients. Were any differences seen?

An effect of menopause was seen in the 'organoid' cultures (line 110) for SBS5 mutations. The cohorts also differed in terms of menopause status (Ext Fig 2) - more detail on the effect of this on mutation accumulation/type amongst the different cohorts is required.

Discussion Line 302- one primary factor (not listed here) for less mutation burden in postmenopausal tissue is that it has undergone involution and there is much less epithelial content, thus clones will have been extinguished.

Minor points:

Fig 4a,b- the colors of the clones expanding in > 2 lobules and the superimposed mutations from the different lesions are not clear. These need to be revised.

Please give more detail on the key features of SBS1 and SBS5 signatures.

Referee #2 (Remarks to the Author):

In their study, Nishimura, Kakiuchi et al. present a reconstruction of the clonal history of breast cancer and related normal or neoplastic clones that reside nearby. To set a baseline of mutagenesis in normal breast tissue, they whole-genome sequence single-cell derived breast organoids and arrive at a mutation rate of roughly 19 SNVs per year prior to menopause. Subsequently, they interrogate the clonal history of breast cancer and nearby clones using laser capture microdissection followed by whole-genome sequencing, which has now become a common and reliable method to detect somatic mutations in normal tissues.

Echoing what has been seen in other normal tissues, apparently normal breast tissue harbours canonical cancers driver mutations, mainly in PIK3R1 and PIK3CA. The most unique feature of precancerous breast tissue identified in this study is the prevalence of a common unbalanced translocation, der(1;16). Early clones harbouring der(1;16) spread through the normal breast and forms a tissue bed from which multiple (pre-)cancerous lesions can emerge.

I find the study exceedingly clear, the findings interesting and the conclusions well-supported. Nevertheless, I would like to see some improvements and adjustments as outlined below.

Comments

- The presence of SBS7a (and to a lesser extent, SBS8) is somewhat puzzling to me. Rather than actually being present in these samples, I suspect the emergence and fitting of these signatures may

be artefactual. One way to test this is to interrogate dinucleotide variants. True SBS7a should be accompanied by numerous CC>TT dinucleotide changes (DBS1), whereas true SBS8 is usually associated with CC>AA dinucleotide changes. Assessing the presence of these mutations can lend weight to the observation that these signatures are present or, if these dinucleotide changes are not present, indicate that these signatures are not truly present in these samples.

- Given that these signatures are absent from the normal breast organoids but present in normal breast LCM cuts, could they in some way reflect artefactual mutations due to the LCM/FFPE experimental design? The validation experiment using targeted resequencing goes some way to assuage this, but from inspecting Supplementary Table 4, it appears the mutations to be validated all resided in genes and therefore might not be representative of the non-genic regions. Is this correct or am I misinterpreting the table?

- If is the case that these signatures are not strongly supported by double base substitutions, I would advise to take a closer look at and revise the mutational signature analysis. From personal experience, the HDP package (<https://github.com/nicolaroberts/hdp>) has always yielded good mutational signature results (that might need further deconvolution), but I'll leave the particulars to the authors to pursue as they see fit.

- Doubts about SBS7a and SBS8 notwithstanding, for the purpose of timing MRCAs and copy number gains (including der(1;16)), it would be best to exclusively use mutations that can be assigned to the clock-like signatures, SBS1 and SBS5, for the best estimate, given only those signatures are present in the normal breast organoids. Including sporadic signatures in this analysis would alter the timing estimate.

- Is there any evidence for haploinsufficiency of BRCA2 in the patient with a BRCA2 germline mutation? While this could have manifested as SBS3, which is not identified in the patient within the study, this phenotype could also manifest as an excess of indels following ID6.

- Is there any effect of carrying a cancer driver mutation on the mutation burden of the normal breast glands sampled? I.e., do those with a PIK3R1/PIK3CA mutation have more somatic mutations than those without? From a quick visual inspection, this seems to be the case for KU1206 in Fig. 4a and the organoids in Fig. 1d, but it would be nice to make this more substantial, if possible. In addition, the text mentions the presence of driver mutations leads to a higher clonality (l. 242-244). It would be good to see a quantification of this and test it statistically.

- An additional analysis for this study is to estimate telomere length from the WGS. This could support the notion laid out in the discussion, that the decrease in mutation rate after menopause, as well as the reduction of mutations due to parity, is related to a decreased rate of cell turnover/a population of cells dormant stem cells becoming reactivated after pregnancy. Telomere length was also used as a line of evidence in the cited tobacco smoking study.

Minor comments

- "Given that ... Luminal A cancer." (l. 354-358). I am not sure I follow this. Would the difference in der(1;16) clone size between pre- and postmenopausal women simply not reflect the time at which these were acquired? I.e., the earlier the acquisition of der(1;16), the larger the size of the clone?

- A study that echoes the findings in this paper of a tissue root leading to multiple cancer lesions is a study on precursors of Wilms tumour and associated benign lesions (PMID: 31806814), fuelled by somatic loss of imprinting of a locus on 11p. Since this is an epigenetic change, this study underscores the point in the discussion that genomic changes may not explain all of carcinogenesis. In addition, this point is also supported by similar findings in malignant rhabdoid tumour, where the

precursor clone was found to be genomically virtually identical to the cancer (PMID: 33658498). These are simply some examples of similar processes observed in other cancers and precancerous lesions, biased to childhood cancers because of my background, and form by no means an exhaustive list.

- Fig 1d: it is difficult to distinguish the colours for SBS1 and SBS5, so I advise using a colour palette that is easier to tell apart. This of course, pertains to all the figures using these colours for SBS1 and SBS5.

- Fig 2a: it would be nice to add confidence intervals to the estimate of the der(1;16) timing (such as those presented in Fig 3e) as a bar alongside the branch. This pertains to Fig 3a-c as well.

- ED Figure 9: the MutationalPatterns plots seem to have a dark blue bar that is unexplained by the legend. What is this signature? It would also be good to use a different colour for it.

I congratulate the authors on an interesting study, it was a joy to read the manuscript.

Tim Coorens

Referee #3 (Remarks to the Author):

In this manuscript by Nishimura and Kakiuchi et al. claim to show the entire life history of breast cancer from its origin to clinical diagnosis. To accomplish this, the authors reconstruct phylogenies from WGS data derived from microdissected FFPE samples of 5 patients, complemented by mutation rate estimation based in numerous clonally-derived organoids from normal and malignant breast tissue. While the topic is an interesting one and the extensive sampling of lesions across the breast make for a rather unique (albeit heterogeneous) dataset, the primary finding that mutant clones arise early and independently in the breast around puberty and decades before cancer development is not novel. This is by now expected given reports in other tissues. The analyses used to arrive at this conclusion are overly complex while relying on standard techniques such as phylogeny reconstruction, mutational signature analysis. Unfortunately, because the presentation of the findings lacked clarity, the reader is left with far more questions than answers and uncertainty regarding the claims.

Major Comments:

The manuscript itself is difficult to follow, lacking clarity on numerous points – but most crucially the key messages from the study. From the abstract, “Here we show the entire life history of breast cancer from its origin to clinically diagnosed cancer using whole-genome sequencing (WGS) followed by phylogenetic analysis of multiple microdissected samples of genetically related cancer and non-cancer clones.” What are the authors trying to claim? Are they characterizing normal tissue? Is the study longitudinal? The abstract, introduction, and summary of findings/discussion seem quite disconnected. As a result, the reader is left to interpret this themselves.

The introduction poorly overviews the field of somatic mutations in normal and pre-cancerous tissue which has exploded in the past years. One of the chief findings of this manuscript is that oncogenic drivers emerge at a relatively early age (in the discussion it is noted that this may occur during

embryogenesis although elsewhere the times center on puberty). However, this is not a novel finding as numerous tissues such as the epidermis, esophagus, endometrium, and hematopoietic system have been demonstrated to harbor somatic alterations in non-malignant, phenotypically normal cells. It would be helpful to frame the current findings in light of what is already known and to discuss how this study differs in approach and/or main findings.

The most interesting finding seems to be the presence of mutant clones harboring known driver alterations such as der(1;16) throughout the mammary gland. Presumably these alterations arising independently at different times and strongly selected for in this tissue and clonally expand relative to neighboring cells that lack this event. This leads to the hypothesis that multiple cancer founders can exist, contributing to genetic heterogeneity in the subsequent cancer. While potentially appealing, this requires further support and clarification. Is this only true of multi-focal cancers such as those selected here. Is this unique to der(1;16) harboring lesions because of loss of E-cadherin? Additionally, how do these observations relate to the findings of Erickson et al (Nature 2022) who used spatial transcriptomics and copy number inference to examine benign and malignant prostate tissue. One might anticipate some parallels in multi-focal prostate lesions.

Beyond the challenges with the text, the main figures are cluttered, overly reliant on text, and lack broad themes/takeaways to support the authors primary findings. Figure 1 includes methods that don't belong in the main text or that could be summarized at a higher-level view with details to be included in the methods section and supplemental figures. The schematics of tissue-samplings (Fig 3 etc) are useful, but distracting and don't seem to provide much insight into the findings that accompany them. One or two could be used as an example but there are far too many too meaningfully digest or contrast. There is also a lack of consistency in the legends leaving the reader with yet more questions than answers about the study.

Throughout the text, the authors discuss clones as being cancerous and yet this is not defined clearly. There are not cancerous clones, but rather clones that exist within normal, pre-cancerous, or cancerous tissue. Presumably, the authors are referring to "clones carrying breast cancer mutations" as noted initially in the abstract. There are numerous instances throughout the manuscript that would benefit from clarifying this.

The methods as written lack the necessary information to reproduce these findings and warrant sufficient additional information as well as a github repository for the code used in analysis. This is especially true for the primary analysis constructing patient phylogenetic trees. There is no information provided on these phylogenies in terms of homoplasy or branch support. There is also little information on how SNV number is converted to chronological age. How do the authors explain the lack of additional SNVs being gained with age in the postmenopausal samples (Fig 1d)? It seems that after the age of ~50 the number of observed SNVs plateaus. Does this have an impact on the chronological ages? From a technical perspective, it seems this could be due to inadequate sampling/coverage of very small clones. Perhaps this could this be addressed via deep targeted sequencing?

Additionally, the fact that there are many more mutations present in the FFPE vs FF and organoid samples raises the concern that these are largely artifactual. While not surprising as this has been

repeatedly seen, the impact on the conclusions/timing estimates is not addressed and potentially concerning.

The ordering of clonal/subclonal events could be examined more extensively through the incorporation of copy number information. However, it is hard to tell the extent to which this information should be included (or not) due to the lack of summary information provided in the primary figures or quality control provided in the methods/figures. Nonetheless given the known role of copy number in breast pathogenesis, it would be important to investigate this.

Minor Comments:

The authors have taken great care to disclose the software versions used for their bioinformatics analysis; however, there is inconsistent citations provided for the tools that are used. Some examples: BWA line 843, biobambam line 844, and GenomonMutationFilter line 849 are all missing a citation, but samtools, Xenome, and others have citations present. All tools and pipelines used should be cited appropriately.

Units are not properly reported leading to confusion throughout the manuscript (e.g., extended data figure 4 "...were increased by 0.1 year to...").

Grammatical errors throughout.

Author Rebuttals to Initial Comments:

Referees' comments:

Referee #1 (Remarks to the Author):

The manuscript by Nishimura et al describes impressive LCM microdissection and whole genome sequence analysis of a range of lesions in a number of breast cancer patients (at least 16 patients in depth) to track mutations in overt lesions and adjacent normal tissue across cm's of breast tissue. They describe driver mutations in the earliest cancer ancestors and the acquisition of additional mutations in cancer foci. Notably, they estimate the evolution of these mutations across time in their cohort of breast cancer patients, concluding an unexpectedly early origin for mutant cancer ancestor clones. The study represents an enormous amount of work, however, further insights into the nature of the earliest driver mutations in the MRCAs are required. One of the major discoveries was a potential role for the der1;16 translocation in a subset of luminal A cancers - a key question is whether this reflects loss of Ecadherin/CDH1. Is there evidence that CDH1 loss is an initial driver of (subsequent) transformation of non-cancer ancestor clones or are there other mutated/deregulated genes such as transcription factor(s) at this locus that serve as the driver? Further analysis of this locus and whether there is haploinsufficiency of CDH1 is warranted as this represents one of the major conclusions of this study.

Reply:

We appreciate the reviewer's comments. To address the reviewer's questions, we first tested whether or not *CDH1* 'mutations' are significantly positively selected in breast cancer samples even without accompanying 16q loss (i.e., heterozygous mutations) using the large TCGA dataset. In total, we found 86 cases harbouring *CDH1* mutations, most (94%) of which had biallelic loss of *CDH1* either by der(1;16) ($n=46$), other $-16q$ ($n=21$), or uniparental disomy (UPD) ($n=14$). Heterozygous mutation was found in only five cases. Except for one frameshift change, all were variants of unknown significance that were rarely registered in the COSMIC database. These findings suggest that *CDH1* is a bona fide 'recessive' tumour suppressor gene and a haploid loss of *CDH1* alone may not be sufficient for positive selection/clonal expansion during breast cancer development.

To further investigate the mechanism of der(1;16)-associated clonal expansion, we newly analysed the effect of allelic imbalances caused by der(1;16), i.e., trisomy 1q and monosomy 16q, on gene expression, using the large TCGA dataset from 323 Luminal A breast cancers, including 103 der(1;16)(+) cases. As shown in **ED_Fig. 10k**, the effect of allele dosage on transcription is evident; the mean expression level of 1q and 16q genes was significantly increased or reduced in der(1;16)(+) cases compared to those without +1q or $-16q$, respectively. In particular, the mean expression level of *CDH1* in der(1;16)(+) cases was almost halved compared with that in those without $-16q$. However, the effect of haploid gain and loss was not confined to *CDH1* but also seen in other known/potential oncogenes and tumour suppressor genes, although the effect was highly variable across genes. For example, several critical tumour suppressor genes on 16q and oncogenes and putative oncogenes on 1q implicated in breast cancer development showed significantly reduced and increased expression in der(1;16)(+) breast cancer samples, respectively. These included *CBFB*, *CTCF*, and *FANCA* on 16q and *BCL9*, *USH2A*, *NCSTN*, *AKT3*, *RYR2*, *LAMC1*, and other genes on Wnt and NOTCH signalling pathways on 1q (**ED_Fig. 10k**). Thus, as is the case with other arm-level chromosomal abnormalities, it is difficult to determine the exact molecular targets of der(1;16), even though *CDH1* haploid loss could at least partly contribute to the positive selection driven by der(1;16). While to elucidate the mechanism of expansion of der(1;16)(+) clones is a key to the understanding of the breast carcinogenesis, particularly that of Luminal A tumours, it is too big a scope of this study but should be better addressed in the future studies.

We thank the reviewer for her/his thoughtful comments suggesting further analysis of the *CDH1* locus and the possible role of *CDH1* haploinsufficiency. On the basis of above results and discussion, the statement regarding the role of *CDH1* haploinsufficiency is too speculative to be based on experimental evidence. So we included these results in the revised manuscript, while the descriptions of the role of haploinsufficiency in discussion were deleted.

Lines 271–285:

“*CDH1* is a putative target of 16q deletion^{26,32} and was mutated in 86 (14.1%) of the 610 TCGA cancer cases. Among these, most cases (94%) had biallelic alteration, where the majority were associated with der(1;16) ($n=46$), followed by other 16q loss ($n=21$) and 16q UPD ($n=14$). Heterozygous mutation was found in only five cases. Except for one frameshift change, all were variants of unknown significance that were rarely registered in the COSMIC database. These findings suggest that *CDH1* is a bona fide recessive tumour suppressor gene, and haploid loss of *CDH1* alone may not be sufficient for positive selection/clonal expansion during breast cancer development. Regarding this, we analysed the effect of the allelic imbalance caused by der(1;16) on gene expression in 323 Luminal A cancer cases, including 103 der(1;16)(+) cases from the TCGA. The mean expression levels of 1q and 16q genes were significantly increased or decreased in der(1;16)(+) cases compared to those in cases without 1q gain or 16q loss, respectively (Extended Data Fig. 10k). In particular, the mean expression levels of *CDH1* in der(1;16)(+) cases without *CDH1* mutations were almost halved compared with those in cases without 16q loss. However, the effect of haploid gain and loss was not confined to *CDH1* but also seen in other known or putative oncogenes and tumour suppressor genes^{27,28}, although the effect was highly variable across genes. Thus, the exact molecular targets of der(1;16) are still elusive.”

It should be clearly stated in the text how many patients per cohort were microdissected in detail. For example, five were analysed from one cohort (line 142) but data for at least 7 women are shown in the main figures, plus other patients in the Extended figures.

Reply:

We apologise for the confusing descriptions. In the original manuscript, we analysed two sets of surgical samples from 8 patients:

#1. FFPE samples from 5 patients having cancer as well as widely distributed proliferative lesions over 3cm areas, which were obtained by reviewing pathology reports of 156 patients (**Fig. 2, ED_Fig. 5**).

#2. Fresh-frozen samples from 3 newly recruited patients, which were used to evaluate the size of expansion of non-cancer clones without der(1;16) (**Fig. 4, ED_Fig. 9**). Among these, one sample unexpectedly contained der(1;16) lesions (**ED_Fig. 9b,c**).

In addition, in this revision, we newly analysed following another set of samples, all carrying der(1;16) lesions, to validate the feature of der(1;16)(+) clones:

#3. 8 der(1;16)(+) FFPE samples that were obtained from 2 premenopausal and 6 postmenopausal patients by screening an independent cohort of 33 patients having Luminal A-like invasive cancer ($n=28$) or its putative precursor lesion (ER(+)/HER2(-)DCIS ($n=5$)) (**Fig. 3, ED_Figs. 7, 8**), which are known to be significantly enriched for der(1;16).

We revised the main text in detail so that it is clear what samples were microdissected as follows:

Lines 130–138:(Sample set #1)

“After reviewing pathology reports of 156 breast cancer patients, we found five patients for whom formalin-fixed paraffin-embedded (FFPE) specimens fulfilling the above-mentioned criteria had been preserved. We obtained multiple micro-scale samples (2.9mm (0.5–10) in diameter on average) from both cancer and non-cancer lesions (13.8/patient) using LCM (**Extended Data Fig. 3**). We analysed a total of 69 LCM samples for somatic mutations and copy-number alterations (CNAs) using WGS, based on which we reconstructed phylogenetic trees (**Methods**). These samples comprised histologically normal lobules ($n=6$), non-proliferative ($n=1$) and proliferative lesions ($n=33$), classic-type lobular carcinoma *in situ* (LCIS) ($n=1$), ductal carcinoma *in situ* (DCIS) ($n=20$), and invasive ductal carcinoma (IDC) ($n=8$). “

Lines 228–232:(Sample set #2)

“Thus, to see whether the large expansion of der(1;16)(+) clones could be explained only by the physiological enlargement of the growing mammary glands in this period, we evaluated the extent to which der(1;16)(-) non-cancer clones can expand after puberty. For this purpose, we microdissected multiple non-cancer lobules of the surgically resected and fresh-frozen specimens from three newly recruited premenopausal breast cancer patients (**Supplementary Table 1, Methods**).“

Lines 210–221:(Sample set #3)

“To confirm this, we screened another set of 33 specimens of Luminal A-like invasive cancer ($n=28$) or its putative precursor lesion (ER(+)/HER2(-) DCIS) ($n=5$) for der(1;16) using FISH and identified an additional eight der(1;16)(+) specimens, two from premenopausal and six from postmenopausal patients (**Fig. 3a, Extended Data Figs. 7,8**). As was the case with der(1;16)(+) clones in the index specimens (**Fig. 2a, Extended Data Fig. 5a–c**), which were all from premenopausal patients, the two der(1;16)(+) clones in premenopausal patients showed a macroscopic expansion over an area >20mm in diameter (**Fig. 3b, Extended Data Fig. 7**), supporting the above-mentioned hypothesis. By contrast, most of the remaining der(1;16)(+) clones from six postmenopausal patients were found in cancer lesions, rarely involving non-cancer lesions, and if ever, the surrounding der(1;16)(+) non-cancer lesions were confined within small lobules <10mm in diameter (**Fig. 3a,b, Extended Data Fig. 8a–e**). To exclude the possibility that this was due to the late acquisition of der(1;16), we estimated the timing of der(1;16) acquisition in five of the six postmenopausal patients based on phylogenetic analysis.”

A Table summarising the data from deep sequence analysis of multiple lesions from key patients would be helpful. Further, a Table documenting the types of atypia would be useful.

Reply:

We performed targeted capture sequencing for two purposes; 1) to identify the lesions sharing the same mutations with MRCA for an extended set of lesions that were not analysed in WGS and 2) to evaluate true positive rate in WGS mutation calling. The results of sequencing for the latter purpose was already shown in **Supplementary Tables 4, 5** in the original manuscript (**Supplementary Tables 14, 15** in the revised version). As per the request from the reviewer, we generated **Supplementary Table 9**, which summarises the data from targeted sequencing to identify the clones originating from the MRCA in key patients. While we already summarised the type of atypia in **Supplementary Table 2** in the original manuscript, we added more details of the types of atypia of microdissected samples, including benign lesions, in **Supplementary Tables 2**, according to the WHO classification.

The histopathology of the specimens is paramount to the paper: were the proliferative lesions defined by mitotic figures or was KI67 staining also performed? How were normal lobules distinguished from proliferative lesions without atypia?

Reply:

We did not evaluate mitotic figures nor Ki-67 staining to define the proliferative lesions. Mitotic counts and Ki-67 labelling index are generally very low in both proliferative lesions and normal lobules (Ellis IO *et al.*, *Mod Pathol*, 2010, doi: 10.1038/modpathol.2010.56; Huh SJ *et al.*, *Cancer Res*, 2016, doi: 10.1158/0008-5472.CAN-15-1927; Posso M *et al.*, *Cancer Med*, 2017, doi: 10.1002/cam4.1080) and no reliable consensus thresholds for these measures have been proposed to discriminate between proliferative lesions and normal lobules. Thus, for this purpose, there is no established and reliable diagnostic criteria other than the WHO classification, which relies solely on morphology but does not rely on mitotic figures or Ki-67 staining (**Review only table 1**). Therefore, in this study, we discriminate normal lobules from proliferative ones on the basis of the WHO classification (**Methods**). In the clinical setting, apart from the conceptual discussion about what is cancer, pathologists diagnose ‘breast cancer’ on the basis of the WHO classification and as such, it would be well-reasoned to use that criteria to ask the origin of cancer in the current study. Moreover, as long as we consider that proliferative lesions are not cancer, the discrimination between normal lobules and proliferative lesions does not affect our conclusion.

There are at least 4 different types of TDLUs that vary between individuals and with differentiation and menopause status. An entire sequence of lesions should be shown for 3 patients rather than showing a variety of sections from a range of patients in Ext data Fig 3.

Reply:

As per the reviewer's suggestion, we show representative histopathology of each of an entire sequence of different TDLU lesions for 3 patients, rather than collecting a variety of sections from different patients, in ED_Fig. 3c. We do agree that this is a better way of presentation.

Regarding the single cell organoid or 3D culture work: why do subclonal mutations arise given that these structures are not passaged- these are freshly plated cells that are cultured for a very short time and not passaged. In addition, please provide the breakdown of organoids derived from breast milk versus those from BC patients. Were any differences seen?

Reply:

Even in primary culture, subclones can arise by chance due to uneven cell divisions (typically 8-10 divisions) during an expansion of a single cell to several hundreds of cells. Actually, in previous studies, measurable subclonal mutations were observed in single cell-derived cell cultures of iPS cells (~7–10 days), and cells derived from the small and large intestines and the liver (~6 weeks) and primary esophageal epithelium (~2-3 weeks) most likely as a result of uneven cell divisions from a single cell (Kucab JE *et al.*, Cell, 2019, doi: 10.1016/j.cell.2019.03.001; Blokzijl F *et al.*, Nature, 2016, doi:10.1038/nature19768; Yokoyama A *et al.*, Nature, 2018, doi: 10.1038/s41586-018-0811-x). Of interest, those subclonal mutations were shown to be associated with a unique mutational signature (Kucab JE *et al.*, Cell, 2019; Blokzijl F *et al.*, Nature, 2016) (Reviewer only Fig. 1a). In the current study, we also detected a similar signature (Reviewer only Fig. 1a–c), which were specific to subclonal mutations and the number of subclonal mutations showed a positive correlation with cell culture period ($R^2=0.81$, $P=1.4 \times 10^{-6}$; Reviewer only Fig. 1d), supporting that they are mostly acquired during cell culture.

Reviewer only Fig. 1: Subclonal mutations in single cell-derived organoids

The breakdown of organoids derived from breast milk vs. those from BC patients were described in the main Fig. 1 (blue circles for breast milk, pink and purple for BC patients, respectively) as well as ED_Fig. 1c in this revision. We compared the mutation rate between healthy volunteers and premenopausal BC patients and found almost the same rate between the two (17.6/yr vs. 18.5/yr for SNV number ($P=0.42$), 1.1/yr vs. 1.3/yr for indel number ($P=0.28$)). In addition, the presence or absence of BC did not affect mutational burden when

an LME model was applied to the number of SNVs or indels ($P=0.71$ and 0.94 , respectively) (**Supplementary Table 4**).

An effect of menopause was seen in the 'organoid' cultures (line 110) for SBS5 mutations. The cohorts also differed in terms of menopause status (Ext Fig 2) - more detail on the effect of this on mutation accumulation/type amongst the different cohorts is required.

Reply:

LCM samples are usually polyclonal and the observed number of mutations therein is affected by the clonal composition of samples, which is not uniform across different samples, precluding an unbiased comparison between pre- and postmenopausal based on the analysis of LCM samples. For this reason, we evaluated the effect of different factors, including menopause, on the number of acquired mutations, using single cell-derived organoids.

Discussion Line 302- one primary factor (not listed here) for less mutation burden in postmenopausal tissue is that it has undergone involution and there is much less epithelial content, thus clones will have been extinguished.

Reply:

In this experiment, we estimated the number of mutations accumulated in a single cell using WGS of single cell-derived organoids. The estimate is therefore not affected by involuted or extinguished clones after menopause. As long as single cell-derived populations were analysed, we do not have to care about such extinguished or involuted clones.

Minor

points:

Fig 4a,b- the colors of the clones expanding in > 2 lobules and the superimposed mutations from the different lesions are not clear. These need to be revised.

Reply:

We agree with the reviewer's criticism and revised these figures so that they clearly show that many clones that appeared before 1 year of age occupied more than 2 lobules, whereas those arising after 13 years of age rarely involve ≥ 2 lobules (**Fig. 4a, ED_Fig. 9a,b**).

Please give more detail on the key features of SBS1 and SBS5 signatures.

Reply:

We added more details on the key features of SBS1 and SBS5 signatures in the main text as follows:

Lines 113–118:

"When fit to known Catalogue Of Somatic Mutations In Cancer (COSMIC) single base substitution (SBS) signatures, the vast majority of SNVs were assigned to three clock-like signatures²³, SBS1 (9.9%), SBS5 (80.7%), and SBS40 (9.4%). SBS1 is characterised by the prominence of C>T transitions at CpG dinucleotides resulting from the spontaneous deamination of 5-methyl-cytosine²⁴, while SBS5 and SBS40 are 'flat' signatures of unknown aetiology^{24,25}, which are difficult to separate from each other and hence, designated collectively as 'SBS5/40' in the subsequent analyses."

Referee #2 (Remarks to the Author):

In their study, Nishimura, Kakiuchi et al. present a reconstruction of the clonal history of breast cancer and related normal or neoplastic clones that reside nearby. To set a baseline of mutagenesis in normal breast tissue, they whole-genome sequence single-cell derived breast organoids and arrive at a mutation rate of roughly 19 SNVs per year prior to menopause. Subsequently, they interrogate the clonal history of breast cancer and nearby clones using laser capture microdissection followed by whole-genome sequencing, which has now become a common and reliable method to detect somatic mutations in normal tissues.

Echoing what has been seen in other normal tissues, apparently normal breast tissue harbours canonical cancers driver mutations, mainly in PIK3R1 and PIK3CA. The most unique feature of precancerous breast tissue identified in this study is the prevalence of a common unbalanced translocation, der(1;16). Early clones harbouring der(1;16) spread through the normal breast and forms a tissue bed from which multiple (pre-)cancerous lesions can emerge.

I find the study exceedingly clear, the findings interesting and the conclusions well-supported. Nevertheless, I would like to see some improvements and adjustments as outlined below.

Reply:

We are pleased that the reviewer finds the study exceedingly clear, the findings interesting, and the conclusions well-supported. We also thank the reviewer for his constructive suggestions, which we found extremely helpful to improve the manuscript substantially.

Comments

- The presence of SBS7a (and to a lesser extent, SBS8) is somewhat puzzling to me. Rather than actually being present in these samples, I suspect the emergence and fitting of these signatures may be artefactual. One way to test this is to interrogate dinucleotide variants. True SBS7a should be accompanied by numerous CC>TT dinucleotide changes (DBS1), whereas true SBS8 is usually associated with CC>AA dinucleotide changes. Assessing the presence of these mutations can lend weight to the observation that these signatures are present or, if these dinucleotide changes are not present, indicate that these signatures are not truly present in these samples.

- Given that these signatures are absent from the normal breast organoids but present in normal breast LCM cuts, could they in some way reflect artefactual mutations due to the LCM/FFPE experimental design? The validation experiment using targeted resequencing goes some way to assuage this, but from inspecting Supplementary Table 4, it appears the mutations to be validated all resided in genes and therefore might not be representative of the non-genic regions. Is this correct or am I misinterpreting the table?

- If it is the case that these signatures are not strongly supported by double base substitutions, I would advise to take a closer look at and revise the mutational signature analysis. From personal experience, the HDP package (<https://github.com/nicolaroberts/hdp>) has always yielded good mutational signature results (that might need further deconvolution), but I'll leave the particulars to the authors to pursue as they see fit.

Reply:

According to the reviewer's suggestions, we re-evaluated mutational signatures in the LCM samples.

To summarise:

- 1) These SBS7a- and SBS8-like signatures were most frequently observed in FFPE/LCM samples (**Reviewer only Fig. 2a**).
- 2) As the reviewer predicted, we did not see any association with these dinucleotide changes that real SBS7a and SBS8 signatures should have accompanied (**Reviewer only Fig. 2b**).
- 3) SigProfiler and MutationalPatterns produced fairly concordant results, which however, were not reproduced in the analysis using the HDP package, where larger fractions were explained by SBS1/5 signatures, while the SBS8- and SBS7a-like signature were not prominent but instead, SBS18 and in some cases, SBS16 accounted for the remaining fractions (**Reviewer only Fig. 2a**).

- 4) Of note, however, we obtained very high validation rates for shared SNVs in the main trunk (9,318/9,393, 99.2%) and private mutations in peripheral branches (158/162, 97.5%) in the phylogenetic trees, which contains 6.9% and 11.1% of mutations assigned to SBS7a- or SBS8-like signatures, respectively (**Reviewer only Fig.2a**). Note that the validation experiments were performed for almost all shared SNVs (9,393/9,766, 96.7%), and randomly selected private SNVs ($n=162$) from both genic and non-genic regions as long as DNA was available for sequencing (**Supplementary Table 14**).

Reviewer only Fig. 2: SBS signatures and DBS mutation types in FFPE LCM samples

Taken together, these results support the reviewer’s prediction that SBS7a- and SBS8-like signatures should be artefacts. However, considering high validation rates for mutation calling, we speculate that they are likely to represent real nucleotide substitutions actually present in the samples, which are most likely associated with formalin-fixation and/or LCM.

Because many of these artefacts are known to have hotspots (Do H *et al.*, Clin Chem, 2015, doi: 10.1373/clinchem.2014.223040), we can eliminate, at least partially, such artefacts by increasing the number of ‘reference’ samples that our mutation callers (Genomon2, Mutect2, and Strelka2) used to filter artefacts. Thus, in this revision, we newly analysed SNVs in an additional 22 DNA samples obtained from surrounding normal interstitial tissues, which were used as the reference to filter such artefacts. When used in the three mutation callers, the newly generated references effectively worked to reduce artefacts. The number of called mutations reduced from 55,052 to 53,806, where we no longer detected SBS8-like signatures in any algorithms. Although the SBS7a-like signature was still seen in SigProfiler and HDP, it disappeared in MutationPatterns (**ED_Fig. 4**). In this revision, we adopted the results from MutationalPatterns, which provides the simplest explanation of the mutation process using SBS1, 5/40, 2, and 13. We updated **Online Methods** accordingly.

- Doubts about SBS7a and SBS8 notwithstanding, for the purpose of timing MRCAs and copy number gains (including der(1;16)), it would be best to exclusively use mutations that can be assigned to the clock-like

signatures, SBS1 and SBS5, for the best estimate, given only those signatures are present in the normal breast organoids. Including sporadic signatures in this analysis would alter the timing estimate.

Reply:

We agree with the reviewer's comment. The problem, however, is now less relevant, because after filtering artefacts as described above, all the mutations in the main trunk, or those acquired until the emergence of MRCA, are exclusively assigned to SBS1 or SBS5/SBS40. Thus we can estimate the timing of the acquisition of der(1;16) and that of the emergence of MRCA using all mutations.

- Is there any evidence for haploinsufficiency of BRCA2 in the patient with a BRCA2 germline mutation? While this could have manifested as SBS3, which is not identified in the patient within the study, this phenotype could also manifest as an excess of indels following ID6.

Reply:

Yes. As many as 29–36% of breast cancer patients with germline *BRCA2* mutations develop without acquiring second hit mutations (Maxwell LM *et al.*, Nat Commun, 2017, doi: 10.1038/s41467-017-00388-9; Inagaki-Kawata Y *et al.*, Nat Commun, 2020, doi: 10.1038/s42003-020-01301-9), suggesting haploinsufficiency of *BRCA2* in breast cancer development. In addition, SBS3 signature is reported to be seen even in breast/ovarian cancers with monoallelic *BRCA2* alteration, although it is less prominent compared with those with biallelic alteration, also supporting haploinsufficiency of *BRCA2* (Maxwell LM *et al.*, Nat Commun, 2017).

To our knowledge, there have been no publications reporting the analysis of mutational signature in normal mammary epithelium. In response to the reviewer's suggestion, we checked whether or not *BRCA2* haploinsufficiency could manifest itself as an excess of SBS3 and/or ID6 signatures by deconvoluting mutations and indels detected in normal mammary epithelial cells in a patient with a heterozygous germline *BRCA2* mutation (KU1195), but did not detect SBS3 or ID6 signature, even though both signatures were successfully detected in the breast cancer sample carrying biallelic *BRCA2* alterations (**Reviewer only Fig.3**).

Reviewer only Fig. 3: Signature analysis in non-cancer lesions with a pathogenic germline *BRCA2* variant

- Is there any effect of carrying a cancer driver mutation on the mutation burden of the normal breast glands sampled? I.e., do those with a *PIK3R1/PIK3CA* mutation have more somatic mutations than those without? From a quick visual inspection, this seems to be the case for KU1206 in Fig. 4a and the organoids in Fig. 1d, but it would be nice to make this more substantial, if possible.

Reply:

This is one of the important issues we were not able to address in the first submission. As evident from low variant allele frequencies (VAFs) of mutations in normal LCM samples (average median VAF: 0.26 (0.17-0.47)) (Fig. 4d), many LCM samples are likely to be polyclonal, which prevents an accurate estimation of the mutation burden per single clone/cell. In this revision, we addressed this problem by using driver-mutated single cell-derived organoids, instead of using LCM samples. Despite a relatively low frequency of driver-mutated organoids, we found 4 organoids carrying a *PIK3CA* mutation among a total of 64 organoids we established, which were used to investigate the effect of driver mutations on the mutation burden according to the LME model. Based on the LME modelling incorporating known/putative variables that could affect the mutation burden (Methods), *PIK3CA* mutations ($P=4.2 \times 10^{-3}$) are shown to significantly increase the number of somatic mutations, while menopause and parity also have negative impacts on the mutation number (Fig. 1b,c), although the result needs to be verified including more driver-mutated clones. We added this result in the revised manuscript as follows:

Lines 118–124:

“According to the linear mixed-effects (LME) model, the number of SNVs significantly depended on age at sample collection, years after menopause, parity, and the presence of a driver mutation. SNVs were accumulated at 19.5 mutations/genome/year before menopause, which was reduced to 7.1 mutations/genome/year after menopause, while the mutation number was reduced by 50.4/delivery (Fig. 1b,c). The mutation rate was also affected by *PIK3CA* mutations, which increased the number of SNVs by 150.5, although this needs to be validated using additional *PIK3CA*-mutant clones, because the number of driver-mutated samples was still small ($n=4$).”

In addition, the text mentions the presence of driver mutations leads to a higher clonality (l. 242-244). It would be good to see a quantification of this and test it statistically.

Reply:

We provided the *P*-value to support significantly larger VAFs for driver-mutated LCM samples, compared with those without driver mutations in the main text as follows:

Lines 239–242:

“The presence of driver mutations was associated with a higher clonality as suggested by a significantly larger median VAF of mutations in driver-mutated versus unmutated samples (0.33 versus 0.25, $P=1.8 \times 10^{-3}$), supporting the role of driver mutations in positive selection (Fig. 4d).”

- An additional analysis for this study is to estimate telomere length from the WGS. This could support the notion laid out in the discussion, that the decrease in mutation rate after menopause, as well as the reduction of mutations due to parity, is related to a decreased rate of cell turnover/a population of cells dormant stem cells becoming reactivated after pregnancy. Telomere length was also used as a line of evidence in the cited tobacco smoking study.

Reply:

This is a good point. According to the reviewer’s suggestion, we tried to measure the telomere lengths based on the WGS data of whole genome-amplified (WGA) DNA from single cell-derived organoids, using Telomerecat (v.3.4.0, <https://github.com/cancerit/telomerecat>. Farmer JHR *et al.*, Sci Rep, 2018, doi: 10.1038/s41598-017-14403-y). However, we barely detected telomere reads in WGS data (Reviewer only Fig. 4a). We speculated that telomere sequences were not successfully amplified in WGA. To test this, we measured telomere lengths in WGA samples using an RT-qPCR-based method (Absolute Human Telomere Length Quantification qPCR Assay Kit, #8918, ScienCell). Telomere sequences were successfully amplified from

non-amplified original DNA, but not from WGA samples (**Reviewer only Fig.4b**). Because the majority of samples from single cell-derived organoids were WGA, we were not able to evaluate telomere length.

Reviewer only Fig. 4: Analysis of telomere length in WGA samples

Minor comments

- “Given that ... Luminal A cancer.” (l. 354-358). I am not sure I follow this. Would the difference in der(1;16) clone size between pre- and postmenopausal women simply not reflect the time at which these were acquired? I.e., the earlier the acquisition of der(1;16), the larger the size of the clone?

Reply:

Thank you for the important question. To test whether the smaller size of postmenopausal der(1;16)(+) clones just reflect the later timing at which their der(1;16) was acquired, we estimated the timing of the acquisition of der(1;16) in 5 postmenopausal patients with der(1;16)(+) breast cancer, in which multiple LCM samples were analysed using WGS, followed by phylogenetic analysis. Importantly, the mean age of the acquisition of der(1;16) in the 5 postmenopausal patients was estimated as 11.7 years old (0-18.7), which is comparable to 10.7 years old (5.9-16.8) in 5 premenopausal patients ($P=0.58$). This suggests that der(1;16) was acquired around puberty/late adolescence, even in tumours in postmenopausal patients. Also considering the lack of non-cancer proliferative lesions or normal lobules having der(1;16), we speculate that there should have been a larger expansion of der(1;16)(+) clones with variable histology before menopause, which however, regressed after menopause in the face of a reduced oestrogen level. We revised the main text and figures, accordingly.

Lines 166–176:

“In particular, the timing of the acquisition of der(1;16) was more accurately pinpointed than that of other driver events, by maximising the posterior probability of the observed numbers of duplicated and unduplicated mutations on 1q arm in der(1;16)(+) MRCA (**Extended Data Fig. 6a–d, Methods**). On an average, der(1;16) in six clones was estimated to be acquired at 10.7 (range, 5.9–16.8) years of age (**Fig. 2a, Extended Data Fig. 5a–c**). We also estimated the average timing at which MRCA emerged as 26.5 (range, 18.1–34.4) years of age, assuming a constant mutation rate until the emergence of the MRCA. For example, two distinct der(1;16) detected in a 48-year-old woman (KU779) were estimated to occur in two mammary cells at the age of 5.9 and 10.0 years, respectively (**Fig. 2a**). These ancestor cells then gave rise to the MRCAs at the age of 18.1 and 22.3 years, respectively, from which a number of non-cancer progenies evolved, followed by the appearance of cancer founders at least >10 years after the initial acquisition of der(1;16).”

Line 220–226:

“To exclude the possibility that this was due to the late acquisition of der(1;16), we estimated the timing of der(1;16) acquisition in five of the six postmenopausal patients based on phylogenetic analysis. Of interest,

the mean age of the acquisition of der(1;16) in the five postmenopausal patients was estimated as 11.7 years (0–18.7), which is comparable to the 10.7 years (5.9–16.8) ($P=0.58$) in premenopausal patients (**Fig. 3c,d**). Thus, we speculate that there should have been a larger expansion of der(1;16)(+) clones, including non-cancer lesions, before menopause, which however, regressed after menopause in the face of reduced oestrogen levels.”

- A study that echoes the findings in this paper of a tissue root leading to multiple cancer lesions is a study on precursors of Wilms tumour and associated benign lesions (PMID: 31806814), fuelled by somatic loss of imprinting of a locus on 11p. Since this is an epigenetic change, this study underscores the point in the discussion that genomic changes may not explain all of carcinogenesis. In addition, this point is also supported by similar findings in malignant rhabdoid tumour, where the precursor clone was found to be genomically virtually identical to the cancer (PMID: 33658498). These are simply some examples of similar processes observed in other cancers and precancerous lesions, biased to childhood cancers because of my background, and form by no means an exhaustive list.

Reply:

Thank you for the important comment and the suggestion of another study supporting the role of epigenetic driver events in cancer development. The phylogenetic analysis using whole genome sequencing provided a unique opportunity to compare the genetic events between cancer and non-cancer clones within the same breast tissue. The comparison revealed no correlation between histology and the number of known driver events/CNAs. This may suggest a possible role of epigenetic driver events and/or microenvironments in the development of cancer, although we cannot completely exclude the presence of still unknown genetic changes that escaped from WGS. Incorporating the reviewer’s suggestion, we revised the corresponding sentences as follows:

Line 303–306:

“Another finding of interest is the lack of consistent correlations between histologies and the number/type of driver events. Although we cannot exclude the possibility of the presence of undetected driver mutations and structural variations, this may suggest the role of epigenetic changes³⁶ and/or locally defined microenvironments in cancer development.”

- Fig 1d: it is difficult to distinguish the colours for SBS1 and SBS5, so I advise using a colour palette that is easier to tell apart. This of course, pertains to all the figures using these colours for SBS1 and SBS5.

Reply:

We agree with the reviewer’s comments and revised the use of the colour panel so that the readers easily distinguish SBS1, SBS5/40 and other signatures.

- Fig 2a: it would be nice to add confidence intervals to the estimate of the der(1;16) timing (such as those presented in Fig 3e) as a bar alongside the branch. This pertains to Fig 3a-c as well.

Reply:

We agree that this is a better presentation of the der(1;16) timing. According to the reviewer’s suggestion, we calculated the 95% CI of the estimated timing of the acquisition of der(1;16) in terms of the number of mutations in **Figs. 2a, 3a, ED_Figs. 5, 8a–d, 9b**.

- ED Figure 9: the MutationalPatterns plots seem to have a dark blue bar that is unexplained by the legend. What is this signature? It would also be good to use a different colour for it.

Reply:

We were sorry for the lack of explanations in the legend. Dark blue bars in Extended Data Fig.9a indicated SBS40. We revised the figure as **ED_Fig.4a**, using different colour panels so that the readers easily distinguish each SBS signature, and made sure all the legends are in place.

I congratulate the authors on an interesting study, it was a joy to read the manuscript.

Reply:

We humbly appreciate the reviewer's commendation of our study.

Referee #3 (Remarks to the Author):

In this manuscript by Nishimura and Kakiuchi et al. claim to show the entire life history of breast cancer from its origin to clinical diagnosis. To accomplish this, the authors reconstruct phylogenies from WGS data derived from microdissected FFPE samples of 5 patients, complemented by mutation rate estimation based in numerous clonally-derived organoids from normal and malignant breast tissue. While the topic is an interesting one and the extensive sampling of lesions across the breast make for a rather unique (albeit heterogeneous) dataset, the primary finding that mutant clones arise early and independently in the breast around puberty and decades before cancer development is not novel. This is by now expected given reports in other tissues. The analyses used to arrive at this conclusion are overly complex while relying on standard techniques such as phylogeny reconstruction, mutational signature analysis. Unfortunately, because the presentation of the findings lacked clarity, the reader is left with far more questions than answers and uncertainty regarding the claims.

Reply:

We are sorry for the misleading descriptions in the original manuscript. We do agree with the reviewer in that mutant clones arise early and independently in the breast around puberty and decades before cancer development is not novel. An early origin of initial cancer mutations in cancer has been inferred by a number of studies using multiple sampling of cancer (Nik-Zainal S *et al.*, Cell, 2012, doi: 10.1016/j.cell.2012.04.023; Mitchell TJ *et al.*, Cell, 2018, doi: 10.1016/j.cell.2018.02.020; Williams N *et al.*, Nature, 2022, doi:10.1038/s41586-021-04312-6) and also anticipated in recent studies on cancer mutations in normal and pre-cancer tissues (Yokoyama A *et al.*, Nature, 2018, doi: 10.1038/s41586-018-0811-x; Moore L *et al.*, Nature, 2020, doi: 10.1038/s41586-020-2214-z). However, it is still unknown when cancer clones arise by what mutations during cancer evolution, while other clones stay on normal or pre-cancer, because these studies were performed solely on cancer samples or normal/non-cancer tissues alone, or the phenotype of analysed samples (for example blood colonies) was unknown. Moreover, the inference of the exact timing of early events based on cancer samples is obscured due to the lack of knowledge about the mutation accumulation in corresponding normal tissues. In this meaning, the entire life history of cancer has been poorly understood.

In the current study, by analysing clonal/near clonal samples with varying histologies, including both cancer, benign breast lesions (BBL), and histologically normal lobules, we have successfully elucidated the entire picture of evolution from the acquisition of first putative drivers to clinically diagnosed cancer, for one of the most common breast cancer subtype characterised by der(1;16). The chronology of early events is better estimated on the basis of the mutation rate measured for single cell-derived organoids from normal mammary epithelium. Combined, these analyses led to a number of novel or unexpected findings:

- 1) The unique pattern of breast cancer evolution in a major breast cancer subtype that is characterised by:
 - a) Frequent association with der(1;16), which is uniformly acquired around puberty or in late adolescence (6-17), followed by the emergence of a 'non-cancer' common ancestor in late 20's- early 30's and the evolution of independent cancer founders thereafter.
 - b) Unexpectedly large expansion of non-cancer clones sharing the same ancestor with cancer lesions before cancer diagnosis.
 - c) Frequent evolution of multiple independent cancer founders from common non-cancer ancestors, uniquely contributing to intra tumour heterogeneity.
 - d) Poor association between phenotype (cancer vs. BBL) and driver genetic events, suggesting the role of epigenetic events and/or microenvironments.

These findings could not be obtained without analysing both cancer and non-cancer clones at the same time.

In addition, through the analysis of single cell-derived organoids and LCM samples from normal breast tissues, we also revealed:

- 2) The unique profile of mutation accumulation in the mammary epithelium that is distinct from that in other tissues. The mutation accumulation in breast tissues synchronises with the women's life cycle and is significantly affected by the parameters related to known breast cancer risks, such as delayed menopause and parity. This also provided the basis for estimating the timing for driver events to occur.
- 3) Pervasive mutations of *PIK3CA* and *PIK3R1* genes involving apparently normal mammary lobules.

We believe that these findings provide new insight into the breast cancer pathogenesis and early diagnosis, prevention of breast cancer.

Meanwhile, we do agree with the reviewer's criticism that the presentation of the findings lacked clarity. Thus, in this submission, we fully revised the abstract and the main text, as well as figures and supplementary materials. We hope that the revised manuscript successfully answers the reviewer's concerns and helps the reviewer appreciate the novelty and significance of our study.

Major Comments:

The manuscript itself is difficult to follow, lacking clarity on numerous points – but most crucially the key messages from the study. From the abstract, “Here we show the entire life history of breast cancer from its origin to clinically diagnosed cancer using whole-genome sequencing (WGS) followed by phylogenetic analysis of multiple microdissected samples of genetically related cancer and non-cancer clones.” What are the authors trying to claim? Are they characterizing normal tissue? Is the study longitudinal? The abstract, introduction, and summary of findings/discussion seem quite disconnected. As a result, the reader is left to interpret this themselves.

Reply:

Thank you for these critical comments. We are sorry for the lack of clarity, particularly with regard to the key messages we want to deliver. In response to the reviewer's criticism, we fully revised the manuscript to make the key messages clear.

As stated in the answer to the general comments above, our primary purpose was to elucidate the entire life history of breast cancer, by addressing the key questions that previous studies on normal/pre-cancer tissues cannot answer: when cancer arises from their non-cancer ancestors by acquiring what mutations, while other related clones were still normal or remained pre-cancer, and what is the difference in mutations between cancer clones and their non-cancer relatives? Phylogenetic analyses using multi-sampling of cancer specimens have been used to infer the life history of cancer in terms of driver events. However, the analysis of cancer tissue alone frequently obscures the order of early driver events that are often assigned together to a long major trunk in the phylogenetic tree (Nik-Zainal *S et al.*, *Cell*, 2012, doi:10.1016/j.cell.2012.04.023; Yates LR, *et al.*, *Nat Med*, 2015, doi:10.1038/nm.3886; Gundem G *et al.*, *Nature*, 2015, doi:10.1038/nature14347). Moreover, it does not help map the timing at which phenotypically cancer clones emerged or track the fate of other related non-cancer clones. To answer these questions, the analysis of both cancer and non-cancer lesions is absolutely needed, although this is frequently hampered by the fact that at the time of cancer diagnosis or surgery, related non-cancer clones are likely swept out by rapidly expanded cancer clones and no longer present. In the current study, we performed phylogenetic analysis including both cancer and non-cancer tissues, taking advantage of breast cancer specimens showing unique histology containing both cancer and precancerous lesions.

The novel findings in this study were derived from this unique study design, which to our knowledge, had not been employed before to decipher cancer history. These findings were summarised in the response to the reviewer's general comments (please see above) and highlighted in the discussion section in the revised manuscript.

The introduction poorly overviews the field of somatic mutations in normal and pre-cancerous tissue which has exploded in the past years. One of the chief findings of this manuscript is that oncogenic drivers emerge at a relatively early age (in the discussion it is noted that this may occur during embryogenesis although

elsewhere the times center on puberty). However, this is not a novel finding as numerous tissues such as the epidermis, esophagus, endometrium, and hematopoietic system have been demonstrated to harbor somatic alterations in non-malignant, phenotypically normal cells. It would be helpful to frame the current findings in light of what is already known and to discuss how this study differs in approach and/or main findings.

Reply:

Thank you for these suggestions. First, we are sorry for the introduction that poorly overviewed the field of somatic mutations in normal and pre-cancer tissues because of the strict limitation in the word count. However, for the purpose of highlighting the significance of the current study, we believe that the short sentences does not fail to provide the key concept derived from these studies:

“In view of cancer development, a key observation through these studies is that clonal outgrowth in normal or non-cancer tissues is quite common, often pervasive, and frequently driven by common cancer mutations¹. This immediately points to an important implication to the early history of cancer that one or more of those positively selected clones should be destined for subsequent cancer development^{1,3}.”

We are also sorry that the reviewer feels that the major claims in the present study are unclear. The early emergence of oncogenic drivers is not novel and not the main claim in this study. Again, our main claims are summarised in the response to the reviewer’s general comments (please see above). Life history of cancer has been investigated using multiple sampling studies for many types of cancers. As stated in the response to the previous comments, the analysis of cancer tissue alone frequently obscures the order of early driver events that are often assigned together to a long major trunk in the phylogenetic tree. Moreover, it does not help map the timing at which phenotypically cancer clones emerged or track the fate of other related clones. To answer these issues, the analysis of both cancer and non-cancer lesions is absolutely needed. Our findings underscore the importance of including both cancer and related non-cancer lesions together. We summarised these points, what is new and what is already known, and what are the problems in previous studies on cancer history in Introduction and Discussion.

Line 74–84: (Introduction)

“Here among key questions that studies on normal tissues cannot answer are: when cancer arises from these non-cancer clones by acquiring what additional mutations, while other clones partially sharing common mutations are still normal or pre-cancer, and what is the difference in mutation profile between cancer clones and those non-cancer relatives? Phylogenetic analyses using multi-sampling of cancer specimens have been used to infer the life history of cancer in terms of driver events. However, the analyses of cancer tissue alone frequently obscure the order of early driver events that are often assigned together to a long major trunk in the phylogenetic tree⁹⁻¹¹. Moreover, it does not help map the timing at which cancer clones emerged or track the fate of other related non-cancer clones. To answer these issues, the analyses of both cancer and non-cancer lesions are absolutely needed, although these are frequently hampered by the fact that at the time of cancer diagnosis or surgery, genetically related non-cancer clones are likely swept out by rapidly expanded cancer clones^{8,12,13}.”

Line 287–295: (Discussion)

“Through phylogenetic analyses, we successfully traced the evolution of breast cancer and precursor lesions, from the acquisition of initial driver alterations to the development of clinically diagnosed disease. The absolute timing and the order of early driver events were more accurately estimated than in previous studies^{9-11,33,34} by analysing both cancer and non-cancer lesions and by using the rate of mutation accumulation measured for normal mammary epithelium. As demonstrated in a recent study on myeloproliferative neoplasms (MPN)³⁵, the first driver events occurred long before the cancer diagnosis, around puberty or late adolescence, or in one case, as early as in early infancy. However, unlike the case with the MPN study, discrimination between cancer and non-cancer clones along the evolutionary tree was enabled to some time point after the acquisition of initial driver events.”

The most interesting finding seems to be the presence of mutant clones harboring known driver alterations such as der(1;16) throughout the mammary gland. Presumably these alterations arising independently at different times and strongly selected for in this tissue and clonally expand relative to neighboring cells that lack this event. This leads to the hypothesis that multiple cancer founders can exist, contributing to genetic heterogeneity in the subsequent cancer. While potentially appealing, this requires further support and clarification. Is this only true of multi-focal cancers such as those selected here. Is this unique to der(1;16) harboring lesions because of loss of E-cadherin?

Reply:

Thank you for the insightful comments and important questions, which are essentially asking to what extent this hypothesis could be applicable to breast cancer development in general. In this study, we found a widespread expansion of clones in all 5 specimens initially selected for analysis. However, this observation could be biased by the choice of those specimens showing multiple satellite lesions that were large enough (≥ 3 mm in diameter) to obtain a sufficient amount of DNA for WGS (**Lines 128–132**). In fact, 4 out of the 5 specimens harboured der(1;16), suggesting that this mode of cancer development might be characteristic of der(1;16)(+) cancers.

To evaluate this, in the original manuscript, we surveyed surgical specimens from an additional, unselected 33 patients with Luminal A-like IDC/DCIS for the presence of der(1;16) using FISH analysis, which identified a total of 8 der(1;16)(+) specimens from 2 premenopausal and 6 postmenopausal patients. We confirmed the widespread expansion of cancer and/or proliferative lesions carrying der(1;16) in the two premenopausal cases. Combined with another der(1;16)(+) clone incidentally identified in KU1215 (**ExD_Fig. 9b,c**), a total of 9 independent der(1;16) clones identified in 7 premenopausal patients (**Fig. 3b, ED_Fig. 9c**) had a widespread expansion, where multiple cancer founders were confirmed in 3 patients. The expansion of der(1;16)(+) lesions was less extensive in 6 postmenopausal specimens. Of interest, however, even in these cases, der(1;16) was estimated to be acquired around puberty in 5 postmenopausal cases (**Fig. 3c,d**). Thus, it is speculated that in these postmenopausal cases, der(1;16)(+) cancer and non-cancer lesions might have expanded before menopause but regressed thereafter, likely due to reduced oestrogen levels. Taken together, the widespread clonal expansion harbouring multiple cancer founders is considered to be a common feature of the development of der(1;16) positive breast cancers, although this might not totally be unique to der(1;16)(+) lesions but could also be found in other subtypes, at least in an *AKT1*-mutated case (KU582) (**ED_Fig. 5d**).

Meanwhile, it is difficult to answer whether the hypothesis of multiple cancer founders is only true of multi-focal cancers such as those selected here, because without multifocal cancer lesions, we cannot demonstrate multiple cancer founders. However, it is of interest to ask whether this is a common pattern of breast cancer development in der(1;16)-negative breast cancers. In fact, we detected similar widespread proliferative lesions in almost all premenopausal Luminal A-like IDC/DCIS specimens, of which 73% (8/11 cases) had multiple cancer lesions, suggesting the possibility that some of these widespread lesions were clonal. To confirm this, we need to analyse many der(1;16)-negative cancers. Unfortunately, this is not feasible for technical reasons, because recovering enough DNA from microdissected FFPE archives frequently causes problems in this experimental design. This would be out of the scope of this study and should be addressed in the future. We described new data regarding der(1;16) clones and discuss these points in the revised manuscript as follows:

Lines 209–226:

“The unexpected enrichment of der(1;16) in the five index cases suggested that the widespread expansion of satellite lesions of varying histology was a common feature of der(1;16)(+) breast cancer. To confirm this, we screened another set of 33 specimens of Luminal A-like invasive cancer ($n=28$) or its putative precursor lesion (ER+)HER2(-) DCIS ($n=5$) for der(1;16) using FISH and identified an additional eight der(1;16)(+) specimens, two from premenopausal and six from postmenopausal patients (**Fig. 3a, Extended Data Figs. 7,8**). As was the case with der(1;16)(+) clones in the index specimens (**Fig. 2a, Extended Data Fig. 5a–c**), which were all from premenopausal patients, the two der(1;16)(+) clones in premenopausal patients showed a macroscopic

expansion over an area >20mm in diameter (**Fig. 3b, Extended Data Fig. 7**), supporting the above-mentioned hypothesis. By contrast, most of the remaining der(1;16)(+) clones from six postmenopausal patients were found in cancer lesions, rarely involving non-cancer lesions, and if ever, the surrounding der(1;16)(+) non-cancer lesions were confined within small lobules <10mm in diameter (**Fig. 3a,b, Extended Data Fig. 8a–e**). To exclude the possibility that this was due to the late acquisition of der(1;16), we estimated the timing of der(1;16) acquisition in five of the six postmenopausal patients based on phylogenetic analysis. Of interest, the mean age of the acquisition of der(1;16) in the five postmenopausal patients was estimated as 11.7 years (0–18.7), which is comparable to the 10.7 years (5.9–16.8) ($P=0.58$) in premenopausal patients (**Fig. 3c,d**). Thus, we speculate that there should have been a larger expansion of der(1;16)(+) clones, including non-cancer lesions, before menopause, which however, regressed after menopause in the face of reduced oestrogen levels.”

Lines 307–319:

“It should be noted that such a unique pattern of cancer evolution could be biased by the selection of specimens harbouring multiple satellite BBL lesions for LCM, which was highly enriched for der(1;16). The analysis of an additional cases with der(1;16) confirmed that the presence of persistent non-cancer clones in a large area is an intrinsic feature of der(1;16)(+) breast cancer at least in premenopausal cases. The parallel evolution of multiple independent der(1;16) clones in two cases supports the strong driver role of der(1;16) in puberty or late adolescence. Accounting for 20% of all breast cancers and one-third and two-thirds of luminal A and invasive lobular breast cancers, respectively, der(1;16) defines a major subtype of breast cancers. However, it is still open to question whether or not this pattern of cancer evolution is also common in other breast cancer subtypes. It was observed at least in an *AKT1*-mutated case (KU582). Mutations affecting *PIK3CA* and *PIK3R1* are among the most frequent targets of somatic mutations in breast cancer^{27,28} and also common in apparently normal mammary lobules (10/66 lobules) (**Fig. 4a, Extended Data Fig. 9a,b**). However, none of the clones carrying these mutations showed a widespread expansion. Further investigations are needed to clarify this.”

Additionally, how do these observations relate to the findings of Erickson et al (Nature 2022) who used spatial transcriptomics and copy number inference to examine benign and malignant prostate tissue. One might anticipate some parallels in multi-focal prostate lesions.

Reply:

In their paper, Erickson et al. demonstrated an expansion of non-cancer clones detected by one or more copy number abnormalities (CNAs), from which a cancer clone evolved by acquiring additional CNAs (Erickson A *et al.*, Nature, 2022, doi: 10.1038/s41586-022-05023-2). Similar observations were also reported in breast cancer (Newburger DE *et al.* Genome Res, 2013, doi:10.1101/gr.151670.112; Ang D.C *et al.*, Mod Pathol, 2014, doi:10.1038/modpathol.2013.197), as described in Introduction L81–87 in the original manuscript. However, in these studies, with the lack of detailed analysis of somatic mutations, it is largely unknown how many and what types of genetic lesions were acquired in what order and timing until cancer was diagnosed. A recent study also revealed the entire history of myeloproliferative diseases (MPN), successfully estimated the initial timing of the acquisition of *JAK2* mutations and inferred the dynamics of clones. However, with the lack of information about the phenotype of each colony, the emergence of cancer cannot be mapped onto the phylogenetic trees, leaving it undetermined when cancer appeared. The current study successfully addresses these issues. Analysing both cancer and non-cancer lesions with *varying* histology using WGS, we have comprehensively detected somatic mutations and CNAs in each clone to clarify the order and the timing of the acquisition of mutations, the impact of mutations/CNAs on histology, and the way intratumor heterogeneity is established.

Beyond the challenges with the text, the main figures are cluttered, overly reliant on text, and lack broad themes/takeaways to support the authors primary findings. Figure 1 includes methods that don't belong in the main text or that could be summarized at a higher-level view with details to be included in the methods section and supplemental figures. The schematics of tissue-samplings (Fig 3 etc) are useful, but distracting

and don't seem to provide much insight into the findings that accompany them. One or two could be used as an example but there are far too many too meaningfully digest or contrast. There is also a lack of consistency in the legends leaving the reader with yet more questions than answers about the study.

Reply:

Thank you for these comments and suggestions. We agree that “the main figures are cluttered, overly reliant on text, and lack broad themes/takeaways to support our primary findings”. In response to the reviewer’s criticism, we fully reconstruct the main figures. For example, the figures explaining the methods (**Fig. 1a–c**) are moved to **ED_Figs. 1, 2**. Also we present the schematics of tissue-samplings and phylogenetic trees only for one representative case carrying der(1;16)(**Fig. 2**), while those for the remaining three cases are moved to **ED_Fig. 5**. We carefully checked the consistency in the legends for all figures and the inconsistent descriptions were amended, accordingly.

Throughout the text, the authors discuss clones as being cancerous and yet this is not defined clearly. There are not cancerous clones, but rather clones that exist within normal, pre-cancerous, or cancerous tissue. Presumably, the authors are referring to “clones carrying breast cancer mutations” as noted initially in the abstract. There are numerous instances throughout the manuscript that would benefit from clarifying this.

Reply:

Thank you for this suggestion and we are sorry for the lack of clarity regarding the definition of cancer and non-cancer clones. In the introduction (**Lines 72–77**), we are using “cancer clones” and “non-cancer clones”, assuming that the readers will understand what “cancer” means, because otherwise we could not discuss the evolution of cancer. Meanwhile, when we refer to particular clones within real samples (**for example, Lines 161–165, Lines 190–195, and Lines 198–201**), we are consistently using “cancer” and “noncancer” clones as those that exist in ‘histologically confirmed’ cancer and non-cancer tissues/samples, respectively (**ED_Fig. 3**). In addition, we can also reasonably define all ancestors of clones within histologically confirmed non-cancer tissues/samples as non-cancer clones, because ancestors of normal clones cannot be cancer. Actually, these are the only practical ways of defining clones in real samples. We carefully revised the manuscript to make the meaning of cancer and non-cancer clones clear in every context they appear. Despite the use of “cancer clones” and “non-cancer clones” in two different contexts, we believe that no confusions arise to interpret the manuscript.

To clear the misunderstanding, we never defined “cancer clones” as “clones carrying breast cancer mutations” or used the term “cancer clones” in such a meaning. In particular, when we stated in the abstract that “evolution of clones carrying breast cancer mutations is common in apparently normal mammary epithelium”, we do not mean that those clones are cancerous. On the contrary, most of those clones evolved in normal/pre-cancer tissues will not progress to cancer (Kakiuchi N et al., Nat Rev Cancer, 2021, doi:10.1038/s41568-021-00335-3; Yokoyama A et al., Nature, 2019, doi:10.1038/s41586-018-0811-x).

The methods as written lack the necessary information to reproduce these findings and warrant sufficient additional information as well as a github repository for the code used in analysis. This is especially true for the primary analysis constructing patient phylogenetic trees. There is no information provided on these phylogenies in terms of homoplasy or branch support.

Reply:

We are sorry for insufficient information necessary to reproduce the findings. In response to the reviewer’s criticism, we almost fully updated the method section, particularly the description of phylogenetic analysis.

1) All the softwares and algorithms used in this study were publicly available and are summarised in the **Online Methods** section and **Reporting Summary** with URLs. Other private codes newly generated to implement these softwares and algorithms in pipelines are provided as an R markdown file in **Supplementary Information**.
2) We provide more detailed methods to construct phylogenetic trees in **Online Methods (Lines 892–944)**. Homoplasy was observed with regard to cancer phenotypes in multiple independent branches in each phylogenetic tree, which was indicated by red (IDC) and pink (DCIS). As described in **Online Methods**, we generated phylogenetic trees using MEGA, which determine branches using a bootstrap method. To guarantee

the correct assignment of branches, we added bootstrap values supporting the observed branches for all phylogenetic trees.

There is also little information on how SNV number is converted to chronological age.

Reply:

As seen from the highly variable numbers of SNVs observed for different LCM samples, the mutation rate is not constant across different branches in phylogenetic trees but variably inflated, particularly for cancer lesions, which had a much larger number of SNVs than expected from the measurement of normal organoids. Thus, it is impossible to uniformly correct the inflation across all lesions and branches. Nevertheless, we are still able to estimate the timing for early events, such as the acquisition of der(1;16) and *AKT1* mutation and the emergence of non-cancer MRCA, for which we could apply an approximately constant mutation rate estimated from normal organoids. This is particularly true of the estimation of the timing of the acquisition of der(1;16), which is thought to be the first genetic event in most cases. We described the detailed method of the estimation in **Online Methods (Lines 1025–1051)** and **ExD_Fig. 4** in the original manuscript. However, it depends on the timing of MRCA and was not directly applied to the calculation of the timing of der(1;16) acquisition in postmenopausal cancer samples, which is newly included in the revised manuscript. In these samples, there were no non-cancer lesions and we were able to analyse only cancer samples. Thus the most recent MRCA already had a larger number of SNVs than expected from patients' age and unreasonably overestimated the timing of the acquisition of der(1;16). To avoid this difficulty, we modified the method of estimating der(1;16) timing that did not depend on the number of SNVs in MRCA but solely depended on the number of duplicated SNVs on 1q+. We described the revised method in **Online Methods** in the revised manuscript (**Lines 969–992**). The codes used for the estimation were also provided in **Supplementary Note 4**.

It seems that after the age of ~50 the number of observed SNVs plateaus. Does this have an impact on the chronological ages?

Reply:

We did not have to take into account the reduced rate of mutation accumulation, because the estimated timing of the acquisition of der(1;16) (11.8 years old, range: 0-19.8) and the emergence of non-cancer MRCA (26.5 years old, range: 18.1-34.4) was much earlier than the age at menopause (~50 years of age).

How do the authors explain the lack of additional SNVs being gained with age in the postmenopausal samples (Fig 1d)?

Reply:

The reduction in mutation rate after menopause is highly significant in multivariable analysis using 23 premenopausal and 24 postmenopausal organoids, which was further confirmed by newly including 17 organoids (9 premenopausal and 8 postmenopausal organoids). The mutation rate is reduced from 19.5/genome/year to 7.1/genome/year but does not become completely plateau. Although the exact reason for the reduced mutation rate is unknown, we speculate that this is related to the reduced cell cycling of mammary gland cells and/or reduced oestrogen levels after menopause. Before menopause, mammary glands repeat proliferation and regression in every menstrual cycle, which might contribute to mutation accumulation (Ramakrishnan R *et al.*, *Mod Pathol*, 2002, doi: 10.1097/01.MP.0000039566.20817.46.; Maria Navarrete AHN *et al.*, *Breast Cancer Res*, 2005, doi: 10.1186/bcr994.). Oestrogen is a well-known mutagen.

From a technical perspective, it seems this could be due to inadequate sampling/coverage of very small clones. Perhaps this could be addressed via deep targeted sequencing?

Reply:

We respectfully disagree with the reviewer's comment. We evaluated the effect of menopause (and other factors) exclusively using single cell-derived organoids (**Fig. 1a, ED_Fig. 1**), simply because LCM samples are not clonal, where it is difficult to correctly determine the number of mutations in a single clone. Thus, all the samples used to estimate the mutation rate are monoclonal and therefore, we expected high sensitivity to

detect clonal SNVs. In fact, according to the simulation using germline SNPs, the sensitivity of detecting SNVs are 93.3% and 91.2% for pre and postmenopausal organoids (**Supplementary Table 13**).

Additionally, the fact that there are many more mutations present in the FFPE vs FF and organoid samples raises the concern that these are largely artifactual. While not surprising as this has been repeatedly seen, the impact on the conclusions/timing estimates is not addressed and potentially concerning.

Reply:

We respectfully disagree with the reviewer's comment. Compared with FF LCM and organoid samples which were largely derived from normal lobules, the vast majority of FFPE samples were taken from proliferative and cancer lesions. Therefore, it is well anticipated that the FFPE samples had a much larger number of mutations compared with fresh-frozen and organoid samples. In fact, the number of mutations detected in 6 FFPE samples from normal lobules was 399.3 (59–609) (KU779-#1/#5/#6 and KU582-#4/#9/#10; **Fig. 2a, ED_Fig. 5d**), which is comparable to the number of mutations detected in FF LCM samples, 481.3 (22–962) (**Fig. 4a, ED_Fig. 9a,b**). Moreover, we also validated the shared mutations in the main trunk, which was used to estimate the timing of der(1;16) and MRCAs and confirmed a very high validation rate (99.2%) (**Online Methods, Supplementary Tables 14,15**). Thus, it is unlikely that artefacts caused by the use of FFPE samples impact on the conclusions/timing estimates.

The ordering of clonal/subclonal events could be examined more extensively through the incorporation of copy number information. However, it is hard to tell the extent to which this information should be included (or not) due to the lack of summary information provided in the primary figures or quality control provided in the methods/figures. Nonetheless given the known role of copy number in breast pathogenesis, it would be important to investigate this.

Reply:

We apologise for the lack of sufficient information about the methods of phylogenetic analysis to determine the order of clonal/subclonal events. In the original submission, we incorporated copy number information to reconstruct phylogenetic trees but did it only in an incomplete or ad hoc manner. In this revision, in response to the reviewer's suggestion, we redid the phylogenetic analysis, incorporating copy number information more consistently and extensively. Primary figures and Online Methods were updated, accordingly. We are sorry that we could not understand what 'summary information' and 'quality control' exactly mean. However, we added the information that summarises the quality control of the results, including the accuracy of the assignment of mutations to present/absent for MEGA input (Methods and Supplementary information), the bootstrap values from MEGA at each branch point (**Figs. 2a, 3a, 4a, ED_Figs. 5, 8a–d, 9a,b**), and the concordance between MEGA/treemut vs. Pyclone-VI (**Supplementary Note 1**). The major changes in the phylogenetic analysis are summarised below:

We reconstruct phylogenetic trees, combining somatic mutation data and copy number information across all LCM samples. To accomplish this, we first determined the branching pattern and branch length of the phylogenetic tree based on the maximum parsimony using MEGA (v.11.0.11) (Tamura K *et al.*, Mol Biol Evol, 2021, doi: 10.1093/molbev/msab120) and then assigned all mutations to individual branches in the tree using an R package, 'treemut', which did this based on an expectation maximisation (EM) algorithm-based approach. Copy number information was required to accurately estimate the mutation state, i.e., 'mutation_present', 'mutation_absent', or 'mutation_unknown', for each mutation in each LCM sample, which were combined for all samples and summarised in a mutation matrix for an input for MEGA. Copy number information was also needed for the treemut input (see below).

A) Generation of the mutation matrix for MEGA

An input for MEGA comprises a list of all mutations detected across all samples, combined with their mutation state, i.e., 'mutation_present', 'mutation_absent', or 'mutation_unknown', for all samples, which was estimated/determined according to the depth, the number of supportive reads, and the copy number status. The mutation status was determined in a conservative manner, because false assignments would prevent a stable estimation of tree structure.

For a given mutation,

1. For all samples with ≥ 2 supportive reads, the mutation status was assigned to 'mutation_present', because variants with ≥ 2 supportive reads were rarely observed in control samples ($< 11,298/6,576,276$ (0.17%)), regardless of copy number state.
2. For all samples with only 1 supportive read, the mutation status was assigned to 'mutation_unknown', because it was impossible to correctly determine the real mutation state of these mutations for these samples, regardless of copy number state.
3. For those samples with no supportive reads,
 - The mutation status was assigned to 'mutation_unknown', when **chromosomal loss or other LOH** was present at the mutation locus,
 - The mutation status was also assigned to 'mutation_unknown', when no supportive read for the mutation was well expected ($P > 0.05$) according to the binomial distribution determined by sequencing depth, **total copy number**, and mutant cell fraction (MCF) estimated from a Gaussian mixed model (Online methods for detail),
 - Otherwise, the mutation status was assigned to 'mutation_absent'.

We confirmed high validation rates for those mutations assigned to 'present' or 'absent' according to these criteria using validation sequencing for 780 randomly selected mutations in two cases, wherein mutation status in 2 and 9 samples, respectively, were evaluated for each mutation (accuracy: 99.4% (3,055/3,072 mutation statuses)) (**Online Methods, Supplementary Table 5**).

B) Assignment of mutations to tree branches using treemut

We used an R package 'treemut' to assign each mutation to a branch by an EM method based on the number of supportive reads and the sequencing depth for all potential mutations for all samples, as well as the tree information from MEGA (branching pattern and branch length). However, because treemut was originally developed for the analysis of monoclonal diploid samples and assumes that VAF=0.5 for mutated loci and VAF=0 for wild-type loci, we corrected variant read counts based on adjusted VAF (aVAF), MCF, total copy numbers (TCN), and minor copy numbers (MCN) for bulk LCM samples, as if they were consisted of a clonal population derived from a single cell (Online Methods). Briefly,

1. Mutant allele number (MAN) is calculated as follows;
$$\text{MAN} = \text{VAF} \times (\text{MCF} \times \text{TCN} + (1 - \text{MCF}) \times 2) / \text{MCF}$$
2. Adjusted VAF (aVAF) is calculated using MAN, TCN, and MCN (**Online Methods**).
3. Corrected variant read counts were calculated by Depth \times aVAF

C) Assignment of CNAs

Excluding der(1;16), a total of 57 CNAs were detected in 34 samples, including 18 with CN gain, 7 with UPD, and 21 with CN loss (**Supplementary Table 6**). Losses or gains of different paternal/maternal alleles as determined by SNP analysis were considered different events. Most of the CNAs were found in isolated samples ($n=47$) and if not, shared by two to five samples each in 6 cases: 8p loss in samples #8, 12, and 18 in KU582, Chromosome 3 gain in samples #11a–e in KU779, Chromosome 10 gain in samples #3, 5, and 6 in KU957, 14q gain and 22q loss in all three samples and subsequent 14q UPD in samples #a and c in TMA114, focal 6q loss in all three samples in TMA125, and focal 5q losses in all three samples in TMA149 (**Supplementary Table 9**). The former CNAs were assigned to the peripheral branch corresponding to the isolated sample, while the latter CNAs were assigned to the branch shared by these two to five samples.

D) Validation of the phylogenetic trees

To validate the method of phylogenetic analysis described above, we reconstructed the phylogenetic trees using an independent algorithm that incorporate copy number information, i.e., Pyclone-VI (v0.1.0) (Gillis *et al.*, BMC Bioinformatics, 2020, doi: 10.1186/s12859-020-03919-2) in two representative cases that accompanied copy number abnormalities (KU779 and KU539). Then the trees from both methods were

compared in terms of the overall topology and composition of driver mutations of corresponding branches (**Supplementary Note 1**). PyClone-VI estimates the clusters of mutations shared by clones/subclones or 'branch' in the tree and the samples that share each cluster on the basis of MCF, the number of mutant and wild-type reads, and TCN/MCN at the mutation locus for all mutations in all samples. These clusters or trunks were then visually ordered so that the samples in a parent branch comprise those in their child branches across all branchpoints (**Supplementary Note 1 for details**). Because there was a limitation in the number of clusters PyClone-VI can analyse at a time, we first separated the trees reconstructed by MEGA11/treemut into 4–5 parts and performed the validation using PyClone-VI for each part. Specifically, we separated each tree into 4 clades corresponding to the top 4 branches and applied PyClone-VI for each clade. Pyclone analysis was also performed for 4 samples chosen from each of the 4 clades by randomly selecting one sample from each clade. We confirmed that except for 3 peripheral branches, 22 out of the 25 branch points were matched between two algorithms. In particular, the branches to which driver mutations were assigned were completely matched. Based on these results, we considered that the results from MEGA/treemut were reproduced by PyClone-VI (**Supplementary Note Figure 1**).

Supplementary Note Figure 1: Trees reconstructed using MEGA/treemut and PyClone-VI

Minor Comments:

The authors have taken great care to disclose the software versions used for their bioinformatics analysis; however, there is inconsistent citations provided for the tools that are used. Some examples: BWA line 843, biobambam line 844, and GenomonMutationFilter line 849 are all missing a citation, but samtools, Xenome, and others have citations present. All tools and pipelines used should be cited appropriately.

Reply:

We apologise for the inconsistent citations. We carefully checked the manuscript for consistency and confirmed that all the softwares used in this study were properly cited with version in **Online Methods** as they first appeared and be summarised in **Reporting Summary** with version and URL information.

Units are not properly reported leading to confusion throughout the manuscript (e.g., extended data figure 4 "...were increased by 0.1 year to...").

Reply:

We confirmed proper use of units and corrected errors throughout the manuscript. As for the issues specifically raised by the reviewer, 0.1 years are correct. Actually, we simulated the timing of der(1;16) acquisition by moving the simulated value from 0 years old to the age of MRCA appearance by 0.1 years for 1,000,000 times.

Grammatical errors throughout.

Reply:

The manuscript was proofread by a native speaker. We also carefully check the grammatical errors throughout the manuscript.

Reviewer Only Table 1: Diagnostic criteria of benign lesions according to WHO classification

Classification in this study	Diagnosis (WHO)	Diagnostic criteria (WHO)
Non-proliferative lesions	Fibroadenoma	A circumscribed breast neoplasm arising from TDLU (terminal duct lobular unit), and featuring a proliferation of both epithelial and stromal elements. The admixture of stromal and epithelial proliferation gives rise to two distinct growth patterns.
	Columnar cell change (CCC)	A lesion of the TDLU that is characterised by enlarged, variably dilated acini lined by columnar epithelial cells that frequently have apical cytoplasmic snouts, in which the epithelial-cell lining is only one or two cell layers thick.
Proliferative lesions without atypia	Usual ductal hyperplasia (UDH)	A lesion characterised by a solid or fenestrated proliferation of benign epithelial cells that often show streaming growth, particularly in the centre of involved spaces. The epithelial cells display a haphazard orientation with respect to one another. The presence of secondary lumina or fenestrations is characteristic of this lesion. The lumina are often peripherally located and tend to be slit-like, as opposed to the very rounded, punched-out lumina seen in ADH and low-grade DCIS.
	Columnar cell hyperplasia (CCH)	A lesion of the TDLU that is characterised by enlarged, variably dilated acini lined by columnar epithelial cells that frequently have apical cytoplasmic snouts, with cellular stratification or tufting more than two cell-layers thick.
	Sclerosing adenosis	A lesion that is composed of a compact proliferation of acinar structures with preservation of the luminal epithelial and the peripheral myoepithelial cell layers together with an investing basement membrane.
	Radial scar	A lobulocentric proliferation that contains benign changes that may include cysts, UDH and sclerosing adenosis, which has a stellate outline with central dense hyalinized collagen and elastosis.
	Intraductal papilloma	A benign lesion that is characterised by finger-like fibrovascular cores covered by an epithelial and myoepithelial cell layer.
Proliferative lesions with atypia	Flat epithelial atypia (FEA)	A neoplastic alteration of the TDLUs characterised by replacement of the native epithelial cells by one to several layers of a single epithelial cell type showing low-grade (monomorphic) cytological atypia.
	Atypical ductal hyperplasia (ADH)	A proliferation of monomorphic, evenly placed epithelial cells involving TDLUs. The epithelial cells lack the streaming, swirling, and overlapping of the cells that define UDH. The cellular monotony and architectural patterns are similar to those seen in low-grade DCIS; however, the proliferation in ADH is either admixed with a second population of non-uniform cells in TDLU spaces or it completely involves a limited number of those spaces.

	Atypical lobular hyperplasia (ALH)	A lesion characterised by a proliferation of generally small, non-cohesive cells, in which uniform cells are present without distorting the involved acini.
LCIS	Classic LCIS	A lesion characterised by a proliferation of generally small, non-cohesive cells, in which more than half of the acini of a lobular unit are distended and distorted by a dyshesive proliferation of the uniform cells.

Reviewer Reports on the First Revision:

Referees' comments:

Referee #1 (Remarks to the Author):

The authors have substantially clarified the figures and manuscript, and addressed the majority of points raised. Although the prevalence of the der(1;16) translocation in precancerous tissue is one of the most unique features of this study, the next step in identification of the main driver/epigenetic mutations is challenging (and beyond scope of this work) given the emergence of multiclonal cancer lesions from this antecedent alteration.

Referee #2 (Remarks to the Author):

The manuscript by Nishimura, Kakiuchi et al. has been significantly improved upon revision, with both figures and text more lucidly conveying the messages and novelty of the study. I appreciate the authors taking the comments to heart, especially regarding the mutational signature analyses and telomere length estimation, despite the latter revealing WGA yields data unusable for analysis of telomeric regions. I have no further comments.

I believe this study will be of great interest to the field and further adds to our understanding of somatic evolution in normal, pre-cancerous and ultimately, cancerous settings.

Tim Coorens

Referee #3 (Remarks to the Author):

In their revised manuscript Nishimura et al. address a number of concerns raised at initial review through additional analyses, documentation and clarification of their cohort and methods. I commend them on their thorough responses. Importantly, the phylogenetic analyses now systematically incorporate CNVs, which represent key events during breast tumorigenesis. Additionally, the new chronological timing estimates no longer depend on SNV burden derived from non-cancer samples, which would likely contribute to inflated values. It is reassuring to clarify that nearly all FFPE samples were malignant, and it is therefore not surprising that SNV burden is substantially higher than in the clonal organoids. While the authors note that it is technically challenging to obtain adequate LCM-derived DNA, comparisons to non-malignant FFPE LCM samples from the same donor would enable calibration of mutation burden and alleviate concerns about artifactual variants. This should be a future goal as technologies improve, as the authors note.

In general, the main methodological issues have been resolved and/or their limitations clarified. With these changes to the text and simplification of the figures, the manuscript is significantly improved and adds to our understanding of early tumor evolution.